# Human Cognition-Inspired Hierarchical Fuzzy Learning Machine

**Junbiao Cui** [1] **Qin Yue** [1] **Jianqing Liang** [1] **Jiye Liang** [1]

## Abstract

Classification is a cornerstone of machine learning research. Most of the existing classifiers assume that the concepts corresponding to classes can be precisely defined. This notion diverges from the widely accepted understanding in cognitive science, which posits that real-world concepts are often inherently ambiguous. To bridge this big gap, we propose a Human Cognition-Inspired Hierarchical Fuzzy Learning Machine (HC-HFLM), which leverages a novel hierarchical alignment loss to integrate rich class knowledge from human knowledge system into learning process. We further theoretically prove that minimizing this loss can align the hierarchical structure derived from data with those contained in class knowledge, resulting in clear semantics and high interpretability. Systematic experiments verify that the proposed method can achieve significant gains in interpretability and generalization performance.

## 1. Introduction

Classification stands as one of the most fundamental and extensively studied problems in machine learning. Over the years, a vast array of classifiers has been developed (Delgado et al., 2014), with their methodologies broadly categorized into two main approaches. The first approach relies directly on the discrete 0-1 loss, as seen in classifiers such as k-nearest neighbors (Cover & Hart, 1967), decision trees (Quinlan, 1986), and naive Bayes (Domingos & Pazzani, 1997), etc. The second approach employs continuous functions as proxies for the 0-1 loss, including the hinge loss used in support vector machine (Cortes & Vapnik, 1995), the exponential loss adopted by AdaBoost (Freund & Schapire, 1997; Friedman et al., 2000), and the

cross-entropy loss widely adopted in deep neural networks (LeCun et al., 2015), etc. Despite their differences, these loss functions share a common underlying assumption, i.e., the concepts corresponding to classes can be precisely defined. This assumption inherently implies that class relation are crispy, where any two classes are either identical or distinct, leaving no room for ambiguity.

In the classification problem, each class corresponds to a concept. Unlike existing classifiers, human solves the classification problem mainly based on concept cognition (Murphy, 2004). Over the centuries, the classical theory of concept representation prevailed, positing that all concepts can be precisely defined. However, in 1953, the philosopher Ludwig Wittgenstein challenged this proposition by exploring the concept of "game", questioning whether a precise definition of such a concept truly exists (Wittgenstein, 1953). This seminal inquiry sparked a paradigm shift, leading the philosophy community to widely accept that real-world concepts are often inherently ambiguous and resist precise definition.

Building on the above consensus, modern cognitive science has developed several influential theories for concept representation. Prototype theory (Rosch, 1975) models a concept using an idealized prototype that encapsulates its most representative features. Exemplar theory (Medin & Schaffer, 1978) represents a concept through a set of typical and representative instances. Knowledge theory (Murphy & Medin, 1985) posits that concepts are deeply embedded within human knowledge system, asserting that human cognition of a concept is inseparable from the contextual and relational structure of this system. As the most sophisticate framework, knowledge theory not only aligns closely with human cognitive processes but also provides a compelling foundation for advancing machine learning models that aim to mimic human-like cognition.

Recently, we proposed a fuzzy learning machine (FLM) inspired by concept cognition (Cui & Liang, 2022). FLM leverages fuzzy set theory (Zadeh, 1965) to capture the inherent fuzziness of concepts (McCloskey & Glucksberg, 1978; Marti et al., 2023) and employs exemplar theory to capture the typicality effects (Smith et al., 1974), achieving good interpretability, robustness, and generalization.

However, FLM still faces limitations in understanding con-

[1] Key Laboratory of Computational Intelligence and Chinese Information Processing of Ministry of Education, School of Computer and Information Technology, Shanxi University, Taiyuan 030006, Shanxi, China. Correspondence to: Jianqing Liang <liangjq@sxu.edu.cn>, Jiye Liang <ljy@sxu.edu.cn>.

*Proceedings of the 42nd International Conference on Machine Learning*, Vancouver, Canada. PMLR 267, 2025. Copyright 2025 by the author(s).

cepts. While it employs exemplar theory for concept representation, an approach well-suited to purely data-driven machine learning. Unlike knowledge theory (Murphy & Medin, 1985), it separates concept representation from human knowledge system, limiting its ability to form a profound understanding of concepts. Recent research in cognitive science further highlights that many concepts are understood through their relations with other concepts (Piantadosi et al., 2024).

Meanwhile, the community has accumulated a wealth of digitized knowledge base, including the conventional knowledge graph, e.g., WordNet (Miller, 1995), ConceptNet (Speer et al., 2017), and the fashionable pre-trained language model, e.g., Word2Vector (Mikolov et al., 2013), GloVe (Pennington et al., 2014), and GPT (OpenAI, 2023), etc. These resources make it feasible to develop the knowledge-driven method for concept cognition.

Accordingly, this paper explores relations between concepts contained in human knowledge system and then leverages these relations to guide the learning of concepts from data, enhancing the model's ability to understand and represent concepts. The main contributions are as follows.

- We propose a unified modeling framework that represents various types of knowledge as fuzzy similarity relation (FSR) on the class space, resulting in the knowledge-infused concept representation.

- We design a novel hierarchical alignment (HA) loss inspired by the principles of concept cognition, enabling the knowledge-infused concept representation to guide the learning process.

- We introduce the FSR-based quotient space theory, and then prove that minimizing the HA loss can align the hierarchical structures derived from both data and knowledge in quotient space.

- Extensive experiments verify the effectiveness of the proposed method in improving interpretability and generalization performance.

## 2. Notions and Related Works

### 2.1. Problem Formalization

***Task.*** Given a $(\mathcal{X}, \mathcal{Y}, f_c)$-classification problem, $\mathcal{X} \subseteq \mathbb{R}^d$, $\mathcal{Y}$, and $f_c : \mathcal{X} \rightarrow \mathcal{Y}$ are input space, class space and unknown classification function, respectively. To solve the classification problem, we need to determine the unknown classification function $f_c$. For the sake of discussion, the classes in $\mathcal{Y}$ are numbered as $1, 2, \cdots, |\mathcal{Y}|$.

***Data.*** Let $\mathcal{D} = \mathcal{X} \times \mathcal{Y}$ be data space. In machine learning, experience is often given in the form of training data, de-noted as $D = \{(x_i, y_i) \,|\, x_i \in \mathcal{X}, \, y_i = f_c(x_i) \in \mathcal{Y}\}_{i=1}^n \subset \mathcal{D}$. The goal is to use $D$ to get $\hat{f}_c : \mathcal{X} \rightarrow \mathcal{Y}$ as close as possible to the unknown $f_c$. $\forall k \in \mathcal{Y}$, let $X_k = \{x_i \,|\, (x_i, y_i) \in D, \, y_i = k\}$ be the set of training samples belonging to class $k$. Let $X = \{x_i \,|\, (x_i, y_i) \in D\}$ be the set of all training samples.

***Class knowledge.*** Let $\mathcal{K}$ be human knowledge system, which contains various types of knowledge. In practice, each class in $\mathcal{Y}$ corresponds to a concept, and the concept name corresponds to a word in natural language. Therefore, we can collect the class knowledge about $\mathcal{Y}$ by concept name, denoted as $K \subset \mathcal{K}$.

### 2.2. Preliminaries

Fuzzy similarity relation is the core notion of the proposed method. Specifically,

**Definition 2.1.** (Bandler & Kohout, 1988) Given a set $A$, the mapping $F : A \times A \rightarrow [0, 1]$ is called a fuzzy binary relation on $A$.

**Definition 2.2.** (Bandler & Kohout, 1988) Given a set $A$ and a fuzzy binary relation $F$ on $A$. If $F$ satisfies (1) reflexivity: $\forall a \in A, R((a, a)) = 1$, and (2) symmetry: $\forall a, b \in A, R((a, b)) = R((b, a))$, the $F$ is called a fuzzy similarity relation (FSR) on $A$.

More preliminaries, including binary relation, fuzzy binary relation, and fuzzy quotient space are given in **Appendix** A.

### 2.3. Fuzzy Learning Machine

In FLM (Cui & Liang, 2022), we achieve concept cognition according to exemplar theory. Specifically,

***Learning.*** The training process of FLM is

$$\min_{\Theta} \sum_{i,j=1, i \neq j}^n \mathcal{L}_{\text{FP}}\left(f\left((x_i, x_j)\,;\Theta\right), y_i, y_j\right) + \gamma \mathcal{R}(\Theta), \quad (1)$$

where $\mathcal{L}_{\text{FP}}$ and $\mathcal{R}$ are fuzziness permissible loss and regularization term, respectively. $\gamma > 0$ is trade-off parameter. $f\left((\cdot, \cdot)\,;\Theta\right)$ is a FSR on $\mathcal{X}$ with learnable parameters $\Theta$. The fuzziness permissible (FP) loss is defined as

$$\mathcal{L}_{\text{FP}}(s_{ij}, y_i, y_j) = \begin{cases} \max(s_{ij} - \alpha, 0), & y_i \neq y_j \\ \max(\beta - s_{ij}, 0), & y_i = y_j \end{cases}, \quad (2)$$

where $s_{ij} = f\left((x_i, x_j)\,;\Theta\right)$ is the predicting similarity between $x_i$ and $x_j$. $0 < \alpha < \beta < 1$ are hyper-parameters to control the degree of concept fuzziness.

***Concept representation.*** Let $\Theta^\circ$ be the local optimal solution of formula (1). The FSR $f\left((\cdot, \cdot)\,;\Theta^\circ\right)$ forms the basis of concept representation. According to exemplar theory, the set of exemplars of class $k$ is defined as

$$E_k = \left\{ \begin{array}{l} x\,|\, x \in X_k, \mu(x, k) \text{ is the top-}n_k^{\text{exe}} \\ \text{largest value in } \{\mu(x_i, k)\,|\, x_i \in X_k\} \end{array} \right\}, \quad (3)$$

where $\mu(x, k) = \frac{1}{|X_k|} \sum_{x_j \in X_k} f((x, x_j); \Theta^\circ)$ and $n_k^{\text{exe}}$ is a manually specified parameter.

**Predicting.** Given a test sample $x \in \mathcal{X}$, FLM predicts the class label of $x$ according to the following formula

$$\hat{y} = \underset{k \in \mathcal{Y}}{argmax} \frac{1}{|E_k|} \sum_{x_i \in E_k} f((x, x_i); \Theta^\circ). \quad (4)$$

To sum up, FLM captures the fuzziness of concept well. However, it ignores relations between concepts contained in human knowledge system, limiting its ability to concept cognition. This paper aims to alleviate the limitation.

# 3. Proposed Method

The overall framework of the proposed HC-HFLM is shown in Figure 1, including method design and theory analysis. The method design is given below, and the theory analysis will be given in **Section** 4.

## 3.1. Method Design

We first introduce the following two definitions.

**Definition 3.1.** Given $\alpha \in (0, 1)$, a set $A$, and a FSR $S$ on $A$. If $\forall a, b \in A, a \neq b, 0 < S(a, b) < \alpha$, then $S$ is called a $\alpha$-FSR on $A$.

**Definition 3.2.** Given a finite set $A = \{a_i\}_{i=1}^{|A|} \subset \mathbb{R}$. $\forall a_i \in A$, let $sort(a_i; A) \in \{1, 2, \cdots, |A|\}$ be the rank of $a_i$ in $A$ by ascending order. Let $\mathbf{b} = sortvec(A) = (b_1, b_2, \cdots, b_{|A|})^T \in \mathbb{R}^{|A|}$ be the vector formed by arranging the elements in ascending order $A$. Obviously, the following propositions are true.
(1) $\forall a_i, a_j \in A, i \neq j, a_i > a_j$ iff $sort(a_i; A) > sort(a_j; A)$.
(2) $\forall a_i \in A, a_i = b_{sort(a_i; A)}$.
(3) $\forall i = 1, 2, \cdots, |A|, b_i \in A$ and $sort(b_i; A) = i$.

### (1) Mining Class FSR from Human Knowledge System

According to knowledge theory (Murphy & Medin, 1985), to understand concepts profoundly we need to obtain the knowledge-infused concept representation from human knowledge system $\mathcal{K}$. It is well known that $\mathcal{K}$ contains various types of knowledge. In this paper, we do not limit the type of class knowledge $K \subset \mathcal{K}$. It can be the attribute description vector of concept name used in literature (Lampert et al., 2014). It can be the embedding vector of concept name by pre-trained language models, such as Word2Vector (Mikolov et al., 2013), GloVe (Pennington et al., 2014), GPT (OpenAI, 2023), etc. It can be the knowledge graph, such as WordNet (Miller, 1995), ConceptNet (Speer et al., 2017), etc. It can also be human themselves.

To ensure the universality, different types of class knowledge $K$ are uniformly modeled as a $\alpha$-FSR on $\mathcal{Y}$, i.e., $\mathbf{S}^K \in$

$(0, 1]^{|\mathcal{Y}| \times |\mathcal{Y}|}$, where $\forall i, j \in \mathcal{Y}$,

$$s_{ij}^K = \begin{cases} 1, & i = j \\ sim(i, j; K), & i \neq j \end{cases}, \quad (5)$$

where $sim(i, j; K) \in (0, \alpha)$ is the similarity between classes $i$ and $j$ measured by class knowledge $K$. $\alpha$ is fuzziness parameter. **Appendix** B.1 gives several types of class knowledge and the corresponding $sim(i, j; K)$.

In practice, the quality of class knowledge $K$ varies significantly, so we need to further refine the class FSR. Let

$$\mathbf{T} = \begin{cases} \mathbf{S}^K, & \text{case } 1: K \text{ is of high quality} \\ \mathbf{S}^{(D,K)^*}, & \text{case } 2: K \text{ is of medium quality} \\ \mathbf{S}_{\text{coa}}^{(D,K)^*}, & \text{case } 3: K \text{ is of low quality} \end{cases} \quad (6)$$

be the refined class FSR, where $\mathbf{S}^K$ is obtained by formula (5). case 1: $\mathbf{S}^K$ needs no refinement. case 2: $\mathbf{S}^K$ needs to be calibrated by training data. **Appendix** B.2 gives the procedure for computing $\mathbf{S}^{(D,K)^*}$. case 3: $\mathbf{S}^K$ needs to be coarsened and then be calibrated by training data. **Appendix** B.3 gives the procedure for computing $\mathbf{S}_{\text{coa}}^{(D,K)^*}$.

In formula (6), $\mathbf{T}$ is a symmetric matrix. Each row of $\mathbf{T}$ is a $|\mathcal{Y}|$-dimensional embedding representation of one concept, which is composed of relations between concepts. Meanwhile, $\mathbf{T}$ is mined from human knowledge system. Therefore, $\mathbf{T}$ can be regarded as the knowledge-infused concept representation.

### (2) Learning Sample FSR from Data

We design a novel hierarchical alignment (HA) loss, which uses class FSR $\mathbf{T}$, the knowledge-infused concept representation, to guide the learning of sample FSR from training data, enhancing model's ability to concept cognition.

**HA loss.** Let $\mathbf{T}$ be the $\alpha$-FSR obtained by formula (6). Let

$$\begin{aligned} V &= \{t_{ij} | \ i, j \in \mathcal{Y}\}, \\ U &= V \cup \{0, \alpha, \beta\}, \\ \mathbf{u} &= sortvec(U) = (u_1, u_2, \cdots, u_{|U|})^T, \end{aligned} \quad (7)$$

where $0 < \alpha < \beta < 1$ are fuzziness parameters. $\forall (x_i, y_i), (x_j, y_j) \in D, x_i \neq x_j$, let $s_{ij} = f((x_i, x_j); \Theta)$ be the predicting similarity between $x_i$ and $x_j$. Based on formula (7), the HA loss is defined as

$$\mathcal{L}_{\text{HA}}(s_{ij}, y_i, y_j) = \begin{cases} u_{r_{ij}-1} - s_{ij}, & s_{ij} \leq u_{r_{ij}-1} \\ 0, & s_{ij} \in (u_{r_{ij}-1}, u_{r_{ij}}) \\ s_{ij} - u_{r_{ij}}, & s_{ij} \geq u_{r_{ij}} \end{cases}, \quad (8)$$

where $r_{ij} = sort(t_{y_i y_j}; U)$. The underlying cognitive principles are as follows.

(a) In the ideal case, samples $x_i$ and $x_j$ can perfectly represent concepts $y_i$ and $y_j$, respectively. The similarity $s_{ij}$ can

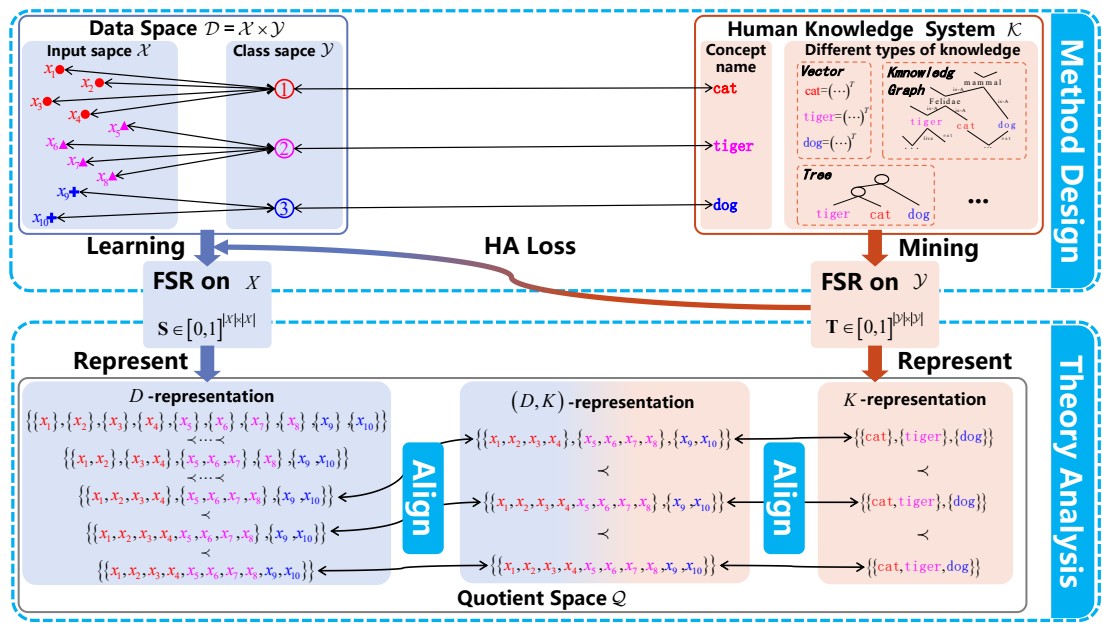

*Figure 1.* The overall framework of HC-HFLM

accurately depict the similarity between $y_i$ and $y_j$. Therefore, $s_{ij}$ should be equal to the similarity between concepts measured by class knowledge, i.e., $s_{ij} = t_{y_i y_j}$.

(b) In the real case, it is difficult that the sample perfectly represent the corresponding concept. For example, it is almost impossible to find a cat in the real world that perfectly represents the concept of "cat". The $s_{ij}$ cannot represent the similarity between concepts $y_i$ and $y_j$ perfectly. Therefore, $s_{ij}$ should be less than the similarity between concepts measured by class knowledge, i.e., $s_{ij} < t_{y_i y_j} = u_{sort(t_{y_i y_j};U)}$.

(c) In the real case, although the similarity $s_{ij}$ cannot perfectly represent the similarity between concepts $y_i$ and $y_j$, it hardly violates the rank. For example, it is difficult to find a cat, a tiger, and a dog in the real world such that cat and dog are more similar than cat and tiger. Therefore, $s_{ij}$ should not be too different from the similarity measured by class knowledge, i.e., $s_{ij} > u_{sort(t_{y_i y_j};U)-1} = \max\limits_{w \in U,\, w < t_{y_i y_j}} w$.

(d) Combining (a)-(c), when $s_{ij}$ falls into the interval $\left( u_{sort(t_{y_i y_j};U)-1}, u_{sort(t_{y_i y_j};U)} \right)$, the loss is 0. Otherwise, the nonzero loss should be generated.

***Training.*** Based on the HA loss, the class FSR $\mathbf{T}$ can guide the learning process. Formally, we have

$$\min_{\Theta} \sum_{i,j=1, i \neq j}^{n} \mathcal{L}_{\text{HA}} \left( f\left( (x_i, x_j)\,;\Theta \right), y_i, y_j \right) + \gamma \mathcal{R}(\Theta), \quad (9)$$

where $f\left( (\cdot, \cdot)\,;\Theta \right)$ is a FSR on $\mathcal{X}$ with learnable parameters

$\Theta$. $\mathcal{R}$ is regularization term. $\gamma > 0$ is trade-off parameter. Formula (9) can be solved efficiently by stochastic gradient descent method (Kingma & Ba, 2015).

Let $\Theta^*$ be the local optimal solution of formula (9), then $f\left( (\cdot, \cdot)\,;\Theta^* \right)$ forms a FSR on $\mathcal{X}$. If HA loss of $f\left( (\cdot, \cdot)\,;\Theta^* \right)$ is 0, then $f\left( (\cdot, \cdot)\,;\Theta^* \right)$ can effectively approximate the fuzzy equivalence relation (FER). In theory, we have

**Theorem 3.3** (see **Appendix** C.1.1 for proof)**.** *For a* $(\mathcal{X}, \mathcal{Y}, f_c)$-*classification problem,* ***Given***
*(a1) fuzziness parameters* $0 < \alpha < \beta < 1$ *(see formula* (2)*).*
*(a2) training data set* $D = \{(x_i, y_i) | x_i \in \mathcal{X}, y_i = f_c(x_i) \in \mathcal{Y}\}_{i=1}^{n}$.
*(a3)* $\alpha$-*FSR* $\mathbf{T}$ *on* $\mathcal{Y}$ *constructed from class knowledge* $K$ *(see **Definition** 3.1 and formula* (6)*).*
*(a4) the local optimal solution* $\Theta^*$ *of formula* (9) *and the FSR* $f\left( (\cdot, \cdot)\,;\Theta^* \right)$ *on* $\mathcal{X}$.
***Let***
*(b1)* $X = \{x_i \mid (x_i, y_i) \in D\}$ *be the set of training samples.*
*(b2)* $\mathbf{S} \in [0,1]^{n \times n}$, $\forall i, j = 1, 2, \cdots, n$, $s_{ij} = f\left( (x_i, x_j)\,;\Theta^* \right)$ *be the predicted FSR on* $X$.
*(b3)* $\hat{\mathbf{S}} = traclo(\mathbf{S})$ *be transitive-closure of* $\mathbf{S}$ *(see **Definition** A.19) and also be a FER on* $X$ *(see **Theorem** 4.1).*
*(b4)* $\mathcal{L}_1 = \sum_{i=1}^{n} \sum_{j=1, j \neq i}^{n} \mathcal{L}_{\text{HA}} (s_{ij}, y_i, y_j)$ *be the corresponding HA loss of* $\mathbf{S}$ *(see formula* (8)*).*
*(b5)* $\mathcal{L}_2 = \sum_{i=1}^{n} \sum_{j=1, j \neq i}^{n} \mathcal{L}_{\text{FP}} (s_{ij}, y_i, y_j)$ *be the corresponding fuzzy permissible loss of* $\mathbf{S}$ *(see formula* (2)*).*
*(b6)* $\mathcal{L}_3 = \sum_{i=1}^{n} \sum_{j=1, j \neq i}^{n} \mathcal{L}_{\text{FP}} (\hat{s}_{ij}, y_i, y_j)$ *be the corre-*

*sponding fuzzy permissible loss of $\hat{\mathbf{S}}$ (see formula* (2)*).*

*If $\mathcal{L}_1 = 0$, then (1) $\mathcal{L}_2 = 0$, (2) $\mathcal{L}_3 = 0$.*

**Theorem** 3.3 indicates that the HA loss is more stringent than the FP loss and inherits the ability of PF loss in approximating FER.

### *(3) Concept Representation and Predicting*

By replacing $f\left((\cdot, \cdot)\,; \Theta^\circ\right)$ in formulas (3) and (4) with $f\left((\cdot, \cdot)\,; \Theta^*\right)$, concept representation and predicting will be completed.

Both $f\left((\cdot, \cdot)\,; \Theta^\circ\right)$ learned by FLM and $f\left((\cdot, \cdot)\,; \Theta^*\right)$ learned by HC-HFLM are FSRs on $\mathcal{X}$. Guided by the knowledge-infused concept representation, the $f\left((\cdot, \cdot)\,; \Theta^*\right)$ integrates the class knowledge $K$, thus the cognition of concepts is profounder. Therefore, $f\left((\cdot, \cdot)\,; \Theta^*\right)$ can obtain better concept representation and predicting result.

**Appendix** B.4 gives the algorithm description of the proposed HC-HFLM.

## 4. Theory Analysis

This section demonstrates how the knowledge-infused conceptual representation guides the learning process.

### 4.1. FSR-based Quotient Space Theory

Literature (Zhang & Zhang, 2014) introduces the FER-based quotient space theory. In practice, it is difficult to obtain FER directly. Hence, we leverage FSR as an alternative for modeling data and knowledge. Consequently, we need to develop the FSR-based quotient space theory.

Literature (Bandler & Kohout, 1988) introduces the transitive-closure of fuzzy binary relation. This paper only focuses on the transitive-closure of FSR, which satisfies the following special properties.

**Theorem 4.1** (see **Appendix** C.2.1 for proof). *Given a finite set $A$ and a FSR $S$ on $A$. The following propositions are true. (1) $\forall k = 0, 1, ..., S^{2^k}$ is a FSR on $A$. (2) $\forall k = 0, 1, ..., S^{2^k} \subseteq S^{2^{k+1}}$. (3) Sequence $t(k) = S^{2^k}$, $k = 0, 1, \cdots$, converges to $traclo(S)$. And $traclo(S)$ is the transitive-closure of $S$. (4) $traclo(S)$ is a FER on $A$.*

Based on **Theorem** 4.1 and the definitions of cut relation and partition (see **Definition** A.17 and A.8), we have the following theorem.

**Theorem 4.2** (see **Appendix** C.2.2 for proof). *Given a finite set $A$ and a FSR on $A$. $\forall \lambda \in [0, 1]$, $traclo(S)^{[\lambda]} = traclo(S^{[\lambda]})$.*

Based on **Theorem** 4.2, we can construct a quotient space from a FSR as follows.

**Definition 4.3.** Given a finite set $A$ and a FSR on $A$. $\mathcal{Q}(A, S) = \left\{ A/traclo\left(S^{[\lambda]}\right) \middle| \lambda \in [0, 1] \right\}$ is called the quotient space derived by $S$.

The meaning of **Definition** 4.3 is as follows.

(a) $\forall \lambda \in [0, 1]$, $S^{[\lambda]}$ is the $\lambda$-cut relation of $S$ (see **Definition** A.17). According to **Theorem** A.18, $S^{[\lambda]}$ is a SR on $A$ (see **Definition** A.4).

(b) $traclo\left(S^{[\lambda]}\right)$ is the transitive-closure of $S^{[\lambda]}$ (see **Definition** A.19). According to **Theorem** 4.1, $traclo\left(S^{[\lambda]}\right)$ is an ER on $A$ (see **Definition** A.5).

(c) $A/traclo\left(S^{[\lambda]}\right)$ is the partition on $A$ derived by $traclo\left(S^{[\lambda]}\right)$ (see **Definition** A.6, A.7, and A.8).

(d) Combining (a)-(c), $\mathcal{Q}(A, S)$ is a set of partitions on $A$ that are derived by $S$.

(e) Given a finite set $A$ and a FER $T$ on $A$. $\forall \lambda \in [0, 1]$, according to **Theorem** A.18, $T^{[\lambda]}$ is an ER on $A$. According to **Definition** A.19, $traclo\left(T^{[\lambda]}\right) = T^{[\lambda]}$. Then $\mathcal{Q}(A, T) = \left\{ A/T^{[\lambda]} \middle| \lambda \in [0, 1] \right\}$ degenerate into the hierarchical structure given in literature (Zhang & Zhang, 2014).

The main properties of **Definition** 4.3 are as follows.

**Theorem 4.4** (see **Appendix** C.2.3 for proof). *Given a finite set $A$ and a FSR on $A$. Let $V = \{S((a, b))| \ a, b \in A\}$, then $\mathcal{Q}(A, S) = \left\{ A/traclo\left(S^{[\lambda]}\right) \middle| \lambda \in V \right\}$.*

**Theorem 4.5** (see **Appendix** C.2.4 for proof). *Given a finite set $A$ and a FSR on $A$, then $\mathcal{Q}(A, S) = \mathcal{Q}(A, traclo(S))$.*

**Theorem** 4.5 establishes the connection between the quotient space derived by FSR and FER. Due to the existence and uniqueness of transitive-closures of FSR (Bandler & Kohout, 1988), the theories developed based on FER in literature (Zhang & Zhang, 2014) can be easily extended to the quotient space derived by FSR.

**Theorem 4.6** (see **Appendix** C.2.5 for proof). *Given a finite set $A$, a FSR $S$ on $A$, and $\preceq$, $\prec$ described in **Definition** A.10. The following propositions are true. (1) $(\mathcal{Q}(A, S), \preceq)$ is a partially ordered set. (2) $\forall \mathbb{P}, \mathbb{Q} \in \mathcal{Q}(A, S)$, $\mathbb{P} \neq \mathbb{Q}$, $\mathbb{P}$ and $\mathbb{Q}$ are comparable under the sense of $\prec$.*

**Theorem 4.7** (see **Appendix** C.2.6 for proof). *Given a finite set $A$ and a FSR $S$ on $A$. Let $U = \left\{ S((a, b)) \middle| a, \ b \in A \right\}$. The following propositions are true. (1) $\forall \mathbb{P}, \mathbb{Q} \in \mathcal{Q}(A, S)$, If $\mathbb{P} \neq \mathbb{Q}$, then $|\mathbb{P}| \neq |\mathbb{Q}|$. (2) $|\mathcal{Q}(A, S)| \leq |A|$. (3) If $|U| > |\mathcal{Q}(A, S)|$, then $\exists V \subset U$, $|V| = |\mathcal{Q}(A, S)|$, such that $\mathcal{Q}(A, S) = \left\{ A/traclo(S^{[\lambda]}) \middle| \lambda \in V \right\}$. (4) If $S$ satisfies transitivity, i.e., $S$ is FER on $A$, then $|U| = |\mathcal{Q}(A, S)|$.*

**Theorem 4.8** (see **Appendix** C.2.7 for proof). *Given a finite set $A$ and two FSRs $R$, $S$ on $A$. If the ranking of elements in $R$ is consistent with that in $S$, i.e., $\forall a, b, c, d \in A$,*

$$\begin{cases} R((a, b)) = R((c, d)) \text{ iff } S((a, b)) = S((c, d)) \\ R((a, b)) > R((c, d)) \text{ iff } S((a, b)) > S((c, d)) \end{cases}, \quad (10)$$

*then* $\mathcal{Q}(A, R) = \mathcal{Q}(A, S)$.

Given the above properties, we can construct a hierarchical structure in the quotient space derived by FSR as follows.

**Definition 4.9.** Given a finite set $A$, a FSR $S$ on $A$, and the $\prec$ described in **Definition** A.10. According to **Theorem** 4.6, $\forall \mathbb{P}, \mathbb{Q} \in \mathcal{Q}(A, S)$, $\mathbb{P} \neq \mathbb{Q}$, $\mathbb{P}$ and $\mathbb{Q}$ are comparable under the sense of $\prec$. According to **Theorem** 4.7, $|\mathcal{Q}(A, S)| \leq |A|$. Without loss of generality, let $\mathcal{Q}(A, S) = \{\mathbb{P}_i\}_{i=1}^{|\mathcal{Q}(A,S)|}$ and $\forall i = 1, 2, \cdots, |\mathcal{Q}(A, S)| - 1$, $\mathbb{P}_i \prec \mathbb{P}_{i+1}$. The $(\mathcal{Q}(A, S), \prec) = [\mathbb{P}_1 \prec \mathbb{P}_2 \prec \cdots \prec \mathbb{P}_{|\mathcal{Q}(A,S)|}]$ is called hierarchical structure derived by $S$.

### 4.2. Represent Data and Knowledge in Quotient Space

***Data representation.*** Guided by class FSR $\mathbf{T}$, formula (9) learns a FSR $f((\cdot, \cdot); \Theta^*)$ from $D$. Let $\mathbf{S} \in [0,1]^{n \times n}$, $\forall i, j = 1, 2, \cdots, n$, $s_{ij} = f((x_i, x_j); \Theta^*)$ be the predicting FSR on training samples set $X$. According to **Definition** 4.3 and 4.9, the hierarchical structure derived by $\mathbf{S}$ is

$$(\mathcal{Q}(X, \mathbf{S}), \prec) = [\mathbb{Q}_1 \prec \mathbb{Q}_2 \prec \cdots \prec \mathbb{Q}_{|\mathcal{Q}(X,\mathbf{S})|}], \quad (11)$$

denoted as $D$-representation.

***Knowledge representation.*** In formula (6), class knowledge $K$ is modeled as a FSR $\mathbf{T}$ on $\mathcal{Y}$. According to **Definition** 4.3 and 4.9, the hierarchical structure derived by $\mathbf{T}$ is

$$(\mathcal{Q}(\mathcal{Y}, \mathbf{T}), \prec) = [\mathbb{O}_1 \prec \mathbb{O}_2 \prec \cdots \prec \mathbb{O}_{|\mathcal{Q}(\mathcal{Y},\mathbf{T})|}], \quad (12)$$

denoted as $K$-representation. It has clear semantics and high interpretability. See **Appendix** B.5 for detailed analysis.

### 4.3. Align Data and Knowledge in Quotient Space

In **Section** 4.2, $K$-representation $(\mathcal{Q}(\mathcal{Y}, \mathbf{T}), \prec)$ and $D$-representation $(\mathcal{Q}(X, \mathbf{S}), \prec)$ are defined on $\mathcal{Y}$ and $X$, respectively. Due to the gap between $\mathcal{Y}$ and $X$, it is not feasible to align these two representations directly. Therefore, we need to construct a bridge between them.

**Definition 4.10.** For a $(\mathcal{X}, \mathcal{Y}, f_c)$-classification problem, given a set of samples $\phi \subset X \subseteq \mathcal{X}$ and a fuzzy binary relation $F$ on $\mathcal{Y}$. $\forall x_i, x_j \in X$, let $G((x_i, x_j)) = F((f_c(x_i), f_c(x_j)))$. $G$ is called fuzzy binary relation on $X$ that is spanned by $F$ and $f_c$, denoted as $G = span(F, f_c, X)$.

For a $(\mathcal{X}, \mathcal{Y}, f_c)$-classification problem, given the set of training samples $X$ and a class FSR $\mathbf{T}$ obtained by formula (6). According to **Definition** 4.10, 4.3, and 4.9, the hierarchical structure derived by $span(\mathbf{T}, f_c, X)$ is

$$\begin{aligned} (\mathcal{Q}(X, span(\mathbf{T}, f_c, X)), \prec) = \\ [\mathbb{P}_1 \prec \mathbb{P}_2 \prec \cdots \prec \mathbb{P}_{|\mathcal{Q}(X, span(\mathbf{T}, f_c, X))|}], \end{aligned} \quad (13)$$

denoted as $(D, K)$-representation.

***First***, we align $K$-representation and $(D, K)$-representation by the following theorem.

**Theorem 4.11** (see **Appendix** C.2.8 for proof). *Given the symbols defined in* **Theorem** *3.3, let* $(\mathcal{Q}(\mathcal{Y}, \mathbf{T}), \prec)$ *and* $(\mathcal{Q}(X, span(\mathbf{T}, f_c, X)), \prec)$ *be the $K$-representation given by formula* (12) *and the $(D, K)$-representation given by formula* (13), *respectively. If* $\forall y \in \mathcal{Y}$, $\exists x \in X$, *such that* $f_c(x) = y$, *then the alignment between* $(\mathcal{Q}(\mathcal{Y}, \mathbf{T}), \prec)$ *and* $(\mathcal{Q}(X, span(\mathbf{T}, f_c, X)), \prec)$ *is as follows.*

*(1)* $|\mathcal{Q}(\mathcal{Y}, \mathbf{T})| = |\mathcal{Q}(X, span(\mathbf{T}, f_c, X))|$.

*(2)* $\forall i = 1, 2, \cdots, |\mathcal{Q}(\mathcal{Y}, \mathbf{T})|$, *the alignment relation between* $\mathbb{O}_i$ *and* $\mathbb{P}_i$ *is (a)* $\forall O \in \mathbb{O}_i$, $\{x \mid x \in X, f_c(x) \in O\} \in \mathbb{P}_i$, *(b)* $\forall P \in \mathbb{P}_i$, $\{f_c(x) \mid x \in P\} \in \mathbb{O}_i$, *(c)* $|\mathbb{O}_i| = |\mathbb{P}_i|$.

***Then***, we align $D$-representation and $(D, K)$-representation by the following theorem.

**Theorem 4.12** (see **Appendix** C.2.9 for proof). *Given the symbols defined in* **Theorem** *3.3, let* $(\mathcal{Q}(X, \mathbf{S}), \prec)$ *and* $(\mathcal{Q}(X, span(\mathbf{T}, f_c, X)), \prec)$ *be the $D$-representation given by formula* (11) *and the $(D, K)$-representation given by formula* (13), *respectively. If the HA loss of $\mathbf{S}$ is* $0$, *then* $\forall i = 1, 2, \cdots, |\mathcal{Q}(X, span(\mathbf{T}, f_c, X))|$, $\exists j \in \{1, 2, \cdots, |\mathcal{Q}(X, \mathbf{S})|\}$, *such that* $\mathbb{P}_i = \mathbb{Q}_j$.

***At last***, we align $D$-representation and $K$-representation by the following corollary.

**Corollary 4.13.** *Given the symbols defined in* **Theorem** *3.3, let* $(\mathcal{Q}(\mathcal{Y}, \mathbf{T}), \prec)$ *and* $(\mathcal{Q}(X, \mathbf{S}), \prec)$ *be the $K$-presentation given by formula* (12) *and the $D$-presentation given by formula* (11), *respectively. If* $\forall y \in \mathcal{Y}$, $\exists x \in X$, *such that* $f_c(x) = y$, *and the HA loss of $\mathbf{S}$ is* $0$, *then*

$$\begin{aligned} &(\mathcal{Q}(\mathcal{Y}, \mathbf{T}), \prec) \\ &\xleftarrow[\text{See } \mathbf{\textit{Theorem }} 4.11]{Align} (\mathcal{Q}(X, span(\mathbf{T}, f_c, X)), \prec) \\ &\xleftarrow[\text{See } \mathbf{\textit{Theorem }} 4.12]{Align} (\mathcal{Q}(X, \mathbf{S}), \prec). \end{aligned} \quad (14)$$

**Corollary** 4.13 shows that minimizing HA loss can align the hierarchical structures of data and class knowledge. This means that the class knowledge contained in human knowledge system is effectively integrated into the learned sample FSR $f((\cdot, \cdot); \Theta^*)$. Thus HC-HFLM can utilize human knowledge system to enhance its understanding of concepts.

## 5. Experiments

### 5.1. Interpretability Analysis

We show the working mechanism of the HC-HFLM on the data set MNIST (LeCun et al., 1998). The class space

*Table 1.* Class knowledge and its representation on handwritten digit classification task

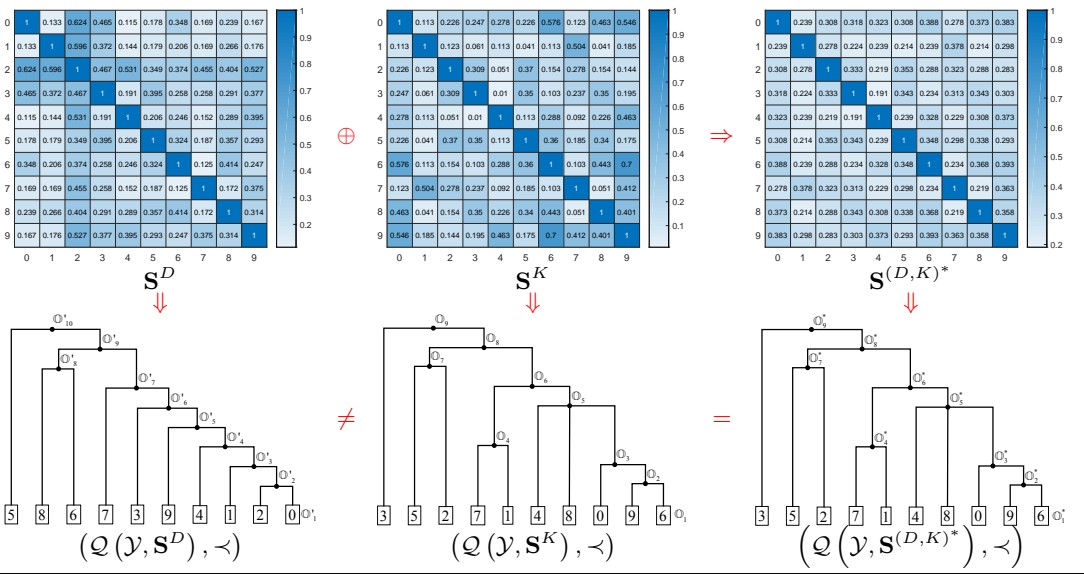

consist of 10 digit characters. For the sake of display, 10 samples are randomly selected for each class to form a training data set. Ten volunteers are recruited to construct a class FSR $\mathbf{S}^K$. Then the calibrated class FSR $\mathbf{S}^{(D,K)^*}$ is obtained by formula (26). **Appendix** D.1 gives more experiment details. We analyze the experimental results from the following three aspects.

### (1) Class Knowledge Representation

Table 1 shows the process of class knowledge representation. We have the following observations.

(a) $\mathbf{S}^K$ is the class FSR constructed by 10 volunteers. $\left(\mathcal{Q}\left(\mathcal{Y},\mathbf{S}^K\right),\prec\right)$ is the hierarchical structure derived by $\mathbf{S}^K$. It can be observed that each class forms an equivalence class in the finest partition $\mathbb{O}_1$. In $\mathbb{O}_1$, the similarity between classes "6" and "9" is the biggest. They are merged to get the second partition $\mathbb{O}_2$. And so on. Finally all the classes are merged into one equivalence class, resulting in the coarsest partition $\mathbb{O}_9$.

(b) $\mathbf{S}^D$ is the estimated class FSR by the pre-trained $f\left((\cdot,\cdot);\Theta^\circ\right)$ (see formula (25)). Comparing $\mathbf{S}^D$ and $\mathbf{S}^K$, it can be observed that the scale of $\mathbf{S}^D$ obtained from data and $\mathbf{S}^K$ obtained from knowledge is quite different. In addition, the ranking of elements is also different.

(c) $\left(\mathcal{Q}\left(\mathcal{Y},\mathbf{S}^D\right),\prec\right)$ is the hierarchical structure derived by $\mathbf{S}^D$. Comparing $\left(\mathcal{Q}\left(\mathcal{Y},\mathbf{S}^D\right),\prec\right)$ and $\left(\mathcal{Q}\left(\mathcal{Y},\mathbf{S}^D\right),\prec\right)$, it can be observed that the difference between $\mathbf{S}^D$ and $\mathbf{S}^K$ directly determines the difference between the hierarchical structure derived by them. Therefore, the hierarchical structure can capture the difference of FSR.

(d) $\mathbf{S}^{(D,K)^*}$ is the optimal solution of formula (26)). $\left(\mathcal{Q}\left(\mathcal{Y},\mathbf{S}^{(D,K)^*}\right),\prec\right)$ is the hierarchical structure derived by $\mathbf{S}^{(D,K)^*}$. Comparing $\mathbf{S}^K$ and $\mathbf{S}^{(D,K)^*}$, the ranking of elements in them are the same. Comparing $\left(\mathcal{Q}\left(\mathcal{Y},\mathbf{S}^K\right),\prec\right)$ and $\left(\mathcal{Q}\left(\mathcal{Y},\mathbf{S}^{(D,K)^*}\right),\prec\right)$, these two hierarchical structures are also the same (see **Theorem** 4.8).

### (2) Data Representation

The left part of Figure 2 gives the hierarchical structure $\left(\mathcal{Q}\left(X,\mathbf{S}\right),\prec\right)$ derived by $\mathbf{S}$. $\mathbf{S}\in[0,1]^{100\times100}$ is the predicting FSR by $f\left((\cdot,\cdot);\Theta^*\right)$. $\left(\mathcal{Q}\left(X,\mathbf{S}\right),\prec\right)$ contains 100 partitions, which is equal to the training samples (see **Theorem** 4.7). In the finest partition $\mathbb{Q}_1$, each sample forms an equivalence class. In the coarsest partition $\mathbb{Q}_{100}$, all samples form an equivalence class. $\mathbb{Q}_{91}$ contains 10 equivalence classes. And each equivalence class is just the set of training samples of a class.

### (3) Alignment between Data and Class Knowledge

In Figure 2, the left part and the right part are the hierarchical structure of data and class knowledge in quotient space, respectively; i.e., $\left(\mathcal{Q}\left(X,\mathbf{S}\right),\prec\right)$ and $\left(\mathcal{Q}\left(\mathcal{Y},\mathbf{S}^{(D,K)^*}\right),\prec\right)$. The alignment between them is as follows.

(a) In the partition $\mathbb{O}_1$, each equivalence class corresponds to a class. In the partition $\mathbb{Q}_{91}$, each equivalence class corresponds to a set of training samples belonging to a class. The two correspond one by one, see the black dotted line.

(b) Based on the alignment between $\mathbb{Q}_{91}$ and $\mathbb{O}_1$, the left and right equivalence classes are merged in the same way, forming coarser partitions $\mathbb{Q}_{92}$ and $\mathbb{O}_2$, respectively. Therefore, $\mathbb{Q}_{92}$ and $\mathbb{O}_2$ are aligned. Analogously, $\mathbb{Q}_{93}$ and $\mathbb{O}_3$,

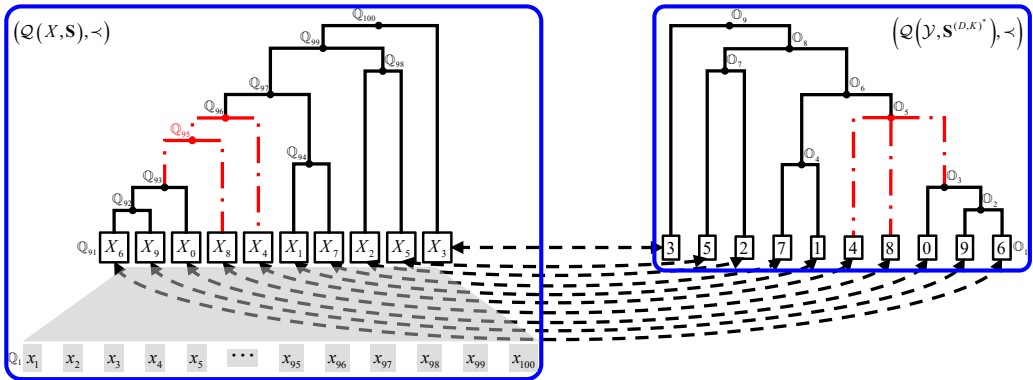

*Figure 2.* The alignment between data and class knowledge on handwritten digit classification task

*Table 2.* The basic information of data sets

| TASK | DATA | | | | CLASS KNOWLEDGE | |
|------|------|------|------|------|------|------|
| | DATA SET | SAMPLE | FEATURE | CLASS | CLASS DESCRIPTION VECTOR | $Tree(\mathcal{Z}, E)$ DERIVED FROM WORDNET |
| COARSE GRAINED CLASSIFICATION | APY | 15399 | 2048 | 32 | 64-D DOMAIN EXPERT | 18 LAYERS, 97 NODES |
| | IMAGENET1K | 1331167 | 2048 | 1000 | 768-D LLMs | 18 LAYERS, 1814 NODES |
| MEDIUM GRAINED CLASSIFICATION | AWA1 | 30475 | 2048 | 50 | 85-D DOMAIN EXPERT | 17 LAYERS, 111 NODES |
| | AWA2 | 37322 | 2048 | 50 | 85-D DOMAIN EXPERT | |
| FINE GRAINED CLASSIFICATION | FLO | 8189 | 2048 | 102 | 1024-D LLMs | N/A |
| | CUB | 11788 | 2048 | 200 | 312-D DOMAIN EXPERT | N/A |

$\mathbb{Q}_{94}$ and $\mathbb{O}_4$ are also aligned.

(c) Based on the alignment between $\mathbb{Q}_{94}$ and $\mathbb{O}_4$, the left equivalence classes are merged twice to get a coarser partition $\mathbb{Q}_{96}$. The right equivalence classes are merged once to get coarser partition $\mathbb{O}_5$, see the red dot line. Therefore, $\mathbb{Q}_{96}$ and $\mathbb{O}_5$ is aligned.

(d) Based on the alignment between the partitions $\mathbb{Q}_{96}$ and $\mathbb{O}_5$, the equivalence classes on the left and right continue to merge in the same way. Therefore, $\forall k = 1, 2, \cdots, 4$, $\mathbb{Q}_{96+k}$ and $\mathbb{O}_{5+k}$ are aligned.

(e) According to (a)-(d), the substructure of the left hierarchical structure $[\mathbb{Q}_{91} \prec \cdots \prec \mathbb{Q}_{94} \prec \mathbb{Q}_{96} \prec \cdots \prec \mathbb{Q}_{100}]$ is the common data-knowledge hierarchical structure $(\mathcal{Q}(X, span(\mathbf{T}, f_c, X)), \prec)$ in formula (13), **Theorem** 4.11, and **Theorem** 4.12.

In summary, this experiment demonstrates that the HC-HFLM models the class knowledge and the traning data as hierarchical structures in quotient space and further aligns these two hierarchical structures.

### 5.2. Generalization Analysis

***Data sets.*** This section verifies the generalization performance of the HC-HFLM on 6 public data sets. Two types of knowledge, i.e., class description vector (CK$_1$) and Word-Net (Miller, 1995) (CK$_2$), are integrated into the HC-HFLM, denoted as CK$_1$-HFLM and CK$_2$-HFLM, respectively. Table 2 gives the basic information of data sets, where NA denotes that the CK$_2$ cannot be obtained for the data set.

***Setting.*** 6 different classifiers are choose for comparison. Among them, k-nearest neighbors (KNN), decision tree (DT), support vector machine (SVM) and naive Bayes (NB) are 4 classical classifiers. Cross entropy classifier (CEC) and FLM are based on deep neural network. **Appendix** D.2 gives more experiment details.

***Results and analysis.*** Table 3 records the test accuracy of 8 methods on 6 data sets, where $--$ denotes that the result can not been obtained within 7 days. We have the following observations.

(a) Compared with the classifiers KNN, DT, SVM and NB,

*Table 3.* The test accuracy of different methods (%)

|          | APY   | IMAGENET1K | AWA1  | AWA2  | FLO   | CUB   |
|----------|-------|------------|-------|-------|-------|-------|
| KNN      | 85.35 | 75.49      | 86.61 | 89.83 | 83.39 | 47.30 |
| DT       | 63.13 | −−         | 63.47 | 70.09 | 42.65 | 21.64 |
| SVM      | 84.54 | −−         | 84.26 | 89.06 | 86.56 | 43.89 |
| NB       | 76.13 | 73.66      | 84.67 | 87.68 | 85.53 | 60.27 |
| CEC      | 89.08 | 75.62      | 88.48 | 91.67 | 93.58 | 61.49 |
| FLM      | 89.42 | 76.02      | 89.95 | 92.79 | 94.05 | 66.19 |
| **$CK_1$-HFLM** | **90.23** | 76.16 | **91.10** | **93.59** | **95.06** | **68.78** |
| **$CK_2$-HFLM** | 90.21 | **76.20** | 90.87 | 93.35 | N/A | N/A |

CEC achieves better performance due to the powerful expression ability of the deep neural networks. When the backbone network is the same, the performance of FLM is superior to CEC. This is because FLM designs more reasonable optimization objective according to the concept cognition principles.

(b) Compared with FLM, $CK_1$-HFLM and $CK_2$-HFLM achieve better performance, which shows that the concept cognition of the proposed method has been improved under the guidance of class knowledge. Especially on the fine-grained classification data set CUB, significant performance improvement is obtained by integrating class knowledge.

(c) On APY, AWA1, and AWA2, the performance of $CK_1$-HFLM integrated with class description vector is superior to $CK_2$-HFLM integrated with WordNet knowledge. On these three data sets, class description vector are constructed by domain experts. Compared with WordNet, the class description vector is more suitable for these tasks.

(d) On ImageNet1K, the performance of $CK_2$-HFLM integrated with WordNet knowledge is superior to the $CK_1$-HFLM integrated with class description vector. The class space is subset (Russakovsky et al., 2015) of words in Word-Net. Compared with the class description vector obtained by the pre-trained language model, the class knowledge contained in WordNet is more suitable for this task.

In summary, the HC-HFLM obtains significant performance improvement benefiting from the guidance of the class knowledge. Meanwhile, the more class knowledge meets the requirements of the task, the greater the performance gain is obtained.

## 6. Conclusion

Inspired by human cognition, we propose a hierarchical fuzzy learning machine. The method constructs the knowledge-infused concept representation by mining class fuzzy similarity relation from human knowledge system and employs a novel hierarchical alignment loss to guide the learning process. Meanwhile, we introduce the fuzzy similarity relation-based quotient space theory. Based on

this theory, we prove that the proposed method can align the hierarchical structures of data and knowledge, which guarantees that the proposed method can leverage human knowledge system to enhance concept cognition. Experimental results validate the proposed method's ability to improve both interpretability and generalization. With its capability to learn from human knowledge system, the proposed method holds great promise for open-world learning tasks, such as zero-shot learning and continual learning.

## Acknowledgements

This work is supported by the National Natural Science Foundation of China (Nos. 62376141, U21A20473, 62376142) and the UK-China Joint Laboratory of Security and Control on Smart Energy 202104041101020.

## Impact Statement

This paper presents work whose goal is to advance the field of Machine Learning. There are many potential societal consequences of our work, none which we feel must be specifically highlighted here.

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

## A. Preliminaries

### A.1. Binary Relation

To unify the expression, we adopt the form of mapping to define binary relation and fuzzy binary relation.

**Definition A.1.** (Bandler & Kohout, 1988) Given a set $A$, the mapping $R : A \times A \to \{0, 1\}$ is called a binary relation on $A$.

**Definition A.2.** (Bandler & Kohout, 1988) Given a set $A$ and a binary relation $R$ on $A$, the three properties of $R$ are defined as:
(1) reflexivity: $\forall a \in A$, $R((a, a)) = 1$,
(2) symmetry: $\forall a, b \in A$, $R((a, b)) = R((b, a))$,
(3) transitivity: $\forall a, b, c \in A$, if $R((a, b)) = 1$, $R((b, c)) = 1$, then $R((a, c)) = 1$.
Obviously, transitivity can be equivalently defined as $\forall a, b \in A$, $R((a, b)) \geq \max_{c \in A} \min[R((a, c)), R((c, b))]$.

**Definition A.3.** (Bandler & Kohout, 1988) Given a set $A$. Let $R$ and $S$ be two binary relations on $A$.
(1) $R \subseteq S$ iff $\forall a, b \in A$, $R((a, b)) \leq S((a, b))$,
(2) $R \subset S$ iff $R \subseteq S$ and $\exists a, b \in A$, such that $R((a, b)) < S((a, b))$.

**Definition A.4.** (Bandler & Kohout, 1988) Given a set $A$ and a binary relation $R$ on $A$, if $R$ satisfies reflexivity and symmetry, then $R$ is called a similarity relation (SR) on $A$.

**Definition A.5.** (Bandler & Kohout, 1988) Given a set $A$ and a SR $R$ on $A$, if $R$ satisfies transitivity, then $R$ is called an equivalence relation (ER) on $A$.

**Definition A.6.** (Rosen, 2007) Given twos set $A$ and $\mathbb{P} = \{P_1, P_2, \cdots, P_m\}$, $m \geq 1$, if $\mathbb{P}$ satisfies the following properties:
(1) $\forall P_i \in \mathbb{P}$, $P_i \neq \phi$,
(2) $\bigcup_{P_i \in \mathbb{P}} = A$,
(3) $\forall P_i, P_j \in \mathbb{P}$, if $i \neq j$, then $P_i \cap P_j = \phi$,
then $P$ is called a partition on $A$. Partition is a special type of quotient set.

**Definition A.7.** (Rosen, 2007) Given a set $A$ and an ER $R$ on $A$. $\forall a \in A$, $[A]_R = \{b|\ b \in A, R((a, b)) = 1\}$ is called an equivalence class of $a$ derived by $R$.

**Definition A.8.** Given a set $A$ and an ER $R$ on $A$. Obviously, $A/R = \{[a]_R|\ a \in A\}$ is a partition on $A$. And $A/R$ is called a partition on $A$ derived by $R$.

**Remark A.9.** According to **Definition** A.6, A.7 and A.8, given a set $A$, the ERs on $A$ and the partitions on $A$ can form a one-to-one mapping. Based on it, in **Proposition** 1 of literature (Cui & Liang, 2022), we model the classification problem as the equivalent relation problem on the input space, which establishes the equivalence between the classification problem and binary classification problem.

**Definition A.10.** Given a set $A$ and two partitions $\mathbb{P}$, $\mathbb{Q}$ on $A$, if $\forall P \in \mathbb{P}$, $\exists Q \in \mathbb{Q}$, such that $P \subseteq Q$, we call that $\mathbb{Q}$ is

coarser than $\mathbb{P}$, denoted as $\mathbb{P} \preceq \mathbb{Q}$. Equivalently, we call that $\mathbb{P}$ is finer than $\mathbb{Q}$, denoted as $\mathbb{Q} \succeq \mathbb{P}$. Additionally, if $\mathbb{P} \preceq \mathbb{Q}$ and $\exists P \in \mathbb{P}, \exists Q \in \mathbb{Q}$, such that $P \subset Q$, we call that $\mathbb{Q}$ is coarser than $\mathbb{P}$ strictly, denoted as $\mathbb{P} \prec \mathbb{Q}$, Equivalently, we call that $\mathbb{P}$ is finer than $\mathbb{Q}$ strictly, denoted as $\mathbb{Q} \succ \mathbb{P}$.

**Remark A.11.** According to **Remark** A.9 and **Definition** A.10, the following propositions are true.
(1) Given a set $A$ and two ERs $R$, $S$ on $A$, if $R \subseteq S$, then $A/R \preceq A/S$.
(2) Given a set $A$ and two partitions $\mathbb{P}$, $\mathbb{Q}$ on $A$. Let $R$ and $S$ be the ERs corresponding to $\mathbb{P}$ and $\mathbb{Q}$ respectively. If $\mathbb{P} \preceq \mathbb{Q}$, then $R \subseteq S$.

**Remark A.12.** Given a set $A$. Let $\mathcal{P}$ be the set of partitions on $A$. Obviously, the $\preceq$ described in **Definition** A.10 satisfies the following properties.
(1) Reflexivity: $\forall \mathbb{P} \in \mathcal{P}, \mathbb{P} \preceq \mathbb{P}$.
(2) Antisymmetry: $\forall \mathbb{P}, \mathbb{Q} \in \mathcal{P}$, if $\mathbb{P} \preceq \mathbb{Q}$ and $\mathbb{Q} \preceq \mathbb{P}$, then $\mathbb{P} = \mathbb{Q}$.
(3) Transitivity: $\forall \mathbb{P}, \mathbb{Q}, \mathbb{R} \in \mathcal{P}$, if $\mathbb{P} \preceq \mathbb{Q}$ and $\mathbb{Q} \preceq \mathbb{R}$, then $\mathbb{P} \preceq \mathbb{R}$.
In summary, $\preceq$ is a order relation on $\mathcal{P}$.

**Lemma A.13.** *Given a set finite $A$. Let $\mathbb{P}$ and $\mathbb{Q}$ be two partitions on $A$. If $\mathbb{P} \neq \mathbb{Q}$ and $\mathbb{P} \preceq \mathbb{Q}$, then $|\mathbb{P}| \neq |\mathbb{Q}|$.*

*Proof.* Reduction to absurdity.

Assume that $|\mathbb{P}| = |\mathbb{Q}| = m$.

Because $|\mathbb{P}| = |\mathbb{Q}| = m$, without losing generality, let $\mathbb{P} = \{P_1, P_2, \cdots, P_m\}$, $\mathbb{Q} = \{Q_1, Q_2, \cdots, Q_m\}$. Meanwhile, because $\mathbb{P} \preceq \mathbb{Q}$, without losing generality, let

$$\forall i = 1, 2, \cdots, m, \, P_i \subseteq Q_i, \tag{15a}$$

$$\forall i = 1, 2, \cdots, m, |P_i| \leq |Q_i|, \text{ i.e. } |Q_i| - |P_i| \geq 0. \tag{15b}$$

Because $\mathbb{P}$ and $\mathbb{Q}$ are both partition on $A$, according to **Definition** A.6,

$$|A| = \sum_{i=1}^{m} |P_i|, \tag{16a}$$

$$|A| = \sum_{i=1}^{m} |Q_i|. \tag{16b}$$

Subtracting the left and right sides of formula (16a) and (16b) respectively,

$$0 = \sum_{i=1}^{m} (|Q_i| - |P_i|). \tag{17}$$

According to formula (17) and (15b),

$$\forall i = 1, 2, \cdots, m, \, |Q_i| - |P_i| = 0, \text{ i.e., } |Q_i| = |P_i|. \tag{18}$$

According to formula (18) and (15a),

$$\forall i = 1, 2, \cdots, m, \, P_i = Q_i. \tag{19}$$

According to formula (19) and (15), $\mathbb{P} = \mathbb{Q}$. It contradicts with $\mathbb{P} \neq \mathbb{Q}$. Therefore, the original proposition is true. $\square$

In this paper, we use $\mathbb{O}$ to represent the partition on class space $\mathcal{Y}$, use $\mathbb{P}$ and $\mathbb{Q}$ to represent the partitions on the set of training samples $X$. Based on them, we use $O \in \mathbb{O}, P \in \mathbb{P}$, and $Q \in \mathbb{Q}$ to represent the equivalence classes in partition $\mathbb{O}, \mathbb{P}$, and $\mathbb{Q}$, respectively.

## A.2. Fuzzy Binary Relation

**Definition A.14.** (Bandler & Kohout, 1988) Given a set $A$, the mapping $F : A \times A \to [0, 1]$ is called a fuzzy binary relation on $A$.

Comparing **Definition** A.1 and A.14, the binary relation is a special type of fuzzy binary relation.

**Definition A.15.** (Bandler & Kohout, 1988) Given a set $A$ and a fuzzy binary relation $F$ on $A$. If $F$ satisfies
(1) reflexivity: $\forall a \in A, R((a, a)) = 1$,
(2) symmetry: $\forall a, b \in A, R((a, b)) = R((b, a))$,
$F$ is called a Fuzzy Similarity Relation (FSR) on $A$.

**Definition A.16.** (Bandler & Kohout, 1988) Given a set $A$ and a FSR $S$ on $A$. If $F$ satisfies transitivity: $\forall a, c \in A$, $F((a, c)) \geq \max_{b \in A} min[F((a, b)), F((b, c))]$, $F$ is called a fuzzy equivalence relation (FER) on $A$.

The following definition intuitively establishes the connection between binary relation and fuzzy binary relation.

**Definition A.17.** (Zhang & Zhang, 2014) Given a set $A$ and a fuzzy binary relation $F$ on $A$, $\forall \lambda \in [0, 1]$, the binary relation, $\forall a, b \in A$,

$$F^{[\lambda]}((a, b)) = \mathbb{I}[F((a, b)) \geq \lambda] = \begin{cases} 1, & F((a, b)) \geq \lambda \\ 0, & \text{otherwise} \end{cases},$$

is called the $\lambda$-cut binary relation, or $\lambda$-cut relation for short.

Based on **Definition** A.17, the following theorem further establishes the connection between FSR (FER) and SR (ER).

**Theorem A.18.** *(The proof is intuitive and omitted.) Given a set $A$ and a fuzzy binary relation $F$ on $A$, $\forall \lambda \in [0, 1]$, let $F^{[\lambda]}$ be the $\lambda$-cut relation of $F$.*
*(1) If $F$ is a FSR on $A$, then $F^{[\lambda]}$ is a SR on $A$.*
*(2) If $F$ is a FER on $A$, then $F^{[\lambda]}$ is an ER on $A$.*

**Definition A.19.** (Bandler & Kohout, 1988) Given a set $A$ and a fuzzy binary relation $F$ on $A$. If the fuzzy binary relation $G$ on $A$ satisfies
(1) transitivity,
(2) $F \subseteq G$,
(3) $\forall$ fuzzy binary relation $H$ on $A$, $F \subseteq H$, if $H$ satisfies transitivity, then $G \subseteq H$,
$G$ is called the transitive-closure of $F$, denoted as $G = traclo(F)$. Obviously, if $F$ satisfies transitivity, then $traclo(F) = F$.

Literature (Bandler & Kohout, 1988) gives general definitions of $P$-closure. The **Definition** A.19 is a special case among them. Moreover, this paper only focuses on the transitive-closure of FSR.

**Definition A.20.** (Bandler & Kohout, 1988) Given a set $A$. Let $R$ and $S$ be two fuzzy binary relation on $A$. The product fuzzy binary relation of $R$ and $S$ is defined as $T = R \otimes S$, $\forall a, b \in A$, $T((a,b)) = \max_{c \in A} \min[R((a,c)), S((c,b))]$.

Given $\otimes$, we can define the power of fuzzy binary relation as follows.

**Definition A.21.** Given a set $A$ and a fuzzy binary relation $F$ on $A$. $F^1 = F$, $F^2 = F^1 \otimes F$, $F^3 = F^2 \otimes F$, $\cdots$, and so on.

**Definition A.22.** Given a finite set $A$ and a fuzzy binary relation $F$ on $A$. Without losing generality, we can sequentially number the elements in $A$ as $a_1, a_2, \cdots, a_{|A|}$. Based on it, $F$ can be equivalently represented by a matrix, i.e.,

$$\mathbf{F} \in [0,1]^{|A| \times |A|}, \forall i, j = 1, 2, \cdots, |A|, f_{ij} = F((a_i, a_j)).$$

Definitions A.19-A.22 are also applicable to binary relation. In this paper, we use matrices $\mathbf{S}^K$, $\mathbf{S}^D$, $\mathbf{S}^{(D,K)}$, $\mathbf{T} \in [0,1]^{|\mathcal{Y}| \times |\mathcal{Y}|}$ to represent the FSRs on class space $\mathcal{Y}$, use matrix $\mathbf{S} \in [0,1]^{n \times n}$ to represent the FSR on the set of training samples $X$, and use $f((\cdot, \cdot); \Theta)$ to represent the parameterized FSR on input space $\mathcal{X}$.

### A.3. Fuzzy Quotient Space

Based on the preliminaries given in **Appendix** A.1 and A.2, literature (Zhang & Zhang, 2014) systematically develops the FER-based fuzzy quotient space theory. We list the contents used in this paper as follows.

**Theorem A.23.** *(Zhang & Zhang, 2014) Given a set $A$, the following assertions are equivalent.*
*(1) Given a FER on $A$ (see **Definition** A.16).*
*(2) Given a a normalized isosceles distance $d$ on $A$, where*

- *normalized: $\forall a, b \in A$, $d(a,b) \in [0,1]$,*

- *isosceles: $\forall a, b, c \in A$, $d(a,b)$, $d(b,c)$, and $d(a,c)$ form a an isosceles triangle and and its congruent legs are the longest side.*

*(3) Given a hierarchical structure on $A$, i.e.,*

- *a set of partitions on $A$, and*

- *these partitions form an ordered queue under the sense of $\prec$ described in **Definition** A.10.*

According to **Definition** A.16, FER needs to satisfy transitivity. In practice, it is relatively difficult to directly obtain

fuzzy binary relation that satisfy transitivity. Therefore, we develop the FSR-based quotient space theory (see **Section** 4.1). Based on it, the class knowledge and data are represented as hierarchical structures in the quotient space (see **Section** 4.2). And by designing a appropriate loss, we align these two hierarchical structures (see **Section** 4.3). As a result, the understanding of class concepts contained in the class knowledge is integrated into the model.

## B. Details of Method

### B.1. Knowledge Modeling

This section discusses different types of class knowledge and the corresponding methods for calculating class FSR.

---

**Algorithm 1** Generate Class Tree Structure from WordNet, dDenoted as $WordNet2ClassTree(\cdots)$

---

**Input:** (1) Knowledge base WordNet.
    (2) Class space $\mathcal{Y}$. $\forall k \in \mathcal{Y}$, let $w_k$ be the concept name of class $k$. $\forall y_1, y_2 \in \mathcal{Y}$, there is no hypernym relation between $w_{y_1}$ and $w_{y_2}$.
**Output:** A tree structure $Tree(\mathcal{Z}, E)$ on $\mathcal{Y}$, where $\mathcal{Z}$ is the set of nodes, $E$ is the set of edges, and $\mathcal{Y} \subset \mathcal{Z}$ is set of leaves.
 1: Initialize the set of words corresponding to class space $\mathcal{Y}$: $W \leftarrow \{w_y | y \in \mathcal{Y}\}$;

**// (1) Mark the nodes corresponding to the words in $W$.**
 2: Initialize queue: $Q \leftarrow \{WordNet.entity.n.01\}$; // $entity.n.01$ is the common hypernym of other words.
 3: **while** $Q$ is not empty **do**
 4:    $node \leftarrow Q.out()$;         // Out of the queue
 5:    **if** $node.word$ belongs to $W$ **then**
 6:       Mark $node$ and its hypernym nodes as valid, recursively;
 7:    **else**
 8:       Mark $node$ as invalid;
 9:    **end if**
10:    $node\_set \leftarrow node.hyponyms()$; // The set of hyponyms of $node$
11:    **for** $node \in node\_set$ **do**
12:       $Q.In(node)$;         // Entering the queue
13:    **end for**
14: **end while**

**// (2) Delete the invalid nodes.**
15: Initialize queue: $Q \leftarrow \{WordNet.entity.n.01\}$;
16: **while** $Q$ is not empty **do**
17:    $node \leftarrow Q.out()$;
18:    **if** $node$ is valid **then**
19:       $node\_set \leftarrow node.hyponyms()$;
20:       **for** $node \in node\_set$ **do**
21:          $Q.In(node)$;
22:       **end for**
23:    **else**
24:       Delete the edge between $node$ and its hypernym node;
25:    **end if**
26: **end while**
27: Return the tree with $WordNet.entity.n.01$ as the root;

---

(1) $K = \{\mathbf{a}_i \mid \mathbf{a}_i \in \mathbb{R}^{d_K}, i \in \mathcal{Y}\}$, where $\mathbf{a}_i$ is a description vector of class $i$. It can be designed by domain experts.

**Algorithm 2** Generate Hierarchical Structure from Class Tree Structure, denoted as $ClassTree2HieStru(\cdots)$

**Input:** (1) Class space $\mathcal{Y}$.
  (2) Class tree structure $Tree\,(\mathcal{Z}, E)$, where $\mathcal{Z}$ is the set of nodes, $E$ is the set of edges, and $\mathcal{Y} \subset \mathcal{Z}$ is set of leaves.
**Output:** A set of partitions on $\mathcal{Y}$.
1: Initialize the set of partitions $H \leftarrow \phi$;
2: Initialize the partition $\mathbb{P} \leftarrow \phi$;
3: Initialize the set of nodes $N \Leftarrow \{Tree\,(\mathcal{Z}, E)\,.root\}$;
4: **while** $|\mathbb{P}| < |\mathcal{Y}|$ **do**
5:   $\mathbb{P}' \leftarrow \phi, N' \leftarrow \phi$;
6:   **for** $node \in N$ **do**
7:     $\mathbb{P}' \leftarrow \mathbb{P}' \bigcup \{node.leaf\_set\}$;
8:     $N' \leftarrow N' \bigcup node.child\_set$;
9:   **end for**
10:   $\mathbb{P} \leftarrow \mathbb{P}', N \leftarrow N'$,
11:   $H \leftarrow H \bigcup \{\mathbb{P}\}$; // The duplicate partitions are deleted.
12: **end while**

---

**Algorithm 3** Generate FER from Class Hierarchical Structure, denoted as $ClassHieStru2FER(\cdots)$

**Input:** (1) Class space $\mathcal{Y}$.
  (2) A hierarchical structure $H$ on $\mathcal{Y}$.
  (3) Fuzzy parameter $0 < \alpha < 1.0$.
  (4) A strictly monotonically increasing function $f$ : $\{1, 2, \cdots, |\mathcal{Y}| - 1\} \rightarrow (0, \alpha)$.
**Output:** A $\alpha$-FSR on $\mathcal{Y}$.
1: Initialize the FSR $\mathbf{S} \in \{0\}^{|\mathcal{Y}| \times |\mathcal{Y}|}$;
2: Sort the partitions in $H$ according to "$\prec$" described in **Definition** A.10. Let $\mathbb{P}_1 \prec \cdots \prec \mathbb{P}_{|H|}$ be the sorted result;
3: **for** $i = 1, 2, \cdots, |\mathcal{Y}|$ **do**
4:   $s_{ii} \leftarrow 1$;
5:   **for** $j = i + 1, i + 2, \cdots, |\mathcal{Y}|$ **do**
6:     **for** $k = |H| - 1, |H| - 2, \cdots, 1$ **do**
7:       **if** $\exists A \in \mathbb{P}_k$, such that $i, j \in A$ **then**
8:         $s_{ij} \leftarrow f\,(|\mathbb{P}_k|), s_{ji} \leftarrow f\,(|\mathbb{P}_k|)$;
9:         **break**; //**Theorem** 2.5 in (Zhang & Zhang, 2014).
10:       **end if**
11:     **end for**
12:   **end for**
13: **end for**

It can also be obtained from pre-trained language models. Given $K$, $\forall i, j \in \mathcal{Y}, i \neq j$, the similarity between classes $i$ and $j$ can be calculated as follows[1].

$$sim(i, j; K) = \frac{\alpha}{2} \left( 1 + \frac{\mathbf{a}_i^T \mathbf{a}_j}{\|\mathbf{a}_i\|_2 \times \|\mathbf{a}_j\|_2} \right) \in (0, \alpha). \quad (20)$$

(2) $K$ is a knowledge graph, e.g., WordNet (Miller, 1995). Each class label corresponding to a word in WordNet. Currently, a large number of methods have been proposed to calculate the semantic similarity between two words based on WordNet (Lastra-Díaz & García-Serrano, 2015; AlMousa et al., 2021). Given the WordNet, $\forall i, j \in \mathcal{Y}, i \neq j$, the simi-

larity between classes $i$ and $j$ can be calculated as follows[2].

$$sim(i, j; K) = \frac{\alpha}{z}\text{WordNet}\,(w_i, w_j) \in (0, \alpha), \quad (21)$$

where $w_i$ and $w_j$ are the words corresponding to class $i$ and $j$, respectively. WordNet $(w_i, w_j)$ is the semantic similarity between $w_i$ and $w_j$. And $z$ is the normalization factor.

In addition, WordNet stores the hypernym relation to represent the "Is-A" relation. For example, the hypernym of 'cat' is 'animal', the corresponding to the semantic is that "cat is a type of animal". Based on the hypernym relation, we can obtain the tree structure on class space by **Algorithm** 1.

$$Tree(\mathcal{Z}, E) = WordNet2ClassTree(\text{WordNet}, \mathcal{Y}), \quad (22)$$

where $\mathcal{Z}$ is the set of nodes, $E$ is the set of edges. In this paper, we assume that there is no hypernym relation between classes in $\mathcal{Y}$. Obviously, $\mathcal{Y} \subset \mathcal{Z}$ is the set of leaves. Given a $Tree(\mathcal{Z}, E)$ on $\mathcal{Y}$, we can generate FERs on $\mathcal{Y}$ from $Tree(\mathcal{Z}, E)$, see **Algorithm** 2 and 3.

(3) In many applications, there is a natural hierarchical structure between classes (Silla & Freitas, 2011; Wang et al., 2022). The class knowledge $K$ can be represented as a tree structure $Tree(\mathcal{Z}, E)$ on $\mathcal{Y}$, where $\mathcal{Z}$ is the set of nodes, $E$ is the set of edges. If there is no hypernym relation between classes in $\mathcal{Y}$, then $\mathcal{Y} \subset \mathcal{Z}$ is the set of leaves. By **Algorithm** 2, we can drive a hierarchical structure on $\mathcal{Y}$ from $Tree(\mathcal{Z}, E)$, i.e.,

$$H = ClassTree2HieStru(Tree(\mathcal{Z}, E)). \quad (23)$$

Given the above hierarchical structure $H$ on $\mathcal{Y}$, according to **Theorem** A.23, we can obtain a $\alpha$-FER on $\mathcal{Y}$, i.e.,

$$\mathbf{S} = ClassHieStru2FER(H) \text{ (see **Algorithm** 3). } (24)$$

Let $\mathbf{S}^K = \mathbf{S}$, which is the required $\alpha$-FER on $\mathcal{Y}$.

The following example demonstrates how to use the hypernym relation in WordNet to construct a class FSR.

**Example B.1.** Given the class space $\mathcal{Y} = \{\text{car}, \text{airplane}, \text{tiger}, \text{cat}, \text{dog}\}$ and WordNet 3.0. Figure 3 shows the process of sequentially calling **Algorithm** 1, 2, and 3 to obtain a FSR on $\mathcal{Y}$.

(1) As shown in Figure 3(a), the WordNet tree with "entity. n.01" as the root node has a total of 96308 nodes, of which 74897 are leaf nodes, and the depth is 20.

(2) Input $\mathcal{Y}$ and WordNet into **Algorithm** 1. By deleting nodes that are not related to $\mathcal{Y}$, **Algorithm** 1 obtains a tree structure $Tree(\mathcal{Z}, E)$ on $\mathcal{Y}$ (see Figure 3(b)). Its depth is 15, and there are 31 nodes and 5 leaves. Each leaf corresponds to a class in $\mathcal{Y}$. In $Tree(\mathcal{Z}, E)$, the position of classes can

---

[1] In practice, we can introduce a small constant $\epsilon > 0$ to make the value of $sim(\cdot, \cdot; \cdot)$ fall into the interval $(0, \alpha)$.

[2] Same as 1.

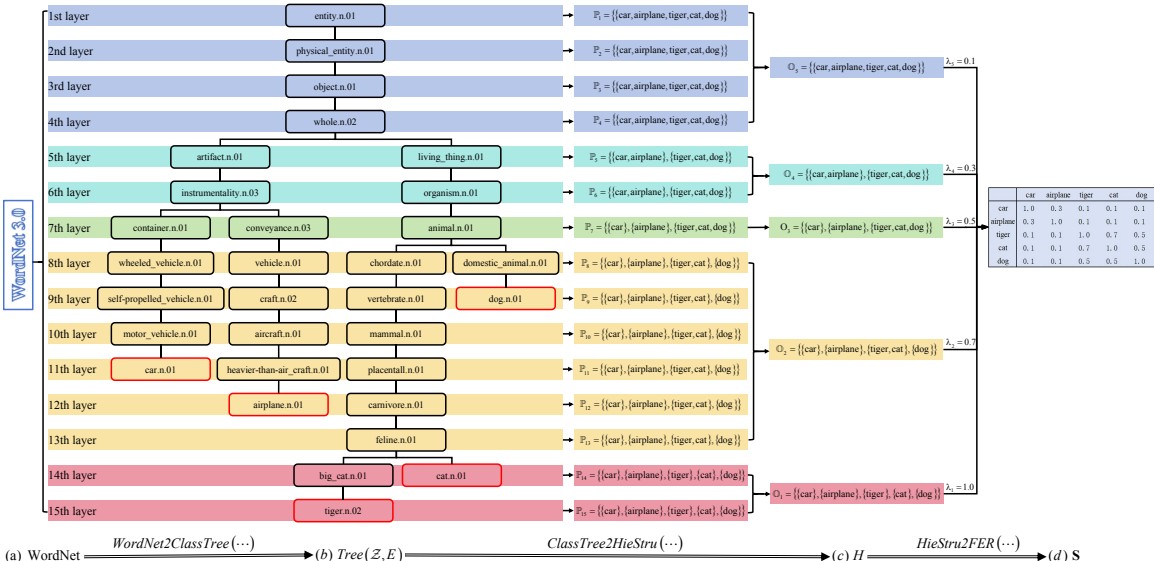

*Figure 3.* The execution processes of **Algorithm** 1, 2, and 3

reflect the semantic similarity between them. For example, the path between "cat" and "tiger" is shorter than the path between "cat" and "dog", which indicates that the semantic similarity between "cat" and "tiger" is bigger.

(3) Next, we input $Tree(\mathcal{Z}, E)$ into **Algorithm** 2. We show intermediate results of **Algorithm** 2 between Figure 3(b) and Figure 3(c). There are 5 different partitions among 15 layers in $Tree(\mathcal{Z}, E)$.

$\mathbb{P}_1 = \cdots = \mathbb{P}_4 = \{\{car, airplane, tiger, cat, dog\}\}$,
$\mathbb{P}_5 = \mathbb{P}_6 = \{\{car, airplane\}, \{tiger, cat, dog\}\}$,
$\mathbb{P}_7 = \{\{car\}, \{airplane\}, \{tiger, cat, dog\}\}$
$\mathbb{P}_8 = \cdots = \mathbb{P}_{13} = \{\{car\}, \{airplane\}, \{tiger, cat\}, \{dog\}\}$,
$\mathbb{P}_{14} = \mathbb{P}_{15} = \{\{car\}, \{airplane\}, \{tiger\}, \{cat\}, \{dog\}\}$.

Obviously, the above 5 partitions form a ordered queue under the sense of $\prec$ (see Figure 3(c)).

(4) We input $H$ into **Algorithm** 3. We show intermediate results of **Algorithm** 3 between Figure 3(c) and Figure 3(d). Given the hierarchical structure $H$ and the set of thresholds $\{\lambda_i\}_{i=1}^5$, by executing lines 3-12 of **Algorithm** 3, we can obtain a FSR $\mathbf{S}$ on $\mathcal{Y}$, as shown in Figure 3(d). Obviously, $\mathbf{S}$ is also a FER on $\mathcal{Y}$.

### B.2. Knowledge Calibration

In practice, the sources of data $D$ and class knowledge $K$ are different, so the scales by which they characterize class similarity are not entirely the same. In addition, there may be certain differences in the class similarity reflected by data and knowledge (Wang et al., 2020). For example, "whales" and "sharks" have significant visual similarities.

In the biology, "whales" and "sharks" belong to two major categories, "mammal" and "fish", respectively. Therefore, it is necessary to calibrate $\mathbf{S}^K$ obtained by formula (5) before using it guide the learning process,

#### B.2.1. OPTIMIZATION MODELING

By solving the formula (1), FLM can learn FSR on input space $\mathcal{X}$ from the training data $D$. Let $\Theta^\circ$ be the (local) optimal solution of formula (1). We can use $f((\cdot, \cdot); \Theta^\circ)$ and training data $D$ to estimate the similarity between classes, i.e., $\mathbf{S}^D \in [0, 1]^{|\mathcal{Y}| \times |\mathcal{Y}|}$, $\forall i, j \in \mathcal{Y}$,

$$s_{ij}^D = \begin{cases} 1, & i = j \\ \frac{\sum_{x_k \in X_i} \sum_{x_l \in X_j} f((x_k, x_l); \Theta^\circ)}{|X_i| \times |X_j|}, & i \neq j \end{cases} . \quad (25)$$

Then, we design the following optimization problem[3]

$$\min_{\mathbf{S}^{(D,K)}} \left\| \mathbf{S}^{(D,K)} - \mathbf{S}^D \right\|_F^2$$
$$\begin{aligned} s.t. \quad & s_{ij}^{(D,K)} = s_{kl}^{(D,K)}, \ \forall i,j,k,l \in \mathcal{Y}, \ s_{ij}^K = s_{kl}^K \\ & s_{ij}^{(D,K)} > s_{kl}^{(D,K)}, \ \forall i,j,k,l \in \mathcal{Y}, \ s_{ij}^K > s_{kl}^K \\ & s_{ii}^{(D,K)} = 1, \ \forall i \in \mathcal{Y} \\ & s_{ij}^{(D,K)} = s_{ji}^{(D,K)}, \ \forall i,j \in \mathcal{Y}, i \neq j \\ & 0 < s_{ij}^{(D,K)} < \alpha, \ \forall i,j \in \mathcal{Y}, i \neq j \end{aligned} . \quad (26)$$

For formula (26), we have the following statements.
(1) The objective function measures the closeness between variable $\mathbf{S}^{(D,K)}$ and $\mathbf{S}^D$.

---

[3]In practice, a small constant $\epsilon > 0$ can be introduced to ensure strict inequality constraints

(2) The first two constraints ensure that the ranking of elements in optimization variable $\mathbf{S}^{(D,K)}$ is consistent with that in $\mathbf{S}^K$, which can maintain the hierarchical structure derived by $\mathbf{S}^K$ (see **Theorem** 4.8 and **Definition** 4.9).
(3) The last three constraints require that optimization variable $\mathbf{S}^{(D,K)}$ is a $\alpha$-FSR on $\mathcal{Y}$ (see **Definition** 3.1).
(4) It can be transformed into a standard convex quadratic programming problem (Boyd & Vandenberghe, 2004).

### B.2.2. OPTIMIZATION SOLVING

In formula (26), the size of the optimization variable and constraint condition are $O\left(|\mathcal{Y}|^2\right)$ and $O\left(|\mathcal{Y}|^4\right)$, respectively. When $|\mathcal{Y}|$ is large, solving it directly faces huge time costs and space costs. Fortunately, we can transform it into the following.

$$
\begin{array}{ll}
\min_{\mathbf{t}\in\mathbb{R}^{|V^K|}} & \mathbf{t}^T\mathbf{Dt} - 2\bar{\mathbf{s}}^T\mathbf{Dt} \\
s.t. & t_i < t_{i+1},\ \forall i,j = 1,2,\cdots,\left|V^K\right|-1 \\
& 0 < t_i < \alpha,\ \forall i = 1,2,\cdots,\left|V^K\right|
\end{array},\ (27)
$$

where

$$
V^K = \left\{ s_{ij}^K \mid i,j\in\mathcal{Y},\ i\neq j \right\}, \tag{28a}
$$

$$
\begin{aligned}
&\forall i = 1,2,\cdots,\left|V^K\right|, \\
&I_i = \left\{ (k,l) \mid k,l\in\mathcal{Y}, sort\left(s_{kl}^K;V^K\right)=i \right\},
\end{aligned} \tag{28b}
$$

$$
\forall i = 1,2,\cdots,\left|V^K\right|,\ \bar{s}_i = \frac{\sum_{(k,l)\in I_i} s_{kl}^D}{|I_i|}, \tag{28c}
$$

$$
\bar{\mathbf{s}} = \left(\bar{s}_1,\bar{s}_2,\cdots,\bar{s}_{|V^K|}\right)^T \in \mathbb{R}^{|V^K|}, \tag{28d}
$$

$$
\mathbf{D} \in \mathbb{R}^{|V^K|\times|V^K|},\forall i,j = 1,2,\cdots,\left|V^K\right|, \tag{28e}
$$

$$
d_{ij} = \begin{cases} |I_i|, & i=j \\ 0, & i\neq j \end{cases}.
$$

According to (28), $\left|V^K\right| < \frac{1}{2}|\mathcal{Y}|^2$. In practice, class knowledge $K$ is usually expressed by higher-order semantic information i.e., $\left|V^K\right| \ll \frac{1}{2}|\mathcal{Y}|^2$. Therefore, formula (27) can be solved more efficiently. Importantly, by solving formula (27), we can obtain the optimal solution of formula (26).

**Theorem B.2** (see **Appendix** C.3.1 for proof). *Let $\mathbf{t}^*$ be the optimal solution of formula (27). Let $\mathbf{S}^{(D,K)^*} \in [0,1]^{|\mathcal{Y}|\times|\mathcal{Y}|}, \forall i,j\in\mathcal{Y}$,*

$$
s_{ij}^{(D,K)^*} = \begin{cases} 1, & i=j \\ t^*_{sort\left(s_{ij}^K;V^K\right)}, & i\neq j \end{cases}. \tag{29}
$$

*The following propositions are true.*
*(1) $\mathbf{S}^{(D,K)^*}$ is a $\alpha$-FSR on $\mathcal{Y}$.*
*(2) $\mathbf{S}^{(D,K)^*}$ is the optimal solution of formula (26).*

### B.3. Knowledge Coarsening and Calibration

In practice, the $\mathbf{S}^K$ obtained by formula (5) may contain a large number of elements with small differences. These

may be caused by insufficient quality of $K$, or by the characteristics of formula (5). It is necessary to pre-processing $\mathbf{S}^K$ appropriately before using it.

**First**, we introduce coarsening function $f_{\mathrm{coa}}: V^K \to [0,1]$ which needs to satisfy the following conditions.

$$
\forall u,v\in V^K, u<v,\ f_{\mathrm{coa}}(u) \leq f_{\mathrm{coa}}(v), \tag{30a}
$$

$$
\exists u,v\in V^K, u<v,\ \text{such that } f_{\mathrm{coa}}(u) = f_{\mathrm{coa}}(v). \tag{30b}
$$

**Second**, we use $f_{\mathrm{coa}}$ to coarsen $\mathbf{S}^K$ as follows.

$$
\mathbf{S}_{\mathrm{coa}}^K \in [0,1]^{|\mathcal{Y}|\times|\mathcal{Y}|}, \forall i,j\in\mathcal{Y}, s_{\mathrm{coa}_{ij}}^K = f_{\mathrm{coa}}\left(s_{ij}^K\right). \tag{31}
$$

**Third**, we replace $\mathbf{S}^K$ in formula (26) with the coarsening result $\mathbf{S}_{\mathrm{coa}}^K$. According to **Theorem** B.2, we can obtain

$$
\begin{array}{ll}
\min_{\mathbf{r}\in\mathbb{R}^{|V^K|}} & \mathbf{r}^T\hat{\mathbf{D}}\mathbf{r} - 2\hat{\mathbf{s}}^T\hat{\mathbf{D}}\mathbf{r} \\
s.t. & r_i < t_{i+1},\ \forall i,j = 1,2,\cdots,\left|V_{\mathrm{coa}}^K\right|-1 \\
& 0 < r_i < \alpha,\ \forall i = 1,2,\cdots,\left|V_{\mathrm{coa}}^K\right|
\end{array},\ (32)
$$

where

$$
V_{\mathrm{coa}}^K = \left\{ f_{\mathrm{coa}}\left(s_{ij}^K\right) \mid i,j\in\mathcal{Y},\ i\neq j \right\}, \tag{33a}
$$

$$
\begin{aligned}
&\forall i = 1,2,\cdots,\left|V_{\mathrm{coa}}^K\right|, \\
&\hat{I}_i = \left\{ (k,l) \mid k,l\in\mathcal{Y}, sort\left(f_{\mathrm{coa}}\left(s_{kl}^K\right);V_{\mathrm{coa}}^K\right)=i \right\},
\end{aligned} \tag{33b}
$$

$$
\forall i = 1,2,\cdots,\left|V_{\mathrm{coa}}^K\right|,\ \hat{s}_i = \frac{\sum_{(p,q)\in\hat{I}_i} s_{pq}^D}{\left|\hat{I}_i\right|}, \tag{33c}
$$

$$
\hat{\mathbf{s}} = \left(\hat{s}_1,\hat{s}_2,\cdots,\hat{s}_{|V_{\mathrm{coa}}^K|}\right)^T \in \mathbb{R}^{|V_{\mathrm{coa}}^K|}, \tag{33d}
$$

$$
\hat{\mathbf{D}} \in \mathbb{R}^{|V_{\mathrm{coa}}^K|\times|V_{\mathrm{coa}}^K|},\forall i,j = 1,2,\cdots,\left|V_{\mathrm{coa}}^K\right|, \tag{33e}
$$

$$
d_{ij} = \begin{cases} \left|\hat{I}_i\right|, & i=j \\ 0, & i\neq j \end{cases}.
$$

Compared with formula (32) and (27), the size of optimization variables and constraints in formula (32) are both $O\left(\left|V_{\mathrm{coa}}^K\right|\right)$. According to formula (30b), (31), and (33a), $\left|V_{\mathrm{coa}}^K\right| < \left|V^K\right|$. In practice, by setting appropriate coarsening functions $f_{\mathrm{coa}}$, we can achieve $\left|V_{\mathrm{coa}}^K\right| \ll \left|V^K\right|$. This can not only reduce the scale of optimization problem, but also make more rational use of class knowledge $K$.

**At last**, let $\mathbf{r}^*$ be the optimal solution of formula (32). Similar to (29), we can obtain a calibrated $\alpha$-FSR on $\mathcal{Y}$. $\mathbf{S}_{\mathrm{coa}}^{(D,K)^*} \in [0,1]^{|\mathcal{Y}|\times|\mathcal{Y}|}, \forall i,j\in\mathcal{Y}$,

$$
s_{\mathrm{coarse}\ ij}^{(D,K)^*} = \begin{cases} 1, & i=j \\ r^*_{sort\left(f_{\mathrm{coa}}\left(s_{ij}^K\right);V_{\mathrm{coa}}^K\right)}, & i\neq j \end{cases}. \tag{34}
$$

$\mathbf{S}_{\mathrm{coa}}^{(D,K)^*}$ is not the optimal solution to the formula (26). However, $\mathbf{S}_{\mathrm{coa}}^{(D,K)^*}$ selectively preserves the more robust

sorting information contained in $\mathbf{S}^K$. In addition, let $\mathbf{S}^K$ be obtained by formula (5). And let $U = \left\{ s_{ij}^K \middle| i, j \in \mathcal{Y} \right\}$. According to **Theorem** 4.4 and 4.7, if $|U| > |\mathcal{Y}|$, then there are at least $|U| - |\mathcal{Y}|$ redundant elements in $U$. Therefore, the coarsening of $\mathbf{S}^{K \to \mathcal{Y}}$ is reasonable.

### B.4. Algorithm Description of HC-HFLM

**Algorithm** 4 shows the workflow of the proposed method.

### B.5. The Interpretability of Knowledge Representation

The following example demonstrates the FSR and hierarchical structure derived by class knowledge.

**Example B.3.** Given a class space $\mathcal{Y} = \{\text{cat}, \text{tiger}, \text{dog}\}$.

(1) Based on the concept names corresponding to the classes, we can collect class knowledge $K$ related to $\mathcal{Y}$ from human knowledge system $\mathcal{Y}$. Assuming that the class FSR constructed from class knowledge $K$ is

$T((\text{cat}, \text{cat})) = T((\text{tiger}, \text{tiger})) = T((\text{dog}, \text{dog})) = 1 >$
$T((\text{cat}, \text{tiger})) = T((\text{tiger}, \text{cat})) = 0.7 >$
$T((\text{cat}, \text{dog})) = T((\text{dog}, \text{cat})) = 0.6 >$
$T((\text{tiger}, \text{dog})) = T((\text{dog}, \text{tiger})) = 0.3.$

The matrix representation of $T$ is

$$\mathbf{T} = \begin{pmatrix} 1 & 0.7 & 0.6 \\ 0.7 & 1 & 0.3 \\ 0.6 & 0.3 & 1 \end{pmatrix}.$$

(2) The view of Euclidean space. Given the class FSR matrix $\mathbf{T}$, each row of $\mathbf{T}$ corresponds to a description vector of a concept. And the feature of every dimension corresponds to a concept, which has clear semantics. Therefore, the FSR matrix $\mathbf{T}$ has high interpretability.

(3) The view of quotient space. Given the class FSR $T$, according to **Definition** 4.3 and 4.9, we can obtain the hierarchical structure derived by $T$, i.e.,

$$(\mathcal{Q}(\mathcal{Y}, T), \prec) = \begin{bmatrix} \mathbb{O}_1 = \{\{\text{cat}\}, \{\text{tiger}\}, \{\text{dog}\}\} \prec \\ \mathbb{O}_2 = \{\{\text{cat}, \text{tiger}\}, \{\text{dog}\}\} \prec \\ \mathbb{O}_3 = \{\{\text{cat}, \text{tiger}, \text{dog}\}\} \end{bmatrix}.$$

For $(\mathcal{Q}(\mathcal{Y}, T), \prec)$, we have the following observations.

- When threshold $\lambda = 1$, we obtain the finest partition $\mathbb{O}_1 = \mathcal{Y}/traclo\left(T^{[1]}\right)$. In $\mathbb{O}_1$, each class forms an equivalent class.

- When threshold $\lambda = 0.7$, "cat" and "tiger" are merged to obtain an equivalent class $\{\text{cat}, \text{tiger}\}$. So we obtain the coarser partition $\mathbb{O}_2$.

- When threshold $\lambda = 0.6$, "cat" and "dog" are merged to obtain an equivalent class $\{\text{cat}, \text{tiger}, \text{dog}\}$. So we obtain the coarsest partition $\mathbb{O}_3$.

- When threshold $\lambda \in [0, 0.6)$, we obtain the coarsest partition $\mathbb{O}_3$.

From the above process, we can seen that the ">" relation of class similarity is transformed into the "$\prec$" relation between partitions in quotient space. And the a set partitions forms a hierarchical structure in quotient space. According to **Theorem** 4.8, the same class similarity ranking can lead to the same hierarchical structure. Therefore, the hierarchical structure can effectively capture the understanding of concepts contained in class knowledge $K$.

In addition, each element in the hierarchical structure is a partition on $\mathcal{Y}$, and each element in the partition is a subset of $\mathcal{Y}$, both of which have clear semantics and high interpretability.

## C. Proofs

### C.1. Proofs of Theorems in Section 3

#### C.1.1. PROOF OF THEOREM 3.3

*Proof.* For proposition (1).

For sake of discussion, we copy formula (7) as follows.

$$V = \{t_{ij} \mid i, j \in \mathcal{Y}\}, U = V \cup \{0, \alpha, \beta\},$$
$$\mathbf{u} = sortvec(U) = (u_1, u_2, \cdots, u_{|U|})^T.$$

Because $\mathbf{T}$ is a $\alpha$-FSR on $\mathcal{Y}$, therefore $\forall i \in \mathcal{Y}, t_{ii} = 1$ and $\forall i, j \in \mathcal{Y}, i \neq j, 0 < t_{ij} = t_{ji} < \alpha$, therefore $0 \notin V$, $\alpha \notin V$, $\beta \notin V$. Based on it, according to **Definition** 3.2,

$$sort(1; U) = |U|, \qquad u_{|U|} = 1, \tag{35a}$$
$$sort(\beta; U) = |U| - 1, \ u_{|U|-1} = \beta, \tag{35b}$$
$$sort(\alpha; U) = |U| - 2, \ u_{|U|-2} = \alpha, \tag{35c}$$
$$sort(0; U) = 1, \qquad u_1 = 0, \tag{35d}$$
$$\forall v \in V, \ 2 \leq sort(v; U) \leq |U|. \tag{35e}$$

According to the definition of $\mathcal{L}_1$, $\mathcal{L}_1 = 0$, iff

$$\begin{cases} \mathcal{L}_{1a} = \sum_{i,j=1, i \neq j, y_i = y_j}^n \mathcal{L}_{\text{HA}}(s_{ij}, y_i, y_j) = 0, \\ \mathcal{L}_{1b} = \sum_{i,j=1, i \neq j, y_i \neq y_j}^n \mathcal{L}_{\text{HA}}(s_{ij}, y_i, y_j) = 0. \end{cases}$$

Because $\mathcal{L}_{1a} = 0$, according to (8), $\forall i, j = 1, 2, \cdots, n, i \neq j, y_i = y_j, \mathcal{L}_{\text{HA}}(s_{ij}, y_i, y_j) = 0$, i.e.,

$$s_{ij} \in \left[ u_{sort(t_{y_i y_i}; U) - 1}, u_{sort(t_{y_i y_i}; U)} \right].$$

Because $\mathbf{T}$ is a FSR on $\mathcal{Y}$, therefore $t_{y_i y_i} = 1$. According to formula (35a) and (35b),

$$\begin{aligned} u_{sort(t_{y_i y_i}; U) - 1} &= u_{sort(1; U) - 1} = u_{|U| - 1} = \beta, \\ u_{sort(t_{y_i y_i}; U)} &= u_{sort(1; U)} = 1, \end{aligned}$$

---

**Algorithm 4** Human Cognition-Inspired Hierarchical Fuzzy Learning Machine, denoted as HC-HFLM

---

**Input: Task.** The $(\mathcal{X}, \mathcal{Y}, f_c)$-classification problem.
    **Data.** The training data $D = \{(x_i, y_i) \mid x_i \in \mathcal{X}, y_i = f_c(x_i) \in \mathcal{Y}\}_{i=1}^{n}$ and the test data $D_{\text{te}} = \{x_j \mid x_j \in \mathcal{X}\}_{j=n+1}^{n+m}$.
    **Knowledge.** The class knowledge $K$ related to the concepts in class space $\mathcal{Y}$.
    **Setting.** The hyper-parameters for controlling the degree of fuzziness $0 < \alpha < \beta < 1$. The network architecture of FLM $f((\cdot, \cdot); \Theta) = g((h(\cdot; \Theta), h(\cdot; \Theta)))$. The regular term $\mathcal{R}(\cdot)$. The trade-off parameter $\gamma$. The number of examples required for each concept $\forall k \in \mathcal{Y}, n_{\text{exe}_k}$. The way to pre-processing $K$, i.e., the value of "case" in formula (6).
**Output:** The predicting result $\{(x_j, \hat{y}_j) \mid x_j \in D_{\text{te}}\}$.

**// (1) Pre-training FLM (see Section 2.3)**
1: Initialize the learnable parameters $\Theta$ randomly;
2: Using stochastic gradient descent method to solve formula (1) and obtain the local optimal solution $\Theta^\circ$;

**// (2) Mining class FSR from class knowledge $K$ (see Section 3)**
3: Initialize class FSR $\mathbf{T} \leftarrow NULL$;
4: Construct class FSR $\mathbf{S}^K$ from class knowledge $K$ by formula (5) and **Appendix B.1**;
5: **if** case = case1 of formula (6) **then**
6:     $\mathbf{T} \leftarrow \mathbf{S}^K$
7: **else**
8:     Compute $\mathbf{S}^D$ from $D$ and $f((\cdot, \cdot); \Theta^\circ)$ by formula (25);
9:     **if** case = case2 of formula (6) **then**
10:         Construct formula (27) from $\mathbf{S}^K$ and $\mathbf{S}^D$ by formula (28);
11:         Solve formula (27) and obtain optimal solution $\mathbf{t}^*$;
12:         Construct $\mathbf{S}^{(D,K)^*}$ from $\mathbf{t}^*$ by formula (29);
13:         $\mathbf{T} \leftarrow \mathbf{S}^{(D,K)^*}$;
14:     **else if** case = case3 of formula (6) **then**
15:         Coarsening $\mathbf{S}^K$ by formula (31) and obtain $\mathbf{S}^K_{\text{coa}}$;
16:         Construct formula (32) from $\mathbf{S}^K_{\text{coa}}$ and $\mathbf{S}^D$ by formula (33);
17:         Solve formula (32) and obtain optimal solution $\mathbf{r}^*$;
18:         Construct $\mathbf{S}^{(D,K)^*}_{\text{coa}}$ from $\mathbf{r}^*$ by formula (34);
19:         $\mathbf{T} \leftarrow \mathbf{S}^{(D,K)^*}_{\text{coa}}$;
20:     **end if**
21: **end if**

**// (3) Learning sample FSR, guided by T (see section 3)**
22: Initialize the learnable parameters $\Theta \leftarrow \Theta^\circ$;
23: Using stochastic gradient descent method to solve formula (9) and obtain local optimal solution $\Theta^*$;

**// (4) Concept representation and prediction (see section 3)**
24: Using $f(\cdot, \cdot); \Theta^*)$ to select exemplar set for each class in $\mathcal{Y}$ by formula (3) and obtain $E_k$, $\forall k \in \mathcal{Y}$;
25: $\forall x_j \in D_{\text{te}}$, using $f(\cdot, \cdot); \Theta^*)$ and $E_k$, $\forall k \in \mathcal{Y}$ to obtain the predicted class label $\hat{y}_j$ by formula (4);

---

Therefore $s_{ij} \in [\beta, 1]$. According to formula (2), $\mathcal{L}_{\text{FP}}(s_{ij}, y_i, y_j) = max[\beta - s_{ij}, 0] = 0$. Therefore

$$\mathcal{L}_{2a} = \sum_{i=1}^{n} \sum_{j=1, j \neq i, y_i = y_j}^{n} \mathcal{L}_{\text{FP}}(s_{ij}, y_i, y_j) = 0.$$

Because $\mathcal{L}_{1b} = 0$, according to formula (8), $\forall i, j = 1, 2, \cdots, n$, $i \neq j$, $y_i \neq y_j$, $\mathcal{L}_{\text{HA}}(s_{ij}, y_i, y_j) = 0$, i.e.,

$$s_{ij} \in \left[u_{sort(t_{y_i y_i}; U)-1}, u_{sort(t_{y_i y_i}; U)}\right].$$

Because $\mathbf{T}$ is a $\alpha$-FSR on $\mathcal{Y}$, therefore $0 < t_{y_i y_j} < \alpha$. According to **Definition 3.2** and formula (35e), we have $2-1 \leq sort(t_{y_i y_i}; U)-1 < sort(t_{y_i y_i}; U) < sort(\alpha; U)$, and $0 = u_1 \leq u_{sort(t_{y_i y_i}; U)-1} < u_{sort(t_{y_i y_i}; U)} < u_{sort(\alpha; U)} = \alpha$, Therefore $s_{ij} \in [0, \alpha)$. According to formula (2), $\mathcal{L}_{\text{FP}}(s_{ij}, y_i, y_j) = max[s_{ij} - \alpha, 0] = 0$. There-fore

$$\mathcal{L}_{2b} = \sum_{i=1}^{n} \sum_{j=1, j \neq i, y_i \neq y_j}^{n} \mathcal{L}_{\text{FP}}(s_{ij}, y_i, y_j) = 0.$$

Therefore, $\mathcal{L}_2 = \mathcal{L}_{2a} + \mathcal{L}_{2b} = 0 + 0 = 0$. So proposition (1) is proven.

For proposition (2).

Because $\mathcal{L}_1 = 1$, according to proposition (1) $\mathcal{L}_2 = 0$. According to **Theorem** 2 in literature (Cui & Liang, 2022), $\mathcal{L}_3 = 0$. So proposition (2) is proven.

In summary, proposition (1) and (2) are proven. $\square$

**C.2. Proofs of Theorems in Section 4**

C.2.1. PROOF OF THEOREM 4.1

*(1) We give some properties of the $\otimes$ operation described in Definition A.20.*

**Lemma C.1.** *Given a set $A$ and a fuzzy binary relation $F$ on $A$. The following propositions are true.*

*(1) If $F$ satisfies reflexivity, then $F^2 = F \otimes F$ satisfies reflexivity.*

*(2) If $F$ satisfies reflexivity, then $F \subseteq F^2 = F \otimes F$.*

*(3) If $F$ satisfies symmetry, then $F^2 = F \otimes F$ satisfies symmetry.*

*(4) $F$ satisfies transitivity iff $F^2 = F \otimes F \subseteq F$.*

*Proof.* For proposition (1).

If $F$ satisfies reflexivity, i.e., $\forall a \in A, F((a,a)) = 1$, then $\forall b \in A$,

$$
\begin{aligned}
&F^2((b,b)) \\
&= \max_{c \in A} min[F((b,c)), F((c,b))] \\
&= max \left\{ \begin{array}{l} \max_{c \in A, c \neq b} min[F((b,c)), F((c,b))], \\ min[F(b,b), F((b,b))] \end{array} \right\}, \\
&= max \left\{ \max_{c \in A, c \neq b} min[F((b,c)), F((c,b))], \ 1 \right\} \\
&= 1
\end{aligned}
$$

i.e., $F^2$ satisfies reflexivity. So proposition (1) is proven.

For proposition (2).

If $F$ satisfies reflexivity, i.e., $\forall c \in A, F((c,c)) = 1$, then $\forall a, b \in A$,

$$
\begin{aligned}
&F^2((a,b)) \\
&= \max_{c \in A} min[F((a,c)), F((c,b))] \\
&= max \left\{ \begin{array}{l} \max_{c \in A, c \neq a} min[F((a,c)), F((c,b))], \\ min[F((a,a)), F((a,b))] \end{array} \right\} \\
&= max \left\{ \begin{array}{l} \max_{c \in A, c \neq a} min[F((a,c)), F((c,b))], \\ min[1, F((a,b))] \end{array} \right\} , \\
&= max \left\{ \max_{c \in A, c \neq a} min[F((a,c)), F((c,b))], \ F((a,b)) \right\} \\
&\geq F((a,b))
\end{aligned}
$$

i.e., $F \subseteq F^2$. So proposition (2) is proven.

For proposition (3).

If $F$ satisfies symmetry, i.e., $\forall a, b \in A, F((a,b)) =$

$F((b,a))$, then $\forall c, d \in A$,

$$
\begin{aligned}
F^2((c,d)) &= \max_{e \in A} min[F((c,e)), F((e,d))] \\
&= \max_{e \in A} min[F((e,d)), F((c,e))] \\
&= \max_{e \in A} min[F((d,e)), F((e,c))] \\
&= F^2((d,c))
\end{aligned}
$$

i.e., $F^2$ satisfies symmetry. So proposition (3) is proven.

For proposition (4).

$F^2 \subseteq F \iff \forall a, b \in A, F((a,b)) \geq F^2((a,b)) = \max_{c \in A} min[F((a,c)), F((c,b))]$, i.e., proposition (4) is proven.

In summary, proposition (1)-(4) are proven. $\qquad \square$

**Lemma C.2.** *Given a set $A$ and two fuzzy binary relation $R, S$ on $A$. If $R \subseteq S$, then $R^2 \subseteq S^2$.*

*Proof.* Because $R \subseteq S$, according to **Definition** A.3, $\forall a, b \in A, R((a,b)) \leq S((a,b))$. Therefore $\forall c, d \in A$,

$$
\begin{aligned}
R^2((c,d)) &= \max_{e \in A} min[R((c,e)), R((e,d))] \\
&\leq \max_{e \in A} min[S((c,e)), S((e,d))] = S^2((c,d))
\end{aligned}
$$

i.e., $R^2 \subseteq S^2$. $\qquad \square$

*(2) Based on these conclusions, we prove Theorem 4.1.*

*Proof.* Using mathematical induction to prove proposition (1).

When $k = 0$, $S^{2^k} = S^1 = S$ is a FSR on $A$.

When $k = 1$, $S^{2^1} = S^2 = S \otimes S$. Because $S$ is a FSR on $A$, therefore $S$ satisfies reflexivity and symmetry. According to proposition (1) and (3) of **Lemma** C.1 $S^2$ satisfies reflexivity and symmetry. Therefore $S^2$ is a FSR on $A$.

When $k = m > 1$, assume that $S^{2^m}$ is a FSR on $A$.

When $k = m + 1$, $S^{2^{m+1}} = S^{2^m} \otimes S^{2^m}$. Because $S^{2^m}$ is a FSR on $A$, therefore $S^{2^m}$ satisfies reflexivity and symmetry, According to proposition (1) and (3) of **Lemma** C.1 $S^{2^{m+1}}$ satisfies reflexivity and symmetry. Therefore $S^{2^{m+1}}$ is a FSR on $A$. So proposition (1) is proven.

For proposition (2).

$\forall k = 0, 1, 2, \cdots, S^{2^{k+1}} = S^{2^k} \otimes S^{2^k}$. Because $S$ is a FSR on $A$, according to proposition (1), $S^{2^k}$ is a FSR on $A$. Therefore $S^{2^k}$ satisfies reflexivity. According to proposition (2) of **Lemma** C.1 $S^{2^k} \subseteq S^{2^k} \otimes S^{2^k} = S^{2^{k+1}}$. So proposition (2) is proven.

For proposition (3).

First, let's prove the convergence. According to proposition(1) $\forall t = 0, 1, 2, \cdots, t(k)$ is a FSR on $A$. According to **Definition** A.14, the sequence $t(k)$ has an upper bound. According to proposition (2), the sequence $t(k)$ is monotonically increasing. Therefore the sequence $t(k)$ is convergent.

Second, let's prove that the sequence $t(k)$ converges to the transitive-closure of $S$. Without losing generality, assume that $t(\hat{k}) = t(\hat{k}+1)$, i.e., $S^{2^{\hat{k}}} = S^{2^{\hat{k}+1}} = S^{2^{\hat{k}}} \otimes S^{2^{\hat{k}}}$.

(a) Because $S^{2^{\hat{k}+1}} = S^{2^{\hat{k}}}$, therefore $S^{2^{\hat{k}+1}} \subseteq S^{2^{\hat{k}}}$. According to proposition (4) of **Lemma** C.1, $S^{2^{\hat{k}}}$ satisfies transitivity.

(b) According to proposition (2), $S \subseteq S^{2^{\hat{k}}}$.

(c) $\forall$ fuzzy binary relation $T$ on $A$, assume that $S \subseteq T$ and $T$ satisfies transitivity. Because $S \subseteq T$, according to **Lemma** C.2

$$S \subseteq T \Longrightarrow S^2 \subseteq T^2 \Longrightarrow S^4 \subseteq T^4 \Longrightarrow \cdots S^{2^{\hat{k}}} \subseteq T^{2^{\hat{k}}}. \quad (36)$$

Meanwhile, because $T$ satisfies transitivity, according to proposition (4) of **Lemma** C.1, $T^2 \subseteq T$. According to **Lemma** C.2,

$$T^2 \subseteq T \Longrightarrow T^4 \subseteq T^2 \Longrightarrow \cdots T^{2^{\hat{k}}} \subseteq T^{2^{\hat{k}-1}}. \quad (37)$$

Combining formula (36) and (37),

$$S^{2^{\hat{k}}} \subseteq T^{2^{\hat{k}}} \subseteq T^{2^{\hat{k}-1}} \subseteq \cdots \subseteq T^2 \subseteq T.$$

(d) Combining (a)-(c), $S^{2^{\hat{k}}}$ satisfies the three propositions in **Definition** A.19. Therefore $S^{2^{\hat{k}}}$ is the transitive-closure of $S$. So proposition (3) is proven.

For proposition (4).

According to proposition (3), $traclo(S)$ can be written in the form of $S^{2^{\hat{k}}}$. And then, according to proposition (1), $traclo(S)$ is a FSR on $A$. Because $traclo(S)$ satisfies transitivity, according to **Definition** A.16, $traclo(S)$ is a FER on $A$. So proposition (4) is proven.

In summary, proposition (1)-(4) are proven. $\qquad \square$

C.2.2. PROOF OF THEOREM 4.2

*(1) We introduce the following lemma.*

**Lemma C.3.** *Given a finite set $A$ and a SR matrix $S$ on $A$. Let $G = (A, S)$ be the unweighted graph, where $A$ is the set nodes and $S$ is the adjacency matrix of the graph. $\forall k \in 1, 2, ..., \forall a, b \in A, S^k((a,b)) = 1 \Longleftrightarrow$ in graph $G$, there is a path $\langle a, \cdots, b \rangle$ and the length of $\langle a, \cdots, b \rangle$ is $k$.*

*Proof.* Using mathematical induction to prove.

When $k = 1$, $S^k = S^1 = S$. $\forall a, b \in A$, $S((a,b)) = 1 \Longleftrightarrow$ in graph $G$, there is a edge $(a, b) \Longleftrightarrow$

in graph $G$, there is a path $\langle a, b \rangle$ and the length of path $\langle a, b \rangle$ is 1.

When $k = 2$, $S^2 = S \otimes S$. $\forall a, b \in A$, $S^2((a,b)) = 1 \Longleftrightarrow \max_{c \in A} min[S((a,c)), S((c,b))] = 1 \Longleftrightarrow \underset{c \in A}{\vee} \mathbb{I}[min[S((a,c)), S((c,b))] = 1] = 1 \Longleftrightarrow \underset{c \in A}{\vee}(\mathbb{I}[S((a,c)) = 1] \wedge \mathbb{I}[S((c,b)) = 1]) = 1 \Longleftrightarrow$ in graph $G$, there are two edges $(a,c)$ and $(c,b) \Longleftrightarrow$ in graph $G$, there is a path $\langle a, c, b \rangle$ and the length of path $\langle a, c, b \rangle$ is 2.

When $k = m > 2$, assume that $\forall a, b \in A$, $S^m((a,b)) = 1 \Longleftrightarrow$ in graph $G$, there is a path $\langle a, \cdots, b \rangle$ and the length of path $\langle a, \cdots, b \rangle$ is $m$.

When $k = m + 1$, $S^{m+1} = S^m \otimes S$. $\forall a, b \in A$, $S^{m+1}((a,b)) = 1 \Longleftrightarrow \max_{c \in A} min[S^m((a,c)), S((c,b))] = 1 \Longleftrightarrow \underset{c \in A}{\vee} \mathbb{I}[min[S^m((a,c)), S((c,b))] = 1] = 1 \Longleftrightarrow \underset{c \in A}{\vee}(\mathbb{I}[S^m((a,c)) = 1] \wedge \mathbb{I}[S((c,b)) = 1]) = 1 \Longleftrightarrow$ in graph $G$, there is a path $\langle a, \cdots, c \rangle$ and the length of $\langle a, \cdots, c \rangle$ is $m$ and in graph $G$, there is a edge $(c, b) \Longleftrightarrow$ in graph $G$, there is a path $\langle a, \cdots, b \rangle$ and the length of path $\langle a, \cdots, b \rangle$ is $m + 1$.

In summary, the lemma is proven. $\qquad \square$

**Remark C.4.** Comparing **Definition** A.1 and A.14, it can be seen that binary relation is a special type of fuzzy binary relation. Therefor, **Definition** A.19, A.20, and A.21 are applicable to binary relation. Based on it, **Lemma** C.1 and C.2 are applicable to binary relation. Therefor, **Theorem** 4.1 is applicable to binary relation. According to **Theorem** 4.1, given a finite set $A$ and a similarity relation (SR) $S$ on $A$, the the transitive-closure of $S$ can also be written as $S^{2^k}$.

*(2) Based on the above lemma, we prove Theorem 4.2.*

*Proof.* According to **Remark** C.4, $\exists k \in \{0, 1, 2, \cdots\}$, such that $traclo(S) = S^{2^k}$. Because $S$ is a FSR on $A$, according to **Theorem** A.18, $\forall \lambda \in [0,1]$, $S^{[\lambda]}$ is a SR on $A$. According to **Remark** C.4, $\exists l \in \{0, 1, 2, \cdots\}$, such that $traclo(S^{[\lambda]}) = S^{[\lambda]^{2^l}}$. Let $m = max[2^k, 2^l]$. According to **Theorem** 4.1 $traclo(S) = S^m$ and $traclo(S^{[\lambda]}) = S^{[\lambda]^m}$. Therefore, to prove

$$\forall \lambda \in [0,1], \ tracol(S)^{[\lambda]} = tracol(S^{[\lambda]}),$$

we need to prove

$$\forall \lambda \in [0,1], \ S^{m^{[\lambda]}} = S^{[\lambda]^m},$$

we need to prove

$$\forall \lambda \in [0,1], \ \forall a, b \in A, \ S^{m^{[\lambda]}}((a,b)) = S^{[\lambda]^m}((a,b)).$$

According to **Definition** A.17 $\forall \lambda \in [0,1]$, $\forall a,b \in A$, $S^{m^{[\lambda]}}((a,b)) \in \{0,1\}$, $S^{[\lambda]^m}((a,b)) \in \{0,1\}$, Therefore, we need to prove

$$
\begin{aligned}
&\forall \lambda \in [0,1],\ \forall a,b \in A, \\
&S^{m^{[\lambda]}}((a,b)) = 1 \Longleftrightarrow S^{[\lambda]^m}((a,b)) = 1
\end{aligned} \tag{38}
$$

Next, we prove the formula (38).

Given the unweighted $G = \left(A, S^{[\lambda]}\right)$. $\forall a,b \in A$, according to **Lemma** 2 of literature (Cui & Liang, 2022), $S^{m^{[\lambda]}}((a,b)) = \mathbb{I}[S^m((a,b)) \geq \lambda] = 1 \Longleftrightarrow S^m((a,b)) \geq \lambda \Longleftrightarrow$ in graph $G$, there is a path $\langle a,\cdots,b \rangle$ and the length of path $\langle a,\cdots,b \rangle$ is $m$. Meanwhile, according to **Lemma** C.3, in graph $G$ there is a path $\langle a,\cdots,b \rangle$ and the length of path $\langle a,\cdots,b \rangle$ is $m \Longleftrightarrow S^{[\lambda]^m}((a,b)) = 1$. That is to say, the formula (38) is proven. So the theorem is proven. $\square$

### C.2.3. PROOF OF THEOREM 4.4

*Proof.* According to **Definition** A.14 $\forall a,b \in A$, $S((a,b)) \in [0,1]$. Therefore $V = \{S((a,b))| a,b \in A\} \subseteq [0,1]$. And then, according to **Definition** 4.3, $\mathcal{Q}(A,S) = \left\{ A/traclo\left(S^{[\lambda]}\right)\middle| \lambda \in V \right\} \bigcup \{A/traclo\left(S^{[\lambda]}\right)|\lambda \in [0,1] -V\}$. Therefore, to prove

$$
\mathcal{Q}(A,S) = \left\{ A/traclo\left(S^{[\lambda]}\right)\middle| \lambda \in V \right\},
$$

we need to prove

$$
\begin{aligned}
&\left\{ A/traclo\left(S^{[\lambda]}\right)\middle| \lambda \in [0,1] - V \right\} \subseteq \\
&\left\{ A/traclo\left(S^{[\lambda]}\right)\middle| \lambda \in V \right\},
\end{aligned}
$$

we need to prove $\forall \lambda \in [0,1], \lambda \notin V, \exists \omega \in V$, such that

$$
A/traclo\left(S^{[\lambda]}\right) = A/traclo\left(S^{[\omega]}\right).
$$

According to **Definition** A.8, we need to prove $\forall \lambda \in [0,1], \lambda \notin V, \exists \omega \in V$, such that

$$
traclo\left(S^{[\lambda]}\right) = traclo\left(S^{[\omega]}\right).
$$

According to **Definition** A.19, we need to prove

$$
\forall \lambda \in [0,1], \lambda \notin V,\ \exists \omega \in V,\ \text{such that } S^{[\lambda]} = S^{[\omega]}.
$$

According to **Definition** A.17, we need to prove $\forall \lambda \in [0,1], \lambda \notin V, \exists \omega \in V$, such that $\forall a,b \in A$,

$$
\begin{aligned}
S^{[\lambda]}((a,b)) &= \mathbb{I}(S((a,b)) \geq \lambda) = \mathbb{I}(S((a,b)) \geq \omega) \\
&= S^{[\omega]}((a,b)).
\end{aligned} \tag{39}
$$

$\forall \lambda \in [0,1], \lambda \notin V$, Let $\omega = \min\limits_{v \in V,\ v > \lambda} v$. Obviously, $\lambda < \omega$ and $\omega \in V$.

Next we prove formula (39) in two cases. $\forall a,b \in A$,

(a) If $\mathbb{I}(S(a,b) \geq \lambda) = 1$, i.e., $S(a,b) \geq \lambda$. Because $\lambda \notin V$ and $S((a,b)) \in V$, Therefore $S(a,b) > \lambda$, Therefore

$$
S((a,b)) \geq \min\limits_{v \in V,\ v > \lambda} v = \omega,\ \text{i.e., } \mathbb{I}(S(a,b) \geq \omega) = 1.
$$

(b) If $\mathbb{I}(S(a,b) \geq \lambda) = 0$, i.e., $S((a,b)) < \lambda$. Because $\lambda < \omega$, therefore $S((a,b)) < \lambda < \omega$, i.e., $\mathbb{I}(S(a,b) \geq \omega) = 0$.

Combining (a) and (b), the formula (39) is true. So the theorem is proven. $\square$

### C.2.4. PROOF OF THEOREM 4.5

*Proof.* According to **Definition** 4.3,

$$
\begin{aligned}
\mathcal{Q}(A,S) &= \left\{ A/traclo(S^{[\lambda]})\middle| \lambda \in [0,1] \right\}, \\
\mathcal{Q}(A, traclo(S)) &= \left\{ A/traclo\left(traclo(S)^{[\lambda]}\right)\middle| \lambda \in [0,1] \right\} \\
&= \left\{ A/traclo(S)^{[\lambda]}\middle| \lambda \in [0,1] \right\}.
\end{aligned}
$$

Therefore, to prove $\mathcal{Q}(A,S) = \mathcal{Q}(A, traclo(S))$, we need to prove $\forall \lambda \in [0,1]$, $A/traclo(S^{[\lambda]}) = A/traclo(S)^{[\lambda]}$. According to **Definition** A.8, we need to prove $\forall \lambda \in [0,1]$, $traclo(S^{[\lambda]}) = traclo(S)^{[\lambda]}$, According to **Theorem** 4.2, the above formula is true. So the theorem is proven. $\square$

### C.2.5. PROOF OF THEOREM 4.6

*Proof.* For proposition (1).

According to **Definition** 4.3, $\mathcal{Q}(A,S)$ is a set of partitions on $A$. Meanwhile, according to **Remark** A.12 $\preceq$ is a order relation on the partitions on $A$. Therefore $(\mathcal{Q}(A,S), \preceq)$ is a partial ordered set. So proposition (1) is proven.

For proposition (2).

Let $T = tracol(S)$. According to **Theorem** 4.5 $\mathcal{Q}(A,S) = \mathcal{Q}(A,T) = \left\{ A/T^{[\lambda]}\middle| \lambda \in [0,1] \right\}$. According to **Theorem** 4.1, $T$ is a FER on $A$. According to **Theorem** A.18, $\forall \lambda \in [0,1]$, $T^{[\lambda]}$ is an ER on $A$. $\forall 0 \leq \lambda_1 < \lambda_2 \leq 1$, obviously, $T^{[\lambda_2]} \subset T^{[\lambda_1]}$. According to **Remark** A.11 $A/T^{[\lambda_2]} \prec A/T^{[\lambda_1]}$. So proposition (2) is proven.

In summary, the theorem is proven. $\square$

### C.2.6. PROOF OF THEOREM 4.7

*Proof.* For proposition (1).

Because $\mathbb{P}, \mathbb{Q} \in \mathcal{Q}(A,S)$ and $\mathbb{P} \neq \mathbb{Q}$, according to proposition (2) of **Theorem** 4.6, $\mathbb{P}$ and $\mathbb{Q}$ are comparable under the sense of $\prec$. Without losing generality, assume that $\mathbb{P} \prec \mathbb{Q}$, then $\mathbb{P} \preceq \mathbb{Q}$. According to **Lemma** A.13, $|\mathbb{P}| \neq |\mathbb{Q}|$. So proposition (1) is proven.

For proposition (2).

According to **Definition** 4.3, $\forall \mathbb{P} \in \mathcal{Q}(A, S)$, $\mathbb{P}$ is a partition on $A$. And then, according to **Definition** A.6,

$$1 = |\{A\}| \leq |\mathbb{P}| \leq |\{\{a\}|\, a \in A\}| = |A|. \quad (40)$$

Combing proposition (1) and formula (40), $|\mathcal{Q}(A, S)| \leq |A|$. So proposition (2) is proven.

For proposition (3). Let

$$C = \{|\mathbb{P}|\,|\, \mathbb{P} \in \mathcal{Q}(A, S)\}, \quad (41a)$$

$$\forall c \in C, \pi(c) = \left\{ u \,\middle|\, u \in U, \left| A/traclo\left(S^{[u]}\right) \right| = c \right\}, \quad (41b)$$

$$V = \left\{ \max_{u \in \pi(c)} u \,\middle|\, c \in C \right\}. \quad (41c)$$

First, we prove $|V| = |\mathcal{Q}(A, S)|$. According to proposition (1) $|C| = |\mathcal{Q}(A, S)|$. And then, according to formula (41), $|C| = |V|$. Therefore, $|V| = |\mathcal{Q}(A, S)|$.

Second, we prove $V \subset U$. According to formula (41), $V \subseteq U$. Because $|U| > |\mathcal{Q}(A, S)| = |V|$, therefore, $V \subset U$.

At last, we prove $\mathcal{Q}(A, S) = \left\{ A/traclo\left(S^{[\lambda]}\right) \middle| \lambda \in V \right\}$. According to **Theorem** 4.4, $\mathcal{Q}(A, S) = \left\{ A/traclo\left(S^{[\lambda]}\right) \middle| \lambda \in U \right\}$. Therefore, we need to prove

$$\left\{ A/traclo\left(S^{[\lambda]}\right) \middle| \lambda \in U \right\} = \left\{ A/traclo\left(S^{[\lambda]}\right) \middle| \lambda \in V \right\},$$

we need to prove

$$\left\{ A/traclo\left(S^{[\lambda]}\right) \middle| \lambda \in U \right\} \supseteq \left\{ A/traclo\left(S^{[\lambda]}\right) \middle| \lambda \in V \right\}, \quad (42a)$$

$$\left\{ A/traclo\left(S^{[\lambda]}\right) \middle| \lambda \in U \right\} \subseteq \left\{ A/traclo\left(S^{[\lambda]}\right) \middle| \lambda \in V \right\}. \quad (42b)$$

Because $V \subset U$, therefore, (42a) is true.

Next, we prove formula (42b). To prove formula (42b), we need to prove $\forall u \in U, \exists v \in V$, such that

$$A/traclo\left(S^{[u]}\right) = A/traclo\left(S^{[v]}\right).$$

$\forall u \in U, \exists c \in C$, such that $\left| A/traclo\left(S^{[u]}\right) \right| = c$. According to formula (41), $\exists v \in V$ such that $\left| A/traclo\left(S^{[v]}\right) \right| = c$, i.e., $\forall u \in U, \exists v \in V$ such that $\left| A/traclo\left(S^{[u]}\right) \right| = \left| A/traclo\left(S^{[v]}\right) \right|$. According to the inverse negation of proposition (1), $A/traclo\left(S^{[u]}\right) = A/traclo\left(S^{[v]}\right)$, i.e., formula (42b) is true.

So far, formula (42a) and formula (42b) are proven. So proposition (3) is proven.

For proposition (4).

According to **Theorem** 4.4, $\mathcal{Q}(A, S) = \left\{ A/traclo\left(S^{[\lambda]}\right) \middle| \lambda \in U \right\}$, therefore $|U| \geq |\mathcal{Q}(A, S)|$. Therefore, to prove $|U| = |\mathcal{Q}(A, S)|$, we need to prove $\forall \lambda_1, \lambda_2 \in U, \lambda_1 < \lambda_2$,

$$A/traclo\left(S^{[\lambda_1]}\right) \neq A/traclo\left(S^{[\lambda_2]}\right),$$

Because $S$ is a FER on $A$, according to **Theorem** A.18, $\forall u \in U, S^{[u]}$ is an ER on $A$. Therefore $S^{[u]}$ satisfies transitivity. According to **Definition** A.19, $traclo\left(S^{[u]}\right) = S^{[u]}$, therefore we need to prove

$$\forall \lambda_1, \lambda_2 \in U, \lambda_1 < \lambda_2, \ A/S^{[\lambda_1]} \neq A/S^{[\lambda_2]}.$$

According to **Definition** A.8, we need to prove

$$\forall \lambda_1, \lambda_2 \in U, \lambda_1 < \lambda_2, \ S^{[\lambda_1]} \neq S^{[\lambda_2]},$$

we need to prove $\forall \lambda_1, \lambda_2 \in U, \lambda_1 < \lambda_2$,

$$\exists a, b \in A, \ \text{such that } S^{[\lambda_1]}((a, b)) \neq S^{[\lambda_2]}((a, b)).$$

Because $\lambda_1 \in U$, therefore, $\exists a, b \in A$, such that $S((a, b)) = \lambda_1$. In this case,

$$S^{[\lambda_1]}((a, b)) = \mathbb{I}[S((a, b)) \geq \lambda_1] = \mathbb{I}[\lambda_1 \geq \lambda_1] = 1$$
$$\neq S^{[\lambda_2]}((a, b)) = \mathbb{I}[S((a, b)) \geq \lambda_2] = \mathbb{I}[\lambda_1 \geq \lambda_2] = 0,$$

therefore proposition (4) is proven.

In summary, propositions (1)-(4) are proven. So the theorem is proven. $\square$

C.2.7. PROOF OF THEOREM 4.8

*Proof.* Let

$$U = \{ R((a, b))|\, a, b \in A \},$$
$$\mathbf{u} = sortvec(U) = (u_1, u_2, \cdots, u_{|U|})^T,$$
$$V = \{ S((a, b))|\, a, b \in A \},$$
$$\mathbf{v} = sortvec(V) = (v_1, v_2, \cdots, v_{|V|})^T.$$

According to formula (10),

$$|U| = |V|, \quad (43a)$$
$$\forall a, b \in A, sort(R((a, b)); U) = sort(S((a, b)); V), \quad (43b)$$
$$\forall a, b, c, d \in A, \quad (43c)$$
$$R((a, b)) < R((c, d)) \ \text{iff} \ S((a, b)) < S((c, d)).$$

According to **Theorem** 4.4, $\mathcal{Q}(A, R) = \left\{ A/traclo\left(R^{[\lambda]}\right) \middle| \lambda \in U \right\}$, $\mathcal{Q}(A, S) = \left\{ A/traclo\left(S^{[\lambda]}\right) \middle| \lambda \in V \right\}$. Therefore, to prove $\mathcal{Q}(A, R) = \mathcal{Q}(A, S)$, we need to prove

$$\left\{ A/traclo\left(R^{[\lambda]}\right) \middle| \lambda \in U \right\} = \left\{ A/traclo\left(S^{[\lambda]}\right) \middle| \lambda \in V \right\},$$

we need to prove

$$\left\{ A/traclo\left(R^{[\lambda]}\right)\Big| \lambda \in U \right\} \subseteq \left\{ A/traclo\left(S^{[\lambda]}\right)\Big| \lambda \in V \right\},$$
(44a)

$$\left\{ A/traclo\left(S^{[\lambda]}\right)\Big| \lambda \in V \right\} \subseteq \left\{ A/traclo\left(R^{[\lambda]}\right)\Big| \lambda \in U \right\}.$$
(44b)

First, we prove formula (44a). To prove formula (44a), we need to prove $\forall u \in U$, $\exists v \in V$, such that

$$A/traclo\left(R^{[u]}\right) = A/traclo\left(S^{[v]}\right),$$

according to **Definition** A.8, we need to prove $\forall u \in U$, $\exists v \in V$, such that

$$traclo\left(R^{[u]}\right) = traclo\left(S^{[v]}\right),$$

according to**Definition** A.19, we need to prove

$$\forall u \in U, \ \exists v \in V, \ \text{such that } R^{[u]} = S^{[v]},$$

according to **Definition** A.17, we need to prove $\forall u \in U$, $\exists v \in V$, such that $\forall a, b \in A$, $R^{[u]}((a,b)) = S^{[v]}((a,b))$, i.e.,

$$\mathbb{I}\left(R((a,b)) \geq u\right) = \mathbb{I}\left(S((a,b)) \geq v\right). \tag{45}$$

According to **Definition** 3.2 and formula (43a), $sort(u;U) \in \{1,2,\cdots,|U|\} = \{1,2,\cdots,|V|\}$, therefore, let $v = v_{sort(u;U)} \in V$. $\forall u \in U$, Because $u \in U$, therefore, $\exists c, d \in A$, such that $u = R((c,d))$. According to (43b), $sort(S((c,d));V) = sort(R((c,d));U) = sort(u;U) = sort(v;V)$, therefore $v = v_{sort(u;U)} = v_{sort(S((c,d));V)} = S((c,d))$.

Next, we prove formula (45) in two cases.

(a) $\mathbb{I}\left(R((a,b)) \geq u\right) = 1$, i.e., $R((a,b)) \geq u = R((c,d))$. According to (10), $S((a,b)) \geq S((c,d)) = v$, i.e., $\mathbb{I}\left(S((a,b)) \geq v\right) = 1$.

(b) $\mathbb{I}\left(R((a,b)) \geq u\right) = 0$, i.e., $R((a,b)) < u = R((c,d))$. According to(43c), we have $S((a,b)) < S((c,d)) = v$, i.e., $\mathbb{I}\left(S((a,b)) \geq v\right) = 0$.

Combining (a) and (b), formula (45) is true. So formula (44a) is true. Similarly, we can prove formula (44b). So far, formula (44) is proven. So the theorem is proven. $\square$

### C.2.8. PROOF OF THEOREM 4.11

***(1) We introduce the following lemmas.***

**Lemma C.5.** *For a $(\mathcal{X}, \mathcal{Y}, f_c)$-classification problem, given a set of samples $\phi \subset X \subseteq \mathcal{X}$ and a fuzzy binary relation $S$ on $\mathcal{Y}$. Let $span(S, f_c, X)$ be the fuzzy binary relation on*

*X described in **Definition** 4.10. The following propositions are true.*

*(1) If $S$ satisfies reflexivity, then $span(S, f_c, X)$ satisfies reflexivity.*

*(2) If $S$ satisfies symmetry, then $span(S, f_c, X)$ satisfies symmetry.*

*(3) If $S$ satisfies transitivity, then $span(S, f_c, X)$ satisfies transitivity.*

*(4) $\forall \lambda \in [0, 1]$, $span(S, f_c, X)^{[\lambda]} = span\left(S^{[\lambda]}, f_c, X\right)$.*

*(5) The $span(\cdot, \cdot, \cdot)$ can be directly applied to binary relation and proposition (1)-(4) are also true for binary relation.*

*Proof.* For proposition (1).

Because $S$ satisfies reflexivity, i.e., $\forall y \in \mathcal{Y}$, $S((y,y)) = 1$. $\forall x \in X$, $span(S, f_c, X)((x,x)) = S((f_c(x), f_c(x))) = 1$, i.e., $span(S, f_c, X)$ satisfies reflexivity. So proposition (1) is proven.

For proposition (2).

Because $S$ satisfies symmetry, i.e., $\forall y_1, y_2 \in \mathcal{Y}$, $S((y_1, y_2)) = ((y_2, y_1))$. $\forall x_1, x_2 \in X$,

$$\begin{aligned}span(S, f_c, X)((x_1, x_2)) &= S((f_c(x_1), f_c(x_2))) \\ &= S((f_c(x_1), f_c(x_2))) \\ &= span(S, f_c, X)((x_1, x_2)),\end{aligned}$$

i.e., $span(S, f_c, X)$ satisfies symmetry. So proposition (2) is proven.

For proposition (3).

Because $X \subset \mathcal{X}$, therefore $Y = \{ f_c(x)| \ x \in X \} \subseteq \mathcal{Y}$. Because $S$ satisfies transitivity, i.e., $\forall y_1, y_3 \in \mathcal{Y}$, $S((y_1, y_3)) \geq \max_{y_2 \in \mathcal{Y}} \min\left[S((y_1, y_2)), S((y_2, y_3))\right]$. $\forall x_1, x_3 \in X$,

$$\begin{aligned}&span(S, f_c, X)(x_1, x_3) \\ &= S((f_c(x_1), f_c(x_3))) \\ &\geq \max_{y_2 \in \mathcal{Y}} \min\left[S((f_c(x_1), y_2)), S((y_2, f_c(x_3)))\right] \\ &\geq \max_{y_2 \in Y} \min\left[S((f_c(x_1), y_2)), S((y_2, f_c(x_3)))\right] \\ &= \max_{x_2 \in X} \min\left[S((f_c(x_1), f_c(x_2))), S((f_c(x_2), f_c(x_3)))\right],\end{aligned}$$

i.e. $span(S, f_c, X)$ satisfies transitivity. So proposition (3) is proven.

For proposition (4).

$\forall \lambda \in [0, 1], \forall x_1, x_2 \in X$,

$$\begin{aligned}&span(S, f_c, X)^{[\lambda]}(x_1, x_2) \\ &= \mathbb{I}\left(span(S, f_c, X)(x_1, x_2) \geq \lambda\right) \\ &= \mathbb{I}\left(S(f_c(x_1), f_c(x_2)) \geq \lambda\right) \\ &= S^{[\lambda]}(f_c(x_1), f_c(x_2)) = span\left(S^{[\lambda]}, f_c, X\right),\end{aligned}$$

i.e., proposition (4) is proven.

For proposition (5).

Obviously, the above proofs of proposition (1)-(4) are also valid under the sense of binary relation.

In summary, proposition (1)-(5) are proven. So the lemma is proven. $\qquad\square$

**Lemma C.6.** *For a $(\mathcal{X}, \mathcal{Y}, f_c)$-classification problem, given a set of samples $\phi \subset X \subseteq \mathcal{X}$ and a fuzzy binary relation $S$ on $\mathcal{Y}$. Let $span(S, f_c, X)$ be the fuzzy binary relation on $X$ described in **Definition** 4.10. If $\forall y \in \mathcal{Y}, \exists x \in X$, such that $f_c(x) = y$, then $span\left(S, f_c, X\right)^2 = span\left(S^2, f_c, X\right)$.*

*Proof.* Because $\forall y \in \mathcal{Y}, \exists x \in X$, such that $f_c(x) = y$, therefore, $Y = \{f_c(x)|\ x \in X\} = \mathcal{Y}$. $\forall x_1, x_3 \in X$,

$$
\begin{aligned}
&span\left(S, f_c, X\right)^2\left((x_1, x_3)\right)\\
&= \max_{x_2 \in X} \min \begin{bmatrix} span\left(S, f_c, X\right)\left((x_1, x_2)\right),\\ span\left(S, f_c, X\right)\left((x_2, x_3)\right)\end{bmatrix}\\
&= \max_{x_2 \in X} \min\left[S\left((f_c(x_1), f_c(x_2))\right), S\left((f_c(x_2), f_c(x_3))\right)\right]\\
&= \max_{y \in Y = \mathcal{Y}} \min\left[S\left((f_c(x_1), y)\right), S\left((y, f_c(x_3))\right)\right]\\
&= S^2\left((f_c(x_1), f_c(x_3))\right)\\
&= span\left(S^2, f_c, X\right)\left((x_1, x_3)\right),
\end{aligned}
$$

i.e., the lemma is proven. $\qquad\square$

**Lemma C.7.** *For a $(\mathcal{X}, \mathcal{Y}, f_c)$-classification problem, given a set of samples $\phi \subset X \subseteq \mathcal{X}$ and a FSR $S$ on $\mathcal{Y}$. Let $span(S, f_c, X)$ be the fuzzy binary relation on $X$ described in **Definition** 4.10. If $\forall y \in \mathcal{Y}, \exists x \in X$, such that $f_c(x) = y$, then $traclo\left(span\left(S, f_c\right)\right) = span\left(traclo\left(S\right), f_c, X\right)$.*

*Proof.* Because $S$ is a FSR on $\mathcal{Y}$, therefore, $S$ satisfies reflexivity and symmetry. According to **Lemma** C.5, $span\left(S, f_c, X\right)$ satisfies reflexivity and symmetry, i.e., $span\left(S, f_c, X\right)$ is a FSR on $X$.

According to **Theorem** 4.1, $traclo\left(S\right)$ can be written in form of $S^{2^k}$, $k \geq 0$, and $traclo\left(span\left(S, f_c, X\right)\right)$ can be Written in form of $span\left(S, f_c, X\right)^{2^l}$, $l \geq 0$. Let $m = \max\left(k, l\right)$, then $traclo\left(S\right) = S^{2^m}$, $traclo\left(span\left(S, f_c, X\right)\right) = span\left(S, f_c, X\right)^{2^m}$.

Therefore, to prove

$$traclo\left(span\left(S, f_c\right)\right) = span\left(traclo\left(S\right), f_c, X\right),$$

we need to prove

$$span\left(S, f_c, X\right)^{2^m} = span\left(S^{2^m}, f_c, X\right). \quad (46)$$

According to **Lemma** C.6,

$$
\begin{aligned}
&\forall y \in \mathcal{Y}, \exists x \in X, \text{ such that } f_c(x) = y\\
&\Rightarrow span\left(S, f_c, X\right)^2 = span\left(S^2, f_c, X\right)\\
&\Rightarrow span\left(S, f_c, X\right)^4 = span\left(S^4, f_c, X\right)\\
&\Rightarrow \cdots\\
&\Rightarrow span\left(S, f_c, X\right)^{2^{m-1}} = span\left(S^{2^{m-1}}, f_c, X\right)\\
&\Rightarrow span\left(S, f_c, X\right)^{2^m} = span\left(S^{2^m}, f_c, X\right),
\end{aligned}
$$

i.e., formula (46) is true. So the lemma is proven. $\qquad\square$

**Lemma C.8.** *For a $(\mathcal{X}, \mathcal{Y}, f_c)$-classification problem, given a set of samples $\phi \subset X \subseteq \mathcal{X}$ and an ER $R$ on $\mathcal{Y}$. Let $\mathbb{O} = \mathcal{Y}/R$ be the partition $\mathcal{Y}$ derived by $R$. Let $span(R, f_C, X)$ be the binary relation on $X$ described in **Definition** 4.10. According to **Lemma** C.5, $span(R, f_C, X)$ is an ER on $X$. Let $\mathbb{P} = X/span(R, f_c, X)$ be the partition on $X$ derived by $span(R, f_c, X)$. If $\forall y \in \mathcal{Y}, \exists x \in X$ such that $f_c(x) = y$, then the following propositions are true.*

*(1) $\forall O \in \mathbb{O}$, $\{x\,|\,x \in X, f_c(x) \in O\} \in \mathbb{P}$.*

*(2) $\forall P \in \mathbb{P}$, $\{f_c(x)|\ x \in P\} \in \mathbb{O}$.*

*(3) $|\mathbb{O}| = |\mathbb{P}|$.*

*Proof.* For proposition (1).

$\forall O \in \mathbb{O}$, obviously, $\phi \neq O \subseteq \mathcal{Y}$. $\forall y^* \in O$, according to **Definition** A.7 and A.8,

$$O = [y^*]_R = \{y : y \in \mathcal{Y}, R((y^*, y)) = 1\}. \quad (47)$$

Because $\forall y \in \mathcal{Y}, \exists x \in X$ such that $f_c(x) = y$. Without losing generality, assume that $x^* \in X$ and $f_c(x^*) = y^*$. And then,

$$
\begin{aligned}
&[x^*]_{span(R, f_c, X)}\\
&= \{x\,|\,x \in X, span\left(R, f_c, X\right)\left((x^*, x)\right) = 1\}\\
&= \{x\,|\,x \in X, R((f_c(x^*), f_c(x))) = 1\}\\
&= \{x\,|\,x \in X, R((y^*, f_c(x))) = 1\}\\
&= \{x\,|\,x \in X, f_c(x) \in [y^*]_R = O\}.
\end{aligned}
$$

According to **Definition** A.8, $[x^*]_{span(R, f_c, X)} \in \mathbb{P}$, i.e., proposition (1) is proven.

For proposition (2).

$\forall P \in \mathbb{P}$, obviously, $\phi \neq P \subseteq X$, $\forall x^* \in P$, according to **Definition** A.7 and A.8,

$$
\begin{aligned}
P &= [x^*]_{span(R, f_c, X)}\\
&= \{x\,|\,x \in X, span(R, f_c, X)((x^*, x)) = 1\}.
\end{aligned} \quad (48)
$$

Let $y^* = f_c(x^*)$, obviously, $y^* \in \mathcal{Y}$, then

$$
\begin{aligned}
[y^*]_R &= \{y\,|\,y \in \mathcal{Y}, R((y^*, y)) = 1\}\\
&= \{y\,|\,y \in \mathcal{Y}, R((f_c(x^*), y)) = 1\}\\
&= \{f(x)\,|\,x \in X, R((f_c(x^*), f_c(x))) = 1\}\\
&= \{f(x)\,|\,x \in X, span(R, f_c, X)((x^*, x)) = 1\}\\
&= \{f(x)\,|\,x \in X, x \in [x^*]_{span(R, f_c, X)} = P\}.
\end{aligned}
$$

According to **Definition** A.8, $[y^*]_R \in \mathbb{O}$, i.e., proposition (2) is proven.

For proposition (3).

Proposition (1) and proposition (2) actually establish a one-to-one mapping between the equivalence classes in $\mathbb{O}$ and the equivalence classes in $\mathbb{P}$, therefore, $|\mathbb{O}| = |\mathbb{P}|$.

In summary, proposition (1)-(3) are proven. So the lemma is proven. $\square$

**Lemma C.9.** *For a $(\mathcal{X}, \mathcal{Y}, f_c)$-classification problem, given a set of samples $\phi \subset X \subseteq \mathcal{X}$ and two FSRs $R$, $S$ on $\mathcal{Y}$. If $\mathcal{Q}(\mathcal{Y}, R) = \mathcal{Q}(\mathcal{Y}, S)$ and $\forall k \in \mathcal{Y}$, $\exists x \in X$, such that $f_c(x) = k$, then $\mathcal{Q}(X, span(R, f_c, X)) = \mathcal{Q}(X, span(S, f_c, X))$.*

*Proof.* To prove $\mathcal{Q}(X, span(R, f_c, X)) = \mathcal{Q}(X, span(S, f_c, X))$, according to **Theorem** 4.5, we need to prove $\mathcal{Q}(X, traclo(span(R, f_c, X))) = \mathcal{Q}(X, traclo(span(S, f_c, X)))$. Because $\forall k \in \mathcal{Y}$, $\exists x \in X$, such that $f_c(x) = k$, according to **Lemma** C.7, we need to prove

$$\mathcal{Q}(X, span(traclo(R), f_c, X)) = \mathcal{Q}(X, span(traclo(S), f_c, X)). \tag{49}$$

Because $R$ and $S$ are both FSR on $\mathcal{Y}$. According to **Theorem** 4.1, $traclo(R)$ and $traclo(S)$ are both FER on $\mathcal{Y}$. According to **Lemma** C.5, $span(traclo(R), f_c, X)$ and $span(traclo(S), f_c, X)$ are both FER on $X$. According to **Theorem** A.18, $\forall \mu \in [0, 1]$, $span(traclo(R), f_c, X)^{[\mu]}$ and $span(traclo(S), f_c, X)^{[\mu]}$ are both ER on $X$. According to **Definition** 4.3, to prove formula (49), we need to prove

$$\left\{ X/span(traclo(R), f_c, X)^{[\lambda]} \middle| \lambda \in [0, 1] \right\} = \left\{ X/span(traclo(S), f_c, X)^{[\omega]} \middle| \omega \in [0, 1] \right\}.$$

According to **Definition** A.8, we need to prove

$$\left\{ span(traclo(R), f_c, X)^{[\lambda]} \middle| \lambda \in [0, 1] \right\} = \left\{ span(traclo(S), f_c, X)^{[\omega]} \middle| \omega \in [0, 1] \right\}.$$

According to **Lemma** C.5, we need to prove

$$\left\{ span\left(traclo(R)^{[\lambda]}, f_c, X\right) \middle| \lambda \in [0, 1] \right\} = \left\{ span\left(traclo(S)^{[\omega]}, f_c, X\right) \middle| \omega \in [0, 1] \right\}.$$

According to **Definition** 4.10, we need to prove

$$\left\{ traclo(R)^{[\lambda]} \middle| \lambda \in [0, 1] \right\} = \left\{ traclo(S)^{[\omega]} \middle| \omega \in [0, 1] \right\}. \tag{50}$$

Next, we prove formula (50). Because $\mathcal{Q}(\mathcal{Y}, R) = \mathcal{Q}(\mathcal{Y}, S)$, according to **Theorem** 4.5, $\mathcal{Q}(\mathcal{Y}, traclo(R)) = \mathcal{Q}(\mathcal{Y}, traclo(S))$. Because $R$ and $S$ are both FSR on $\mathcal{Y}$, according to **Theorem** 4.1, $traclo(R)$ and $traclo(S)$ are both FER on $\mathcal{Y}$. Based on it, according to **Definition** 4.3, $\left\{ \mathcal{Y}/traclo(R)^{[\lambda]} \middle| \lambda \in [0, 1] \right\} = \left\{ \mathcal{Y}/traclo(S)^{[\omega]} \middle| \omega \in [0, 1] \right\}$. Because $traclo(R)$ and $traclo(S)$ are both FER on $\mathcal{Y}$, according to **Theorem** A.18, $\forall \mu \in [0, 1]$, $traclo(R)^{[\mu]}$ and $traclo(S)^{[\mu]}$ are both ER on $\mathcal{Y}$. According to **Definition** A.8,

$$\left\{ traclo(R)^{[\lambda]} \middle| \lambda \in [0, 1] \right\} = \left\{ traclo(S)^{[\omega]} \middle| \omega \in [0, 1] \right\},$$

i.e., formula (50) is proven. So formula (49) is proven. So the lemma is proven. $\square$

***(2) Based on these conclusions, we prove Theorem 4.11.***

*Proof.* Let $traclo(\mathbf{T})$ be the transitive-closure of $\mathbf{T}$ (see **Definition** A.19). Let

$$U = \{ traclo(\mathbf{T})_{ij} | \ i, j \in \mathcal{Y} \}, \tag{51a}$$

$$V = \left\{ span(traclo(\mathbf{T}), f_c, X)_{ij} \middle| \ x_i, x_j \in X \right\}. \tag{51b}$$

And then,

$$\begin{aligned} V &= \left\{ span(traclo(\mathbf{T}), f_c, X)_{ij} \middle| \ x_i, x_j \in X \right\} \\ &= \left\{ traclo(\mathbf{T})_{f_c(x_i) f_c(x_j)} \middle| \ x_i, x_j \in X \right\} \\ &= \left\{ traclo(\mathbf{T})_{y_i y_j} \middle| \ y_i, y_j \in \mathcal{Y} \right\} \\ &= U. \end{aligned} \tag{52}$$

Because $\mathbf{T}$ is a FSR on $\mathcal{Y}$, according to **Theorem** 4.1, $traclo(\mathbf{T})$ is a FER on $\mathcal{Y}$. According to **Theorem** A.18, $\forall \lambda \in [0, 1]$, $traclo(\mathbf{T})^{[\lambda]}$ is an ER on $\mathcal{Y}$. And then, according to **Definition** A.19,

$$\forall \lambda \in [0, 1], traclo\left(traclo(\mathbf{T})^{[\lambda]}\right) = traclo(\mathbf{T})^{[\lambda]}. \tag{53}$$

Meanwhile, because $traclo(\mathbf{T})$ is FER on $\mathcal{Y}$, according to **Lemma** C.5 $span(traclo(\mathbf{T}), f_c, X)$ s a FER on $X$. According to **Theorem** A.18, $\forall \lambda \in [0, 1]$, $span(traclo(\mathbf{T}), f_c, X)^{[\lambda]}$ is an ER on $X$. And then, according to **Definition** A.19, $\forall \lambda \in [0, 1]$,

$$\begin{aligned} traclo\left(span(traclo(\mathbf{T}), f_c, X)^{[\lambda]}\right) = \\ span(traclo(\mathbf{T}), f_c, X)^{[\lambda]}. \end{aligned} \tag{54}$$

Given $V$, $\mathcal{Q}(\mathcal{Y}, \mathbf{T})$ can be written as follows

$$
\begin{aligned}
\mathcal{Q}(\mathcal{Y}, \mathbf{T}) &= \mathcal{Q}(\mathcal{Y}, traclo(\mathbf{T})) \\
&= \left\{ \mathcal{Y}/traclo\left(traclo(\mathbf{T})^{[\lambda]}\right) \middle| \lambda \in [0,1] \right\} \\
&= \left\{ \mathcal{Y}/traclo(\mathbf{T})^{[\lambda]} \middle| \lambda \in [0,1] \right\} \\
&= \left\{ \mathcal{Y}/traclo(\mathbf{T})^{[\lambda]} \middle| \lambda \in U \right\} \\
&= \left\{ \mathcal{Y}/traclo(\mathbf{T})^{[\lambda]} \middle| \lambda \in V \right\}.
\end{aligned}
\tag{55}
$$

Given $V$, $\mathcal{Q}(X, span(\mathbf{T}, f_c, X))$ can be written as follows

$$
\begin{aligned}
&\mathcal{Q}(X, span(\mathbf{T}, f_c, X)) \\
&= \mathcal{Q}(X, traclo(span(\mathbf{T}, f_c, X))) \\
&= \mathcal{Q}(X, span(traclo(\mathbf{T}), f_c, X)) \\
&= \left\{ X/traclo\left(span(traclo(\mathbf{T}), f_c, X)^{[\lambda]}\right) \middle| \lambda \in [0,1] \right\} \\
&= \left\{ X/span(traclo(\mathbf{T}), f_c, X)^{[\lambda]} \middle| \lambda \in [0,1] \right\} \\
&= \left\{ X/span(traclo(\mathbf{T}), f_c, X)^{[\lambda]} \middle| \lambda \in [0,1] \right\} \\
&= \left\{ X/span(traclo(\mathbf{T}), f_c, X)^{[\lambda]} \middle| \lambda \in V \right\} \\
&= \left\{ X/span(traclo(\mathbf{T}), f_c, X)^{[\lambda]} \middle| \lambda \in [0,1] \right\} \\
&= \left\{ X/span\left(traclo(\mathbf{T})^{[\lambda]}, f_c, X\right) \middle| \lambda \in V \right\}.
\end{aligned}
\tag{56}
$$

For proposition (1).

In formula (55), because $traclo(\mathbf{T})$ is a FER on $\mathcal{Y}$, according to **Theorem** 4.7, $|\mathcal{Q}(\mathcal{Y}, \mathbf{T})| = |V|$. In formula (56), because $span(traclo(\mathbf{T}), f_c, X)$ is a FER on $X$, according to **Theorem** 4.7, $|\mathcal{Q}(X, span(\mathbf{T}, f_c, X))| = |V|$. Soproposition (1) is proven.

For proposition (2).

Without losing generality, we arrange the elements in the set $V$ in descending order to obtain $\lambda_1 > \lambda_2 > \cdots > \lambda_{|V|}$. According to **Definition** A.17, $\forall i = 1, 2, \cdots, |V| - 1$, $traclo(\mathbf{T})^{[\lambda_i]} \subset \mathcal{Y}/traclo(\mathbf{T})^{[\lambda_{i+1}]}$, $span\left(traclo(\mathbf{T})^{[\lambda_i]}, f_c, X\right) \subset span\left(traclo(\mathbf{T})^{[\lambda_{i+1}]}, f_c, X\right)$. And then, according to **Theorem** 4.6, $\mathcal{Y}/traclo(\mathbf{T})^{[\lambda_i]} \prec \mathcal{Y}/traclo(\mathbf{T})^{[\lambda_{i+1}]}$, $X/span\left(traclo(\mathbf{T})^{[\lambda_i]}, f_c, X\right) \prec X/span\left(traclo(\mathbf{T})^{[\lambda_{i+1}]}, f_c, X\right)$. Based on it, according to **Definition** 4.3 and 4.9,

$$
(\mathcal{Q}(\mathcal{Y}, \mathbf{T}), \prec) =
\begin{bmatrix}
\mathbb{O}_1 = \mathcal{Y}/traclo(\mathbf{T})^{[\lambda_1]} \prec \\
\mathbb{O}_2 = \mathcal{Y}/traclo(\mathbf{T})^{[\lambda_2]} \prec \\
\cdots \prec \\
\mathbb{O}_{|V|} = \mathcal{Y}/traclo(\mathbf{T})^{[\lambda_{|V|}]}
\end{bmatrix}, \tag{57}
$$

$$
(\mathcal{Q}(X, span(\mathbf{T}, f_c, X)), \prec) =
\begin{bmatrix}
\mathbb{P}_1 = X/span\left(traclo(\mathbf{T})^{[\lambda_1]}, f_c, X\right) \prec \\
\mathbb{P}_2 = X/span\left(traclo(\mathbf{T})^{[\lambda_2]}, f_c, X\right) \prec \\
\cdots \prec \\
\mathbb{P}_{|V|} = X/span\left(traclo(\mathbf{T})^{[\lambda_{|V|}]}, f_c, X\right)
\end{bmatrix}. \tag{58}
$$

In formula (57) and (58), because $\mathbf{T}$ is a FSR on $\mathcal{Y}$, according to **Theorem** 4.1, $traclo(\mathbf{T})$ is a FER on $\mathcal{Y}$. And then, according to **Theorem** A.18, $\forall i = 1, 2, \cdots, |V|$, $traclo(\mathbf{T})^{[\lambda_i]}$ is an ER on $\mathcal{Y}$. Meanwhile, because $\forall y \in \mathcal{Y}$, $\exists x \in X$ such that $f_c(x) = y$. Based on it, $\forall i = 1, 2, \cdots, |V|$, according to **Lemma** C.8, $\mathbb{O}_i$ and $\mathbb{P}_i$ satisfy the three conditions in proposition (2). So proposition (2) is proven.

In summary, proposition (1) and (2) are proven. So the theorem is proven. $\square$

### C.2.9. Proof of Theorem 4.12

*Proof.* For sake of discussion, let $\bar{\mathbf{T}} = span(\mathbf{T}, f_c, X)$. According to **Definition** 4.3 and 4.9, to prove $\forall i = 1, 2, \cdots, |\mathcal{Q}(X, \bar{\mathbf{T}})|$, $\exists j \in \{1, 2, \cdots, |\mathcal{Q}(X, \mathbf{S})|\}$, such that $\mathbb{P}_i = \mathbb{Q}_j$ we need to prove $\forall \mathbb{P} \in \mathcal{Q}(X, \bar{\mathbf{T}})$, $\exists \mathbb{Q} \in \mathcal{Q}(X, \mathbf{S})$, such that $\mathbb{P} = \mathbb{Q}$. we need to prove

$$
\mathcal{Q}(X, \bar{\mathbf{T}}) \subseteq \mathcal{Q}(X, \mathbf{S}). \tag{59}
$$

Next, we prove formula (59). For sake of discussion, we copy formula (7) as follows

$$
V = \{t_{ij} | \ i, j \in \mathcal{Y}\}, \quad U = V \cup \{0, \alpha, \beta\},
$$
$$
\mathbf{u} = sortvec(U) = (u_1, u_2, \cdots, u_{|U|})^T.
$$

According to **Definition** 4.10, let

$$
\bar{V} = \left\{ \bar{t}_{ij} \middle| \bar{t}_{ij} = span(\mathbf{T}, f_c, X)_{ij} = t_{y_i y_j}, i, j = 1, \cdots, n \right\},
$$

and $\bar{V} \subseteq V$. And then, according to **Theorem** 4.4,

$$
\mathcal{Q}(X, \bar{\mathbf{T}}) = \left\{ X/traclo\left(\bar{\mathbf{T}}^{[\lambda]}\right) \middle| \lambda \in \bar{V} \right\}.
$$

Therefore, to prove formula (59), we need to prove $\forall \lambda \in \bar{V}$, $\exists \omega \in [0,1]$, such that $X/traclo\left(\bar{\mathbf{T}}^{[\lambda]}\right) = X/traclo\left(\mathbf{S}^{[\omega]}\right)$. According to **Definition** A.8, we need to prove $\forall \lambda \in \bar{V}$, $\exists \omega \in [0,1]$, such that $traclo\left(\bar{\mathbf{T}}^{[\lambda]}\right) = traclo\left(\mathbf{S}^{[\omega]}\right)$. According to **Theorem** 4.1, we need to prove $\forall \lambda \in \bar{V}$, $\exists \omega \in [0,1]$, such that, $\bar{\mathbf{T}}^{[\lambda]} = \mathbf{S}^{[\omega]}$. According to **Definition** A.17, we need to prove $\forall \lambda \in \bar{V}$, $\exists \omega \in [0,1]$, such that $\forall i, j = 1, 2, \cdots, n$, $\mathbb{I}[\bar{t}_{ij} \geq \lambda] = \mathbb{I}[s_{ij} \geq \omega]$. According to **Definition** 4.10, $\forall i, j = 1, 2, \cdots, n$, $\bar{t}_{ij} = t_{y_i y_j}$. Therefore, we need to prove $\forall \lambda \in \bar{V}$, $\exists \omega \in [0,1]$, such that $\forall i, j = 1, 2, \cdots, n$,

$$
\mathbb{I}\left[t_{y_i y_j} \geq \lambda\right] = \mathbb{I}\left[s_{ij} \geq \omega\right]. \tag{60}
$$

Next, we prove formula (60). $\forall \lambda \in \bar{V} \subseteq V$, according to formula (35e), $2 \leq sort(\lambda, U) \leq |U|$. Therefore $1 \leq sort(\lambda, U) - 1 \leq |U| - 1$. Let $\omega = u_{sort(\lambda, U) - 1}$, according to formula (35d) and (35a), $0 = u_1 \leq \omega < \lambda \leq u_{|U|} = 1$, i.e., $\omega \in [0, 1)$. Because $L_1 = 0$, therefore, $\forall i, j = 1, 2, \cdots, n, i \neq j, \mathcal{L}_{HA}(s_{ij}, y_i, y_j; \alpha, \beta, \mathbf{T}) = 0$, i.e., [4]

$$s_{ij} \in \left( u_{sort(t_{y_i y_i}; U) - 1}, u_{sort(t_{y_i y_i}; U)} \right). \tag{61}$$

Next, we prove that $\forall i, j = 1, 2, \cdots, n, \mathbb{I}\left[ t_{y_i y_j} \geq \lambda \right] = \mathbb{I}[s_{ij} \geq \omega]$.

(a) If $i = j$. Because $\mathbf{T}$ is a FSR on $\mathcal{Y}$, $\mathbf{S}$ is a FSR on $X$, therefore, $\mathbb{I}\left[ t_{y_i y_i} \geq \lambda \right] = \mathbb{I}[1 \geq \lambda] = 1 = \mathbb{I}[1 \geq \omega] = \mathbb{I}[s_{ii} \geq \omega]$.

(b) If $i \neq j$, $\mathbb{I}\left[ t_{y_i y_j} \geq \lambda \right] = 1$, i.e., $t_{y_i y_j} \geq \lambda$. According to formula (7), $sort\left( t_{y_i y_j}; U \right) \geq sort(\lambda; U)$, then $sort\left( t_{y_i y_j}; U \right) - 1 \geq sort(\lambda; U) - 1$. And then, according to formula (61), $s_{ij} > u_{sort(t_{y_i y_i}; U) - 1} \geq u_{sort(\lambda; U) - 1} = \omega$, i.e., $\mathbb{I}[s_{ij} \geq \omega] = 1$.

(c) If $i \neq j$, $\mathbb{I}\left[ t_{y_i y_j} \geq \lambda \right] = 0$, i.e., $t_{y_i y_j} < \lambda$. According to formula (7), $sort\left( t_{y_i y_j}; U \right) < sort(\lambda; U)$, therefore $sort\left( t_{y_i y_j}; U \right) \leq sort(\lambda; U) - 1$. And then, according to formula (61), $s_{ij} < u_{sort(t_{y_i y_i}; U)} \leq u_{sort(\lambda; U) - 1} = \omega$, i.e., $\mathbb{I}[s_{ij} \geq \omega] = 0$.

Combining (a)-(c), formula (60) is true. Therefore, formula (59) is true. So the theorem is proven. $\square$

## C.3. Proofs of Theorems in Appendix B

C.3.1. PROOF OF THEOREM B.2

***(1) To prove the theorem B.2, we introduce the following derivation process.***

**Remark C.10.** Given the following convex quadratic programming problem

$$\begin{aligned} \min_{\mathbf{r}} \quad & \mathcal{J}_1(\mathbf{r}) = \sum_{i=1}^m (r_i - s_i)^2 \\ s.t. \quad & r_i = t_{i+1}, \forall i = 1, 2, \cdots, m - 1 \\ & a < r_i < b, \forall i = 1, 2, \cdots, m \end{aligned} \tag{62}$$

$\forall i = 1, 2, \cdots, m$, let $t = r_i$. Meanwhile, let $\bar{s} = \frac{\sum_{i=1}^m s_i}{m}$. And then

$$\begin{aligned} \mathcal{J}_2(\mathbf{r}) \quad & = \sum_{i=1}^m (r_i - s_i)^2 \\ & = \sum_{i=1}^m \left( r_i^2 - 2 r_i s_i + s_i^2 \right) \\ & = \sum_{i=1}^m r_i^2 - 2 \sum_{i=1}^m r_i s_i + \sum_{i=1}^m s_i^2 \\ & = m t^2 - 2 \left( \sum_{i=1}^m s_i \right) t + \sum_{i=1}^m s_i^2 \\ & = m t^2 - 2 m \bar{s} t + \sum_{i=1}^m s_i^2 \\ & = m \left( t^2 - 2 \bar{s} t \right) + \sum_{i=1}^m s_i^2 \\ & = m \left( t - \bar{s} \right)^2 - m \bar{s}^2 + \sum_{i=1}^m s_i^2 \end{aligned}$$

---

[4]According to (8), $s_{ij}$ should fall into the close interval. In practice, we can introduce a small constant $\epsilon > 0$ into formula (8) such that $s_{ij}$ falls in the open interval.

Let $\mathcal{J}_2(t) = (t - \bar{s})^2$, $\mathcal{J}_3(\mathbf{s}) = \left( \sum_{i=1}^m s_i^2 \right) - m \bar{s}^2$, then $\mathcal{J}_1(\mathbf{r}) = m \mathcal{J}_2(t) + \mathcal{J}_3(\mathbf{s})$. Substituting them into the formula (62), we obtain the following optimization problem

$$\begin{aligned} & \min_{\mathbf{r}} \quad \mathcal{J}_1(\mathbf{r}) = \sum_{i=1}^m (r_i - s_i)^2 \\ & s.t. \quad r_i = t_{i+1}, \forall i = 1, 2, \cdots, m - 1 \\ & \qquad a < t < b \\ \Longleftrightarrow \quad & \min_{t} \quad m \mathcal{J}_2(t) + \mathcal{J}_3(\mathbf{s}) \\ & s.t. \quad a < t < b \\ \Longleftrightarrow \quad & \min_{t} \quad m \mathcal{J}_2(t) \\ & s.t. \quad a < t < b. \end{aligned}$$

Obviously, the last optimization problem in the above equation is a convex quadratic programming problem. Let $t^*$ be the optimal solution of it. Then

$$\mathbf{r}^* \in \mathbb{R}^m, \ \forall i = 1, 2, \cdots, m, \ r_i^* = t^*,$$

is the optimal solution of formula (62).

***(2) Based on the above derivation process, we prove Theorem B.2.***

*Proof.* For proposition (1).

According to formula (29), $\mathbf{S}^{(D,K)^*}$ satisfies reflexivity and symmetry. Because $\mathbf{t}^*$ is the optimal solution of formula (27), therefore,

$$\forall i = 1, 2, \cdots, |V^K|, 0 < t_i^* < \alpha.$$

According to formula (29), $\forall i, j \in \mathcal{Y}, i \neq j, 0 < s_{ij}^{(D,K)^*} < \alpha$. Therefore, $\mathbf{S}^{(D,K)^*}$ is a $\alpha$-FSR on $\mathcal{Y}$. So proposition (1) is proven.

For proposition (2).

Because $S^K$ and $S^D$ both satisfy reflexivity and symmetry. Therefore, we can eliminate the third and fourth constraints and ignore the elements of the main diagonal and lower triangle positions of the matrix of the objective function in the formula (26). Therefore, we obtain the following equivalent optimization problem

$$\begin{aligned} \min_{\mathbf{S}^{(D,K)}} \quad & \sum_{i \in \mathcal{Y}} \sum_{j \in \mathcal{Y}, j > i} \left( \mathbf{s}_{ij}^{(D,K)} - \mathbf{s}_{ij}^D \right)^2 \\ s.t. \quad & s_{ij}^{(D,K)} < s_{jk}^{(D,K)}, \forall i, j, k \in \mathcal{Y}, i < j < k, \\ & s_{ij}^K < s_{jk}^K \\ & s_{ij}^{(D,K)} = s_{jk}^{(D,K)}, \forall i, j, k \in \mathcal{Y}, i < j < k, \\ & s_{ij}^K = s_{jk}^K \\ & 0 < s_{ij}^{(D,K)} < \alpha, \ \forall i, j \in \mathcal{Y}, \ i < j \end{aligned} \tag{63}$$

In formula (63), the $\mathbf{S}^{(D,K)}$ satisfies the first two constraints iff the sorting of the triangular elements of $\mathbf{S}^{(D,K)}$ is consistent with the sorting of the triangular elements of $\mathbf{S}^K$. Let

$V^K = \left\{ s_{ij}^K \mid i, j \in \mathcal{Y}, i \neq j \right\}$. Considering the situation where the triangular elements of $\mathbf{S}^K$ have duplicates, let $I_i = \left\{ (k,l) \mid k,l \in \mathcal{Y}, k < l, sort\left(s_{kl}^K; V^K\right) = i \right\}$. Substituting $V^K$ and $I_i$ into the formula (63), we can obtain the following equivalent optimization problem

$$
\begin{aligned}
\min_{\mathbf{S}^{(D,K)}} \quad & \sum_{i=1}^{|V^K|} \sum_{(k,l)\in I_i} \left(\mathbf{s}_{kl}^{(D,K)} - \mathbf{s}_{kl}^D\right)^2 \\
s.t. \quad & s_{pq}^{(D,K)} < s_{rs}^{(D,K)}, \forall i,j = 1,2,\cdots,\left|V^K\right|, \\
& i < j, \ (p,q) \in I_i, (r,s) \in I_j \\
& s_{pq}^{(D,K)} = s_{rs}^{(D,K)}, \forall i = 1,2,\cdots,\left|V^K\right|, \\
& (p,q),(r,s) \in I_i, (p,q) \neq (r,s) \\
& 0 < s_{ij}^{(D,K)} < \alpha, \ \forall i,j \in \mathcal{Y}, i < j.
\end{aligned} \tag{64}
$$

$\forall i = 1,2,\cdots,\left|V^K\right|$, let $\bar{s}_i = \frac{\sum_{(p,q)\in I_i} s_{pq}^D}{|I_i|}$. According to **Remark** C.10, we can can eliminate the equality constraints in the formula (64) and obtain the following equivalent optimization problem.

$$
\begin{aligned}
\min_{\mathbf{t}\in\mathbb{R}^{|V^K|}} \quad & \mathcal{J}_1(\mathbf{t}) = \sum_{i=1}^{|V^K|} |I_i| (t_i - \bar{s}_i)^2 \\
s.t. \quad & t_i < t_{i+1}, \forall i,j = 1,2,\cdots,\left|V^K\right| - 1 \\
& 0 < t_i < \alpha, \forall i = 1,2,\cdots,\left|V^K\right|
\end{aligned} \tag{65}
$$

Substituting $\bar{\mathbf{s}}$ given in formula (28) and $\mathbf{D}$ into formula (65), we can obtain the following optimization problem

$$
\mathcal{J}_1(\mathbf{t}) = \mathbf{t}^T\mathbf{D}\mathbf{t} - 2\bar{\mathbf{s}}^T\mathbf{D}\mathbf{t} + \bar{\mathbf{s}}^T\mathbf{D}\bar{\mathbf{s}}
$$

By removing the unrelated $\bar{\mathbf{s}}^T\mathbf{D}\bar{\mathbf{s}}$ and let $\mathcal{J}_2(\mathbf{t}) = \mathbf{t}^T\mathbf{D}\mathbf{t} - 2\bar{\mathbf{s}}^T\mathbf{D}\mathbf{t}$, we can obtain the following optimization problem

$$
\begin{aligned}
\min_{\mathbf{t}\in\mathbb{R}^{|V^K|}} \quad & \mathcal{J}_2(\mathbf{t}) = \mathbf{t}^T\mathbf{D}\mathbf{t} - 2\bar{\mathbf{s}}^T\mathbf{D}\mathbf{t} \\
s.t. \quad & t_i < t_{i+1}, \forall i,j = 1,2,\cdots,\left|V^K\right| - 1 \\
& 0 < t_i < \alpha, \forall i = 1,2,\cdots,\left|V^K\right|
\end{aligned} \tag{66}
$$

In formula (66), according to formula (28), $\mathbf{D}$ is a symmetric positive definite matrix. Therefore $\mathcal{J}_2(\mathbf{t})$ is a convex function w.r.t. $\mathbf{t}$. Meanwhile, only linear inequality constraints are involved in the constraint conditions of formula (66). Therefore, formula (66) is a standard convex quadratic programming problem. Let $t^*$ be the optimal solution of formula (66). And then, let $\mathbf{S}^{(D,K)^*} \in \mathbb{R}^{|\mathcal{Y}|\times|\mathcal{Y}|}$, $\forall i,j \in \mathcal{Y}$, we have

$$
s_{i,j}^{(D,K)^*} = \begin{cases} 1, & i = j \\ t_{sort\left(s_{ij}^K; V^K\right)}^*, & i \neq j \end{cases}.
$$

Obviously, $\mathbf{S}^{(D,K)^*}$ satisfies the all constraint conditions of formula (26). Meanwhile, according to **Remark** C.10, $\mathbf{S}^{(D,K)^*}$is the optimal solution of formula (26). So proposition (2) is proven.

In summary, proposition (1) and (2) are proven. So the theorem is proven. $\qquad\square$

# D. Details of Experiments

## D.1. Details of Interpretability Analysis Experiments

*(1) Task.* To show the working mechanism of the proposed method, we conduct the experiments on handwritten digit classification task. The class space is $\mathcal{Y} = \{0, 1, \cdots, 9\}$, which is familiar 10 numbers for human beings. Therefore, it is possible to evaluate the experimental results with common sense.

*(2) Data.* We adopt handwritten digit data set MNIST (LeCun et al., 1998), consisting of 60,000 training samples and 10,000 test samples. Each sample is a $28\times28$ pixel grayscale image containing a handwritten number. For the convenience of presentation, 10 samples were randomly selected for each class to form the training data set.

*(3) Class Knowledge.* In this experiment, we recruit 10 volunteers. each volunteer is asked to construct a $10 \times 10$ rating matrix that is used to measure the similarity between different digits. The score range is $\{0, 1, 2, \cdots, 10\}$. The constructed rating matrix is averaged and normalized, denoted as $\mathbf{S}^K$. The human knowledge system $\mathcal{K}$ is the understanding of visual features of digital characters in the minds of 10 volunteers, and class knowledge $K$ is the rating matrix constructed by 10 volunteers.

*(4) Settings.* Feature extraction network $h : \mathbb{R}^{28\times28} \to \mathbb{R}_+^{10}$ adopts a 5-layer convolutional neural network. FSR network $g : \mathbb{R}_+^{10} \times \mathbb{R}_+^{10} \to [0, 1]$ adopts cosine similarity. We adopt case 2 in formula (6) to obtain the final FSR $\mathbf{T}$. The fuzziness parameters $\alpha = 0.7$, $\beta = 0.9$. No regularization term was used in the experiment. We adopt Adam (Kingma & Ba, 2015) to solve formula (1) and (9), where the learning rate is set as $10^{-3}$. We implement the code based on Pytorch[5] and conduct all experiments on an NVIDIA A100-PCIE-40GB GPU.

## D.2. Details of Generalization Analysis Experiments

To verify the effectiveness of the proposed method, we conduct experiments on 6 public data sets compared with 6 different classifiers.

*(1) Task.* APY and Image-Net1K data sets contain 32 and 1000 classes respectively, involving different classes such as animals, furniture and vehicles. The differences among the classes are large, which are coarse-grained classification tasks. AWA1 and AWA2 data sets share the same class space, which contains 50 species of animals, such as horses, whales, tigers, etc. There are certain differences between the classes. Thus, they are medium-granularity classification tasks. FLO and CUB data sets contain 102 different species of flowers and 200 different species of birds, respectively.

---

[5]https://pytorch.org/

Table 4. The comparison methods and their settings

| Method | Abbreviation | Setting |
|---|---|---|
| K-Nearest Neighbor | KNN | We adopt Euclidean distance. The number of nearest neighbors is selected within $\{1, 3, 5, 7, 9, 11\}$. |
| Decision Tree | DT | We adopt two criterion, i.e., Entropy and Gini, to select split feature. We adopt two strategies, i.e., global search and random split, to determine the split threshold. |
| Support Vector Machine | SVM | We adopt one-to-rest to deal with multiple classification problem. We adopt Gaussian kernel function and the parameters of kernel function are default. The trade-off parameters are selected within $\{0.01, 0.1, 1, 10, 100\}$. |
| Naive Bayes | NB | We adopt four different distribution assumptions to estimate the probability density, i.e., Gaussian, Multinomial, Complement and Bernoulli. |
| Cross Entropy Classifier | CEC | Feature extraction network is designed as $h(\mathbf{x}; \Theta) = ReLU\left(ReLU\left(\mathbf{x}\mathbf{W}_1 + \mathbf{b}_1\right)\mathbf{W}_2 + \mathbf{b}_2\right)\mathbf{W}_3 + \mathbf{b}_3$, where $\Theta = \{\mathbf{W}_1 \in \mathbb{R}^{2048 \times 1024}, \mathbf{b}_1 \in \mathbb{R}^{1024}, \mathbf{W}_2 \in \mathbb{R}^{1024 \times 512}, \mathbf{b}_2 \in \mathbb{R}^{512}, \mathbf{W}_3 \in \mathbb{R}^{512 \times |\mathcal{Y}|}, \mathbf{b}_2 \in \mathbb{R}^{|\mathcal{Y}|}\}$ is the set of learnable parameters. We adopt cross entropy loss of $Softmax\left(h(\mathbf{x})\right)$ and true labels to train network. We adopt Adam (Kingma & Ba, 2015) as optimizer. The size of batch is set to 256. The number of epoch is set to 200. The initial learning rate is set to 1e-3 and decays by 0.5 times every 20 epochs. |
| Fuzzy Learning Machine | FLM | The overall network is $f\left((\cdot, \cdot); \Theta\right) = \frac{\varphi(\cdot;\Theta)^T \varphi(\cdot;\Theta)}{\|\varphi(\cdot;\Theta)\|_2 \times \|\varphi(\cdot;\Theta)\|_2}$, where $\varphi\left(\cdot; \Theta\right) = Sigmoid\left(h\left(\cdot; \Theta\right)\right)$. $h$ is the same as in CEC. We adopt Adam (Kingma & Ba, 2015) as optimizer. The size of batch is set to 1000. The number of epoch is set to 200. The initial learning rate is set to 1e-3 and decays by 0.5 times every 50 epochs. The fuzziness parameters $\alpha = 0.2, \beta = 0.8$. Regularization term is not used in the experiment. |
| HC-HFLM with $K_1$ | $CK_1$-HFLM | The overall network is the same as FLM. The pretrained FLM is used as the initial point. We adopt Adam (Kingma & Ba, 2015) as optimizer. The size of batch is set to 1000. The number of epoch is set to 50. On ImageNet1K data set, the initial learning rate is set to 1e-9. On the remaining data sets, the initial learning rate is set to 1e-3. The learning rate decays by 0.5 times every 30 epochs. For class description vector, we adopt formula (20) in **Appendix** B.1 to calculate the FSR $\mathbf{S}^K$. We adopt case 3 in formula 6 to get final FSR $\mathbf{T}$. On ImageNet1K data set, $\mathbf{S}^K$ is coarsened as 50 levels. On the remaining data sets, $\mathbf{S}^K$ is coarsened as 100 levels. |
| HC-HFLM with $K_2$ | $CK_2$-HFLM | Different from $CK_1$-HFLM, **Algorithm** 1, 2 and 3 are called successively to get FSR $\mathbf{S}^K$ (see **Appendix** B.1). The number of levels is determined by knowledge graph (see Table 2). Based on it, we adopt case2 in formula 6 to get the final FSR $\mathbf{T}$. |

There are small differences between classes. Thus, they are fine-grained classification tasks.

*(2) Data.* For ImageNet1K data set, we adopt common data partition, which contains 1,281,167 training samples and 50,000 test samples. We record the test accuracy of all methods. The remaining 5 data sets are evaluated using 5-fold cross-validation for each method, and the mean accuracy are recorded.

As is known to all, the way of extracting features from images seriously affects the performance of image classification. The focus of this paper is how to integrate knowledge to improve the classification performance rather than designing a better method of extracting features. In order to eliminate the influence of the feature extraction process, the 2048 dimensional features extracted by literature (Xian et al., 2019) are used in the whole experiments for all data sets. The detail information of data sets is summarised in Table 2.

*(3) Class Knowledge.* In this experiment, we adopt two types of knowledge, denoted as $K_1$ and $K_2$.

$K_1$ represents the class description vector. For ImageNet1K data set, the class description vector is obtained by GloVe (Pennington et al., 2014). For the remaining 5 data sets, the class description vectors are designed by domain expert. The detailed descriptions can be found in literature (Xian et al., 2019). We adopt formula (20) in **Appendix** B.1 to calculate class similarity relation $\mathbf{S}^K$.

$K_2$ represents knowledge graph. WordNet (Miller, 1995) is adopted in this experiment. **Algorithm** 1, and 2 and 3 (see

**Appendix** B.1) are called successively to get the class FSR $\mathbf{S}^K$. The class spaces of FLO and CUB data sets are the sets of different followers and different birds, respectively. This makes it difficult to match with WordNet and take advantage of the knowledge provided by WordNet.

*(4) Comparison Methods and Settings.* In this experiment, we adopt 6 different classifiers as comparison methods. K-nearest neighbor (KNN), decision tree (DT), support vector machine (SVM), and naive Bayes (NB) are classical and non-deep neural network methods. Cross entropy classifier (CEC) and FLM are methods based on deep neural network. Because the input features are extracted by the pre-trained deep neural network, we adopt 3 fully-connected layers as backbone for CEC and FLM.

In the proposed HC-HFLM, two types of knowledge $K_1$ and $K_2$ are integrated into FLM, respectively, denoted as $CK_1$-HFLM and $CK_2$-HFLM. For fair comparison, the network architecture is the same as the FLM.

For each method, different parameter settings are adopted and the best experimental results are recorded. For the first four methods, we adopt the implementations by the Scikit-learn library[6]. The remaining methods are implemented based on Pytorch. We conduct all experiments on an NVIDIA A100-PCIE-40GB GPU. More details are listed in Table 4.

---

[6] https://scikit-learn.org/stable/index.html

