# OpenReview forum: "Human Cognition-Inspired Hierarchical Fuzzy Learning Machine"
_ICML.cc/2025/Conference — ICML 2025 poster_

### Official Review · Reviewer_qLb8 · 2025-03-10

**Overall Recommendation:** 3

**Summary:**

The paper extracts the similarities between concepts from the human knowledge system and uses these similarities to guide the learning process. As a result, the similarities between concepts are integrated into the sample similarity, thereby improving model performance. Meanwhile, the paper guarantees the effectiveness of the proposed method through theoretical analysis.

**Claims And Evidence:**

Yes. The claims made in the paper are supported by theoretical and experimental evidence.

**Essential References Not Discussed:**

To the best of my knowledge, no important reference is missing.

**Experimental Designs Or Analyses:**

Yes. I checked the experiments and corresponding analysis in the paper. Specifically, the experimental results in Section 5.1 are consistent with the conclusions in Corollary 4.13. Meanwhile, the experimental results in Section 5.2 indicate that by incorporating class related knowledge, the proposed method can improve the generalization performance of the model, and the higher the quality of the knowledge, the greater the performance improvement.

**Methods And Evaluation Criteria:**

Yes. The proposed method is effective. And the evaluation criteria can verify the advantages of the proposed method

**Other Comments Or Suggestions:**

1. The fonts in Figure 1, Table 1, and Figure 2 should be enlarged.

**Other Strengths And Weaknesses:**

The strengths of the paper are as follows.

1.	In classification problems, each class corresponds to a concept. Unlike most existing classifiers, this paper views the classification problems as the concept cognition problems and seeks universal principles from human cognition to improve classification performance.

2.	The paper develops the fuzzy similarity relation-based quotient space theory and analyses the working mechanism of the proposed method based on this theory. These theories guarantee the interpretability and effectiveness of the proposed method.

3.	The experiments in the paper also verify the interpretability and effectiveness of the proposed method.

The weaknesses of the paper are as follows.

1.	The paper does not discuss the difference between the proposed method and CLIP, which is the typical method to align image and text.

2.	The paper does not provide a detailed explanation on how to coarsen the low-quality knowledge. Specifically, what is the definitions of f_{coa} in formula (25)?

**Questions For Authors:**

Currently, multi-modal pre-trained models, e.g., CLIP, have been proposed, which aims to align image and textual information. What is the difference between these methods and the proposed methods?

**Relation To Broader Scientific Literature:**

The paper develops the fuzzy similarity relation-based quotient space theory and establishes the connection between fuzzy similarity relation-based quotient space theory and fuzzy equivalence relation-based quotient space theory through Theorem 4.5, which enriches the theory of fuzzy quotient space.
At the same time, the paper proposes a universal method for integrating human knowledge into the machine learning model, providing new and promising idea for developing human cognition-inspired machine learning methods.

**Theoretical Claims:**

Yes. I checked the main theoretical results of the paper, including Theorem 3.3, 4.11-4.13, and these conclusions are correct. Meanwhile, Section 5.1 provides a toy example which validates the conclusions in Theorem 4.11,4.12 and Corollary 4.13 step by step.

---

> ### Author Rebuttal · Authors · 2025-03-29
>
> # 0. General Response
> We sincerely appreciate the reviewer’s positive feedback and valuable comments. Below, we provide a point-by-point response to each comment.
>
> # 1. Response to “Weaknesses”
> (1) CLIP (Contrastive Language-Image Pre-training) is a self-supervised learning method that learns representations of images and texts by aligning them in a shared latent space. The proposed method differs from CLIP in several aspects.
>
> - **Learning Paradigm.** CLIP is a self-supervised learning method, whereas the proposed method follows a fully supervised learning paradigm.
>
> - **Interpretability.**  CLIP maps images and texts into a shared Euclidean space with specified dimensions and then aligns them. However, the individual dimension of this space lacks explicit semantic meaning, making the alignment process less interpretable. In contrast, the proposed method establishes alignment between the hierarchical structures derived by class knowledge and data in the quotient space. Each component in the quotient space carries a well-defined semantic meaning, leading to a  interpretable alignment process.
>
> - **Scaling Effect.** CLIP is trained on large-scale image-text pairs, enabling good performance across various downstream tasks. Similarly, the proposed method can be extended to large-scale datasets to learn more generalized representations, enhancing its ability to handle diverse downstream tasks effectively.
>
> - **Application Scenario.** The success of CLIP is largely attributed to the widespread availability of image-text pairs on the Internet. However, in many real-world applications, acquiring large-scale data along with corresponding textual descriptions is challenging. In such cases, leveraging class knowledge to enhance the model’s ability to understand concepts becomes crucial.
>
> (2) Formula (25) defines a class of functions, where any function satisfying the given conditions can serve as a coarsening function. This function eliminates subtle difference in the original class fuzzy similarity relation, preserving a robust and salient ranking of class similarities.
>
> # 2. Response to “Other Comments Or Suggestions”
> In the final version, we will enlarge the fonts in Figures 1, 2 and Table 2 to enhance readability and provide greater convenience for the readers.
>
> # 3. Response to “Questions For Authors”
> See item (1) in **Response to “Weaknesses”**.

---

### Official Review · Reviewer_2rzm · 2025-03-11

**Overall Recommendation:** 4

**Summary:**

In general terms, the paper presents a new and innovative method called Human Cognition-Inspired Hierarchical Fuzzy Learning Machine (HC-HFLM), which is aimed at improving interpretability and performance in classification tasks. This method is based on human cognition and takes into account the ambiguity present in real-world concepts, which cannot always be precisely defined.

## Update after rebuttal:
Thank you to the authors for their detailed responses; my doubts and questions have been resolved.

**Claims And Evidence:**

During the development of the paper the authors demonstrate theoretically that aligning data and knowledge structures improves interpretability and performance. Furthermore, the authors present experiments on various datasets and using various classifiers for comparison. The experiments show promising results compared to traditional classifiers, suggesting that it may have potential applications in open-world learning.

**Essential References Not Discussed:**

The references are good but more recent work integrating structured knowledge bases into neural networks is not mentioned. For example, representation-enhanced neural knowledge integration (Liu et al., 2024) and logical neural networks for knowledge base completion with embeddings and rules (Sen et al., 2022). In addition, there is an interesting and recently published paper on hierarchical representation learning (Yu et al., 2024) which also aligns representations learned via embedding hierarchical tree structures of concepts, it would be good to include it.


References:

Liu, S., Cai, T., and Li, X. Representation-Enhanced Neural Knowledge Integration. arXiv preprint arXiv:2410.07454, 2024.

P. Sen, B. W. Carvalho, I. Abdelaziz, P. Kapanipathi, S. Roukos, and A. Gray, “Logical Neural Networks for Knowledge Base Completion with Embeddings & Rules,” in Proceedings of the 2022 Conference on Empirical Methods in Natural Language Processing, Abu Dhabi, United Arab Emirates, Dec. 2022, pp. 3863–3875.

Yu, J., Zhang, C., Hu, Z. and Ji, Y. Embedding Hierarchical Tree Structure of Concepts in Knowledge Graph Embedding. Electronics. 2024

**Experimental Designs Or Analyses:**

In relation to the evaluation of the robustness and validity of the experimental designs and analyses presented, the paper uses well known public datasets such as MNIST, APY, ImageNet1K, AWA1, AWA2, FLO and CUB. These datasets are suitable for the classification task that is studied, but it should be mentioned if any preprocessing was performed, as it may affect the results. Further, the paper compares HC-HFLM with several traditional and also deep neural network based classifiers, this provides a solid benchmark to evaluate the performance, however the variability of the results is not discussed. Additionally, the results show an improvement in generalization, however it would be useful to discuss how the method performs on noisy data to better evaluate the robustness of HC-HFLM.

**Methods And Evaluation Criteria:**

In this paper, the methods and evaluation criteria are appropriate and well justified for the problem studied. The authors also make use of several datasets and a comparison with several classifiers to validate the effectiveness of the proposed method, however it would be beneficial to include, in addition to accuracy, an evaluation of sensitivity, precision and F1-score. Finally, it would be very interesting to discuss limitations and scenarios in which the method might not be as effective.

**Other Comments Or Suggestions:**

- In the introduction section, near line 051, “.. employs exemplar theory to capture the typicality effects (Smith et al., 1974; Rosch, 1975)”, you should quote (Medin & Schaffer, 1978) instead of (Rosch, 1975).

- Figure 2 looks very small and misses the details, which makes it difficult to understand, it is recommended to present a larger image.

- In section 5.2, near line 371, it says “WordNet (Miller, 1995) (CK1)”, but it should read “WordNet (Miller, 1995) (CK2)”.

- In section 5.2, near line 376, it is recommended that you name the techniques instead of their abbreviations (KNN, DT, SVM and NB). They are described in the supplementary material but it would be good to write their full names when mentioning them for the first time.

- In Table 3, the symbol “--” means that it takes more than 7 days for training? Also, what does N/A mean in this context?

**Other Strengths And Weaknesses:**

Strengths:

- The methodology presented by the authors has a considerable degree of originality as it uses fuzzy similarity along with human cognition for classification.

- The paper contributes significantly to the use of fuzzy learning machine, quotient spaces and knowledge based approaches for better classification methods.

- The paper addresses a problem present in classification methods, which is to assume that classes are well defined.

- The authors present well-defined experiments on 6 datasets, showing an improvement over classical classification methods.

- The presentation of the paper is well articulated and includes demonstrations that support its formulation.


Weaknesses:

- It would be useful to discuss how the method performs with noisy data and thus better evaluate its robustness to HC-HFLM.

- It would be beneficial to include, in addition to accuracy, an evaluation of sensitivity, precision and F1-score.

- It would be very interesting to discuss limitations and scenarios in which the method might not be as effective.

- It would be useful to include details on training efficiency, computational requirements and an explicit comparison of execution time with other methods.

**Questions For Authors:**

- The proposed method is intended for classification, but could it be expanded for other machine learning tasks?

- What are the limitations of the proposed method and under which scenarios is it not so effective?

**Relation To Broader Scientific Literature:**

The paper is based on theories of cognitive science, especially focusing on the ambiguity of concepts and context dependence (Wittgenstein, 1953; Rosch, 1975). Within the context of ML, the authors expand the idea of using human cognition of concepts to improve performance, as explored in previous work (Cui and Liang, 2022) with their Fuzzy Learning Machine (FLM). In addition the paper introduces fuzzy similarity relations (FSR) to model class knowledge, based on the fuzzy set theory (Zadeh, 1965), this allows capturing the ambiguity of concepts. The proposed hierarchical alignment loss integrates this knowledge into the learning process. This extends previous work on hierarchical learning (Silla & Freitas, 2011; Wang et al., 2020). Finally the paper develops the FSR-based quotient space theory for modeling data and knowledge, this expands previous work on quotient spaces (Zhang & Zhang, 2014).


References:

Wittgenstein, L. Philosophical Investigations. Blockwell Publishing, 1953.

Rosch, E. Cognitive representations of semantic categories. Journal of Experimental Psychology: General, 104(3): 192, 1975.

Cui, J. and Liang, J. Fuzzy learning machine. In Advances in Neural Information Processing Systems, pp. 3669336705, 2022.

Zadeh, L. A. Fuzzy sets. Information and Control, 8(3): 338–353, 1965.

Silla, C. N. and Freitas, A. A. A survey of hierarchical classification across different application domains. Data Mining and Knowledge Discovery, 22:31–72, 2011.

Wang, Y., Hu, Q., Zhu, P., Li, L., Lu, B., Garibaldi, J. M., and Li, X. Deep fuzzy tree for large-scale hierarchical visual classification. IEEE Transactions on Fuzzy Systems, 28(7):1395–1406, 2020.

Zhang, L. and Zhang, B. Quotient Space Based Problem Solving: A Theoretical Foundation of Granular Computing. Tsinghua University Press, Beijing, China, 2014.

**Theoretical Claims:**

The paper contains several formulations and demonstrations, which are developed in detail in the annexes. Personally I have briefly reviewed the main theoretical claims and have not found any issues. Several demonstrations are extensive and demand a detailed review, which is time limiting.

---

> ### Author Rebuttal · Authors · 2025-03-30
>
> # 0. General Response
> We thank reviewer for the appreciation of our work and valuable  comments. Below, we provide a point-by-point response to each comment.
> # 1. Response to “Methods And Evaluation Criteria”
> (1) We add the standard deviation of accuracy for all methods based on 5-fold cross-validation, except for ImageNet1K dataset. The experimental results are as follows.
> ||APY|AWA1|AWA2|FLO|CUB|
> |:--:|:--:|:--:|:--:|:--:|:--:|
> |KNN|85.35$\pm$0.46|86.61$\pm$0.51|89.83$\pm$0.28|83.39$\pm$1.00|47.30$\pm$1.07|
> |DT|63.13$\pm$0.76|63.47$\pm$0.45|70.09$\pm$0.56|42.65$\pm$0.81|21.64$\pm$1.07|
> |SVM|84.54$\pm$0.55|84.26$\pm$0.56|89.06$\pm$0.12|86.56$\pm$1.26|43.89$\pm$0.74|
> |NB|76.13$\pm$1.06|84.67$\pm$0.70|87.68$\pm$0.18|85.53$\pm$1.22|60.27$\pm$0.27|
> |CEC|89.08$\pm$0.72|88.48$\pm$0.14|91.67$\pm$0.35|93.58$\pm$0.91|61.49$\pm$2.47|
> |FLM|89.42$\pm$0.36|89.95$\pm$0.29|92.79$\pm$0.31|94.05$\pm$0.83|66.19$\pm$0.92|
> |**CK$_1$-HFLM**|**90.23$\pm$0.42**|**91.10$\pm$0.28**|**93.59$\pm$0.17**|**95.06$\pm$0.65**|**68.78$\pm$0.27**|
> |**CK$_2$-HFLM**|90.21$\pm$0.32|90.87$\pm$0.27|93.35$\pm$0.26|N/A|N/A|
>
> The results demonstrate that the proposed method outperforms all compared methods and exhibits a small standard deviation, indicating the stability of the performance.
>
> (2) For binary classification, the hierarchical structure on class space is unique. In this case, the proposed method degenerates into fuzzy learning machine, where class knowledge only guides the selection of fuzzy parameters. Additionally, when the quality of class knowledge is poor or not relevant to the task, the performance gain brought by the proposed method will be limited.
> # 2. Response to “Experimental Designs Or Analyses”
> (1) For fairness in comparison, the 2048-dimensional features and class description vectors used for all datasets are sourced from (Xian et al., 2019) without any additional preprocessing.
>
> (2) See item (1) in Section 1.
>
> (3) Due to time constraints, we conduct label noise experiments on the APY dataset. We introduce label noises with 4 different ratios, i.e., 5%, 10%, 15%, and 20%. For each ratio, label noises are added randomly and the process is repeated 5 times. The experimental results are as follows.
> ||5%|10%|15%|20%|
> |:---:|:---:|:---:|:---:|:---:|
> |KNN|85.17|84.80|84.22|83.47|
> |DT|59.05|56.22|52.91|49.20|
> |SVM|77.44|76.57|75.70|74.72|
> |NB|75.07|74.72|74.46|74.17|
> |CEC|88.81|87.42|85.79|83.94|
> |FLM|88.93|86.57|85.70|85.57|
> |**CK$_1$-HFLM**|**89.75**|**89.26**|**88.62**|**88.02**|
> |**CK$_2$-HFLM**|89.61|89.17|88.53|87.87|
>
> The experimental results show that as the label noise ratio increases, the proposed method experiences the least performance degradation, highlighting its robustness against the label noise.
> # 3. Response to “Essential References Not Discussed”
> Thank you for the important literatures. Recent methods have successfully integrated structured knowledge with neural networks for knowledge graph representation and completion task. We will add the related discussions in the final version.
> # 4. Response to “Weaknesses”
> (1) See item (3) in Section 2.
>
> (2) See item (1) in Section 1.
>
> (3) See item (2) in Section 1.
>
> (4) We report the average time (s) for running one epoch (mean over 50 epochs) for CEC, FLM, and the proposed method. Due to challenges in utilizing GPU acceleration for the other comparison methods, their running times are not comparable. All experiments are conducted on an NVIDIA A100-PCIE-40GB GPU, and the results are as follows.
> ||FLO|CUB|APY|AWA1|AWA2|ImageNet1K|
> |:---:|:---:|:---:|:---:|:---:|:---:|:---:|
> |CEC|0.29|0.33|0.51|0.99|1.10|88.29|
> |FLM|0.37|0.52|0.72|1.33|1.63|86.17|
> |**CK$_1$-HFLM**|0.78|1.12|1.55|2.92|3.52|169.50|
> |**CK$_2$-HFLM**|N/A|N/A|0.94|1.77|2.18|113.61|
>
> CEC computes the loss based on single sample, while FLM calculates the loss based on sample pair, leading to a longer runtime compared to CEC. The proposed method builds on FLM by incorporating class knowledge and further refining class similarities into different levels, leading to a longer runtime than FLM. Overall, the runtime of the proposed method is approximately 2-3 times that of CEC, which remains within an acceptable range.
>
> # 5. Response to “Other Comments Or Suggestions”
> Thank you for your thoughtful correction. We will incorporate the corresponding modifications in the final version. Additionally, N/A denotes the absence of the corresponding type of knowledge for these datasets, and we will include this explanation in the main paper.
> # 6. Response to “Questions For Authors”
> (1) The proposed method effectively models the hierarchical structure, positioning it as a promising approach for knowledge graph representation learning. It is also applicable to structured prediction tasks, such as ranking and hierarchical classification. Additionally, it provides an interpretable solution for aligning two information sources, making it well-suited for multimodal learning tasks.
>
> (2) See item (2) in Section 1.

---

### Official Review · Reviewer_DRfi · 2025-03-12

**Overall Recommendation:** 4

**Summary:**

The authors propose a human cognition-inspired classifier. The method first mines the fuzzy similarity relation between concepts from human knowledge system. And then the authors design the hierarchical alignment loss based on the principles of concept cognition. Using this loss, the fuzzy similarity relation between concepts guides the learning process. At the same time, the authors demonstrate that minimizing the hierarchical alignment loss can achieve hierarchical structures in class space and sample space, thereby aligning data and knowledge in the quotient space. Finally, the experiments verify the advantages of the proposed method in interpretability and improving the generalization performance.

**Claims And Evidence:**

The claims in this paper are supported by clear and convincing evidence.

**Essential References Not Discussed:**

As far as I know, no key reference is missing.

**Experimental Designs Or Analyses:**

The experimental results in Section 5.1 verify the theoretical conclusions in Section 4. By minimizing the proposed hierarchical alignment loss, the hierarchical structure on the training set is indeed aligned with the hierarchical structure on the class space.

**Methods And Evaluation Criteria:**

The proposed method makes sense for the problem at hand. So dose the evaluation criteria.

**Other Comments Or Suggestions:**

1.	In order to make readers better follow the idea of the paper, some content in the Appendix should be adjusted to the main text, especially the relevant processing procedures involved in formulas (5) and (6).

2.	The font in Figures 1 and 2 should be enlarged. The same issue also appears in Figure 3 of the Appendix.

**Other Strengths And Weaknesses:**

**Strengths**

1.	The proposed method is novel. The authors use cognitive science principles to guide the design of the model, improving its interpretability and providing a new solution for the development of humanoid intelligence.

2.	The theoretical analysis is sound. The authors propose a new hierarchical alignment loss to align training data and class knowledge, and this loss has the theoretical guarantee in aligning data and knowledge.

3.	The experimental results are sufficient. The authors not only provide an example of hierarchical alignment in the quotient space, but also verify the effectiveness of the proposed method on the public datasets.

**Weaknesses**

1.	The authors develop the fuzzy similarity relation-based quotient space theories. In order to help readers understand the thought process of the paper, the connection between these new theories and existing quotient space theories as well as the process of developing these new theories should be further discussed.

**Questions For Authors:**

As stated in the paper, the fuzzy learning machine also solves the classification problems from the view of concept cognition. Why can the proposed method significantly improve the performance of the fuzzy learning machine?

**Relation To Broader Scientific Literature:**

The proposed method achieves the alignment between hierarchical structures of data and knowledge. What's more, each component in these hierarchical structures has clear semantics, so this alignment has strong interpretability. In many practical applications, aligning data and knowledge is a common requirement. Therefore, the proposed method is expected to achieve successful applications in more scenarios.

**Theoretical Claims:**

I checked the relevant proofs. The authors provide detailed proof steps. Given these proofs, the conclusion of the paper can be verified. Meanwhile, the lower part of Figure 1 presents a clear example. As stated in the main conclusion i.e., Corollary 4.13, the hierarchical structure corresponding to data is aligned with the hierarchical structure corresponding to class knowledge.

---

> ### Author Rebuttal · Authors · 2025-03-29
>
> # 0. General Response
> We thank reviewer for the appreciation of our work and valuable comments. Below, we provide a point-by-point response to each comment.
>
> # 1. Response to “Weakness”
>
> - The fuzzy equivalence relation (FER) is a more restrictive form of the fuzzy similarity relation (FSR).
> In practical applications, obtaining the FER is generally more challenging than obtaining the FSR.
> Therefore, we employ the FSR to model class knowledge and sample similarity.
>
> - The existing fuzzy quotient space theory is based on the FER and is therefore not suitable for analyzing the method proposed. In response, we develop a quotient space theory based on the FSR.
>
> - **Theorem** 4.5 establishes the connection between the quotient spaces derived by FSR and FER, which allows the existing
>  theoretical results in FER-based quotient space to be extended to FSR-based quotient space.
>
> We will incorporate the above content in the final version.
>
> # 2. Response to “Other Comments Or Suggestions”
>
> 1. In the final version, we will incorporate the relevant content into the main paper to improve the overall coherence and readability.
>
> 2. In the final version, we will enlarge the figures in this paper to enhance readability and provide greater convenience for the readers.
>
> # 3. Response to “Questions For Authors”
>
> Unlike fuzzy learning machine, the proposed method further incorporates the class knowledge contained in human knowledge system into the sample fuzzy similarity relation. As a result, the proposed method enhances the model's understanding of concepts, leading to the performance gain.

---

### Official Review · Reviewer_CTni · 2025-03-12

**Overall Recommendation:** 3

**Summary:**

This paper advocates solving classification problems from the perspective of concept cognition. Inspired by human cognition, this paper utilizes the relationships between concepts embedded in the human knowledge system to guide the learning process. This deepens the model's understanding of concepts. In addition, this paper develops the fuzzy similarity relation-based quotient space theory and then analyses the working mechanism of the proposed method. Meanwhile, the experimental results demonstrate that the proposed method can improve the generalization performance by incorporating human knowledge system.

**Claims And Evidence:**

The main claims in this paper are supported by theoretical analysis.

**Essential References Not Discussed:**

The references are complete and sufficient for understanding this paper.

**Experimental Designs Or Analyses:**

The experimental results in Section 5.2 demonstrate that the proposed method can deepen the model's understanding of concepts by incorporating human knowledge system, thereby improving the model's testing accuracy.

**Methods And Evaluation Criteria:**

The proposed method is effective for aligning data and class knowledge and then improving the generalization performance.

**Other Comments Or Suggestions:**

In order to enhance the readability of this paper, some content in the Appendix should be placed in the main text, such as the definition of “finer” and “coarser”, and the definition of “cut-relation”, etc.

**Other Strengths And Weaknesses:**

Strengths:

1.	In human cognition, there are rich relationships between concepts. Inspired by this, this paper integrates these relationships between concepts contained in human knowledge into the model, which improves the interpretability and generalization performance. It provides a new inspiring way to develop human cognition-inspired machine learning methods.

2.	This paper has a solid theoretical analysis, which guarantees the effectiveness of the proposed method.

3.	The proposed method models different types of knowledge as fuzzy similarity relation on class space, which has good universality.

Weaknesses:

1.	In formula (9), the specific form of regularization term R (\Theta) and the value of the hyper-parameter \gamma are not given, which affects the reproducibility of the experiments.

**Questions For Authors:**

Section 5.2 adopts fully connected networks as the backbone network for the deep neural network methods, the fuzzy learning machine, and the proposed method. As is known to all, the fully connected network is the simplest network structure. Why not

**Relation To Broader Scientific Literature:**

Many practical problems can be abstracted as aligning two pieces of information from different sources. Most existing methods achieve the alignment in Euclidean space. Unlike it, this paper achieves the alignment of two pieces of information in quotient space, and this alignment has clear semantics and strong interpretability. Therefore, the proposed method provides a new promising approach for aligning information from different sources.

**Theoretical Claims:**

I have checked the main conclusion, i.e. Corollary 4.13. The relevant proof process should be correct. Meanwhile, Figure 2 in Section 5.1 provides a typical example of this conclusion.

---

> ### Author Rebuttal · Authors · 2025-03-29
>
> # 0. General Response
>
> We sincerely appreciate the reviewer’s positive feedback and valuable comments. Below, we provide a point-by-point response to each comment.
>
> # 1. Response to “Weakness”
>
> - In the experiments, to directly highlight the performance improvement achieved by incorporating class knowledge contained in human knowledge system, the proposed method does not include the regularization term.
> - In practical applications, appropriate regularization term can be introduced based on domain knowledge, which is expected to further enhance performance.
>
> # 2. Response to “Other Comments Or Suggestions”
>
> To improve the overall coherence and readability, we will incorporate the relevant content into the main paper in the final version.
>
> # 3. Response to “Questions For Authors”
>
> The primary goal of the experiments is to demonstrate that the proposed method improves model performance by incorporating the class knowledge contained in human knowledge system rather than merely aiming for higher accuracy. As the reviewer pointed out, employing more complex and advanced neural networks could potentially yield better accuracy. However, the subject lies beyond the scope of this paper.

---

### Decision · Program_Chairs · 2025-05-01

**Decision:**

Accept (poster)

**Comment:**

This paper received four effective reviews, and all of them are positive. Overall, the paper is of good quality and should be accepted.